# The Future of the Correlated Electron Problem

A. Alexandradinata,[1,2] N.P. Armitage,[3,*] Andrey Baydin,[4] Wenli Bi,[5] Yue Cao,[6] Hitesh J. Changlani,[7,8] Eli Chertkov,[1,2] Eduardo H. da Silva Neto,[9,10] Luca Delacretaz,[11] Ismail El Baggari,[12] G.M. Ferguson,[12] William J. Gannon,[13] Sayed Ali Akbar Ghorashi,[14] Berit H. Goodge,[15] Olga Goulko,[16,17] Gaël Grissonnanche,[18] Alannah Hallas,[19] Ian M. Hayes,[20] Yu He,[21,22] Edwin W. Huang,[1,2] Anshul Kogar,[23] Divine Kumah,[24] Jong Yeon Lee,[25] Anaëlle Legros,[3] Fahad Mahmood,[26,2] Yulia Maximenko,[2,26] Nick Pellatz,[27] Hryhoriy Polshyn,[28] Tarapada Sarkar,[20] Allen Scheie,[29] Kyle L. Seyler,[30] Zhenzhong Shi,[31] Brian Skinner,[32] Lucia Steinke,[33,8] Komalavalli Thirunavukkuarasu,[34] Thaís Victa Trevisan,[35] Michael Vogl,[36] Pavel A. Volkov,[37] Yao Wang,[38] Yishu Wang,[3] Di Wei,[39,40] Kaya Wei,[8] Shuolong Yang,[41] Xian Zhang,[42] Ya-Hui Zhang,[25] Liuyan Zhao,[43] and Alfred Zong[44,45]

[1] *Institute of Condensed Matter Theory, University of Illinois at Urbana-Champaign, Urbana, 61801 IL, USA*

[2] *Department of Physics, University of Illinois at Urbana-Champaign, Urbana, 61801 IL, USA*

[3] *The Institute for Quantum Matter and the Department of Physics and Astronomy, The Johns Hopkins University, Baltimore, MD 21218, USA*

[4] *Department of Electrical and Computer Engineering, Rice University, Houston, TX 70005, USA*

[5] *Department of Physics, University of Alabama at Birmingham, Birmingham, AL 35294, USA*

[6] *Materials Science Division, Argonne National Laboratory, Lemont, IL 60439, USA*

[7] *Department of Physics, Florida State University, Tallahassee, FL 32306, USA*

[8] *National High Magnetic Field Laboratory, Tallahassee, FL 32304, USA*

[9] *Department of Physics, University of California, Davis, CA 95616, USA*

[10] *Department of Physics, Yale University, New Haven, CT 06511, USA*

[11] *Kadanoff Center for Theoretical Physics, University of Chicago, Chicago, IL 60637, USA*

[12] *Department of Physics, Cornell University, Ithaca, NY 14853, USA*

[13] *Department of Physics and Astronomy, University of Kentucky, Lexington, KY 40506, USA*

[14] *Department of Physics, William & Mary, Williamsburg, VA 23187, USA*

[15] *School of Applied and Engineering Physics, Cornell University, Ithaca, NY 14853, USA*

[16] *Boise State University, Department of Physics, Boise, ID 83725, USA*

[17] *Department of Physics, University of Massachusetts Boston, Boston, MA 02125, USA*

[18] *Laboratory of Atomic and Solid State Physics, Cornell University, Ithaca, NY 14853, USA*

[19] *Department of Physics and Astronomy and Quantum Matter Institute, University of British Columbia, Vancouver, B.C., Canada V6T 1Z1*

[20] *Department of Physics, Maryland Quantum Materials Center, University of Maryland, College Park, MD 20742, USA*

[21] *Department of Physics, University of California at Berkeley, Berkeley, CA 94720, USA*

[22] *Department of Applied Physics, Yale University, New Haven, CT 06511, USA*

[23] *Dept. of Physics and Astronomy, Univ. of California at Los Angeles, Los Angeles, CA 90095, USA*

[24] *Department of Physics, North Carolina State University, Raleigh, NC 27695, USA*

[25] *Department of Physics, Harvard University, Cambridge, MA 02138, USA*

[26] *F. Seitz Materials Research Laboratory, University of Illinois at Urbana-Champaign, Urbana, IL 61801, USA*

[27] *Department of Physics, University of Colorado, Boulder, CO 80309, USA*

[28] *Department of Physics, University of California, Santa Barbara, CA 93106, USA*

[29] *Neutron Scattering Division, Oak Ridge National Laboratory, Oak Ridge, TN 37831, USA*

[30] *Department of Physics, California Institute of Technology, Pasadena, CA 91125, USA*

[31] *Department of Physics, Duke University, Durham, NC 27708, USA*

[32] *Department of Physics, Ohio State University, Columbus, OH 43210, USA*

[33] *Department of Physics, University of Florida, Gainesville, FL 32611, USA*

[34] *Department of Physics, Florida Agricultural and Mechanical University, Tallahassee, FL 32307, USA*

[35] *Ames Laboratory, Ames, IA 50011, USA*

[36] *Department of Physics, King Fahd University of Petroleum and Minerals, 31261 Dhahran, Saudi Arabia*

[37] *Department of Physics and Astronomy, Center for Materials Theory, Rutgers University, Piscataway, NJ 08854, USA*

[38] *Department of Physics and Astronomy, Clemson University, Clemson, SC 29631, USA*

[39] *Geballe Laboratory for Advanced Materials, Stanford University, Stanford, CA 94305, USA*

[40] *Department of Applied Physics, Stanford University, Stanford, CA 94305, USA*

[41] *Prtizker School of Molecular Engineering, The University of Chicago, Chicago, IL 60637, USA*

[42] *Department of Mechanical Engineering, Stevens Institute of Technology, Hoboken, NJ 07030, USA*

[43] *Department of Physics, University of Michigan, Ann Arbor, MI 48109, USA*

[44] *Department of Physics, Massachusetts Institute of Technology, Cambridge, MA 02139, USA*

[45] *Department of Chemistry, University of California Berkeley, Berkeley, CA 94720, USA*

(Dated: July 15, 2022)

arXiv:2010.00584v3 [cond-mat.str-el] 13 Jul 2022

A central problem in modern condensed matter physics is the understanding of materials with strong electron correlations. Despite extensive work, the essential physics of many of these systems is not understood and there is very little ability to make predictions in this class of materials. In this manuscript we share our personal views on the major open problems in the field of correlated electron systems. We discuss some possible routes to make progress in this rich and fascinating field. This manuscript is the result of the vigorous discussions and deliberations that took place at Johns Hopkins University during a three-day workshop January 27, 28, and 29, 2020 that brought together six senior scientists and 46 more junior scientists. Our hope, is that the topics we have presented will provide inspiration for others working in this field and motivation for the idea that significant progress can be made on very hard problems if we focus our collective energies.

## Contents

The model of non-interacting electrons is well established in solid-state physics. It is a remarkable fact of nature that, for many materials, the effects of electron-electron interactions can be best captured by *ignoring* the correlations they produce. In other systems, interactions can often be included as a perturbation and manifest through renormalizing parameters such as the effective mass, without altering the qualitative behavior. Such systems can be adiabatically connected to an interaction-free system. There are, however, other materials whose properties explicitly manifest strong interactions, which adiabatic connection to an interaction-free system is not possible, or is not useful. Such strongly correlated electron systems host a tremendous variety of fascinating macroscopic phenomena including high-temperature superconductivity, quantum spin-liquids, fractionalized topological phases, and strange metals. Despite many years of intensive work, the essential physics of many of these systems is still not understood, and we do not have an overall perspective on strong electron correlations. Moreover, our predictive power for such systems is lacking. This topic is central to a broader range of scientific disciplines, such as atomic and molecular physics, nuclear and high energy physics, astrophysics, and chemistry, where many-body effects are significant. Despite decades of intensive research, there has been relatively limited progress on an overall picture. Is a unified perspective even possible? Or is the "Anna Karenina Principle" in effect[1] – all non-interacting systems are alike; each strongly correlated system is strongly correlated in its own way?

In thinking about the future of the correlated electron problem, myriad questions abound. Is there a general definition of a strongly correlated material? Is a general framework to understand strong electronic correlations possible? Are numerical approaches essential? Can we develop general frameworks to better make predictions? What new experiments can we design that give essential insight to heretofore unrecognized correlations? Is "hidden order" ubiquitous? Can we hope to understand exotic superconductors in the same way we understand conventional superconductors? What would a "solution" to the "problem" even look like? Is there *a problem*? Or are there *many problems*? What is the future of correlated electrons? In searching for "the future" should we

*Electronic address: npa@jhu.edu

[1] https://en.wikipedia.org/wiki/Anna_Karenina_principle

come back to the possible avenues not fully explored in the past, or invest in completely new directions, or do both?

On January 27, 28, and 29, 2020, a workshop (organized by NPA) was held at The Johns Hopkins University to try to answer these and other questions[2]. Six senior scientists gave lectures on the first day on their ideas to solve parts of the correlated electron problem[3]. On days 2 and 3, 46 more junior scientists brainstormed, debated, and wrote about their different approaches to understanding correlated electrons. This manuscript is the result of those vigorous deliberations. This manuscript was written through collaborative writing software such as Google Docs, Slack, and Overleaf. Subject topics and the general format was suggested by NPA, but the ultimate topics were chosen by consensus on the morning of the second day. 80% of the text was written collectively in the first 72 hours after the workshop. All 47 coauthors contributed to the writing and proofing, both at the workshop and afterwards. NPA edited this manuscript.

It is important to note that this manuscript is *not* a review and no attempt to be complete has been made. The topics and opinions expressed are idiosyncratic, and reflect the particular interests and preferences of the people who spoke at and attended the workshop. It presents their collective vision for the future of the correlated electron problem. We hope that this document can serve as a starting point for further debate. And although we do our best to anticipate what directions will be important in the future, we do so with the full expectation (and hope) that much of the below will become irrelevant as some person in some laboratory somewhere in the world will look at some new data coming out of a new experiment on a new material and say, "That's funny..."[4]. And we will learn even more about the incredible numbers of ways that electrons can behave in solids.

## I. WHAT IS "THE" PROBLEM?

There is no consensus on the role of strong electron correlations in solids. Moreover, at present, there is no agreed single definition as to what constitutes *the* correlated electron problem. As such, for the purposes of this manuscript, we adopt the following working defini-

tion: *a correlated electron problem is one in which interactions are so strong or have a character such that theories based on the underlying original "bare" particles fail even qualitatively to describe the material properties.* These original free "particles" of a strongly correlated system could be electrons, or spin-flips, or local vibrations. For instance, exactly solvable models with emergent free quasiparticles (*e.g.* the Kitaev spin liquid (Kitaev, 2006)) *are* strongly correlated by this definition[5]. The optimistic hope is that a large class of such problems can be understood using a set of similar underlying principles, which are as of yet not understood by us. We believe that such principles should either provide us with a blueprint for a robust predictive power for material realizations *or* an understanding of why this is not possible. While it may be possible to arrive at such principles by general reflection, our judgment is that a general "solution" to the strongly correlated problem will most likely be identified in the common features of candidate solutions to individual strongly correlated "problems." Far from being a mere after-thought to those specific solutions, a general principle arrived at in this way should provide the predictive power needed to find new useful materials with interesting phenomena. In this spirit, we will review some current problems in condensed matter physics that we feel most likely to be fertile in this regard.

As this workshop was intended to be forward-looking, we present below topics that we see as representative of the *future* of the correlated electron problem. The discovery in 1986 of cuprate high-temperature superconductivity (Bednorz and Müller, 1986) was not the start of this field, but gave strong impetus to a vast number of researchers to join it. We thus introduce (A) the field of correlated superconductors at the outset. We then follow with a survey of (B) quantum spin liquids and (C) strange metals, which grew initially from the field of correlated superconductivity, but which currently represent independent thriving fields of study in their own right. We address (D) quantum criticality and competing orders, which appear to be common among many of correlated systems. In addition, we include a section on (E) correlated topological materials, which has seen significant outgrowth following the discovery of three-dimensional (3D) symmetry protected topological insulators (Hsieh *et al.*, 2008). Lastly, in contrast to these topical material classes, we also discuss the possibility of (F) returning to "legacy materials", which can exhibit similar correlated electron physics and often carry the benefit of accumulated knowledge and perhaps simpler materi-

---

[2] https://physics-astronomy.jhu.edu/the-future-of-the-correlated-electron-problem-workshop/

[3] The original lecturers for the workshop were A. Kapitulnik, A.J. Leggett, M.B. Maple, M. Norman, P. Riseborough, and G.A. Sawatzky. MBM was unfortunately unable to travel to Baltimore and so T.M. McQueen generously gave a lecture on materials aspects of the correlated electron problem. However, MBM's slides were used as a reference for the writing of this mansucript.

[4] The quote "The most exciting phrase to hear in science, the one that heralds new discoveries, is not 'Eureka!' (I found it!) but 'That's funny' " has been ascribed to Issac Asimov, but various other attributions exist.

---

[5] Perhaps even this definition is problematic. Are heavy fermion Fermi liquids with masses approximately a 1000 times (Andres *et al.*, 1975) the free electron mass strongly correlated by this definition? Perhaps not, but most would agree that they are strongly correlated. This leaves us with the only unassailable definition, which is that we know a strongly correlated system "when we see it." (Stewart, 1964)

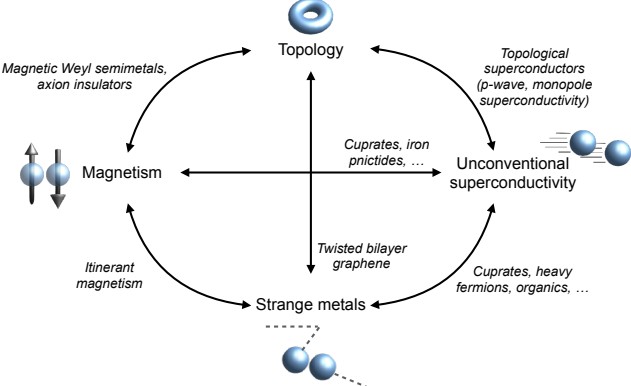

FIG. 1 Properties such as magnetism, topology, superconductivity and strange metal behavior coexist in some material's phase diagrams. For each binary link connecting these properties, we list a few examples of systems in which both phenomena can be found and which are natural candidates to study the specific relation between these properties.

als synthesis. Several of these phases or properties can be intertwined in strongly correlated materials (Fernandes *et al.*, 2019) and the link between them is mysterious. Understanding the potential causality or competition between these phenomena could help unveil some universal mechanism between different families of compounds. In Fig. 1 we show some candidate systems for the study of topology, unconventional superconductivity, magnetism and strange metal behavior. The topics discussed below are undoubtedly not exhaustive and we anticipate an outgrowth of new research areas as completely new subjects emerge and merge. Indeed, this is what makes this field so particularly dynamic and exciting.

To address the complexity of the correlated electron problem, sophisticated experimental probes, methods for material design and growth, as well as theoretical and numerical tools have been developed in recent years. We survey a number of these efforts, ranging from the development of spectroscopic and microscopic techniques to efforts to study materials under conditions of ever-increasing extremity. And we make some suggestions about what we believe is needed experimentally and theoretically to make progress here. Finally, at the risk of stating the obvious, it must be stressed that in order to make progress, time, energy, and resources must be brought to bear. Funded research, meetings, and publications should have the general treatment of correlated systems as their subject, even if some of these efforts will stray into speculation.

## II. WHAT ARE THE PROBLEMS?

### A. Correlated Superconductors

#### 1. Definition of the Problem

Many strongly correlated metals exhibit superconductivity at sufficiently low temperatures. A comprehensive understanding of these systems is necessary not only for achieving practical applications but also for clarifying many other exotic phenomena of condensed matter. Here, we focus on superconductors that cannot be well-described by the Migdal-Eliashberg theories or their extensions based on electron-phonon coupling as the pairing mechanism. So far, this list includes but is not limited to cuprates (Bednorz and Müller, 1986), iron pnictides (Kamihara *et al.*, 2008), iron chalcogenides (Hsu *et al.*, 2008; Mizuguchi *et al.*, 2008), ruthenates (Maeno *et al.*, 1994), nickelates (Li *et al.*, 2019a), heavy fermion compounds (Fisk *et al.*, 1988; Steglich *et al.*, 1979; Stewart, 1984), and organics (Jérome *et al.*, 1980), and perhaps twisted bilayer graphene (TBG) (Cao *et al.*, 2018). The discussion presented here is in no way complete, so we refer the interested reader to a number of excellent review articles (Armitage *et al.*, 2010a; Dagotto, 2005; Hosono and Kuroki, 2015; Keimer *et al.*, 2015; Mackenzie *et al.*, 2017; Norman, 2011; Si and Steglich, 2010).

Although the standard, weak-coupling BCS expression for the superconducting transition temperature places an upper limit on the superconducting critical temperature in many materials (if a moderate Coulomb retardation is assumed) (Cohen and Anderson, 1972; Moussa and Cohen, 2006), the energy scale of electron-electron interactions in correlated electron systems is generally much higher, suggesting that electronic mechanisms for superconductivity have the potential to produce high-$T_c$. Correlated superconductors generally support a richer set of superconducting ground states, including those with odd-parity and time-reversal symmetry breaking, and of course higher orbital angular momentum wave functions (*e.g.* $d$-wave).

#### 2. Possible Structure of a Solution

A particularly ambitious goal for correlated superconductivity would be to find a theory that would identify a mechanism for the formation of Cooper pairs analogous to the phonon-mediated superconductivity that applies to conventional superconductors. Such a theory might also offer insight into the connection between the exotic normal state properties and the superconducting properties of correlated superconductors. For example, the normal states of many correlated superconductors feature non-Fermi liquid transport (see Sec. II.C) and quantum critical behavior (see Sec. II.D), as well as charge and spin density waves and pseudogap states, but it remains unclear what role these properties play in the formation

of superconductivity.

It is possible, however, that it may be difficult to implicate a specific pairing interaction in these systems that would be analogous to the BCS Migdal-Eliashberg phonon mechanism. This is reminiscent of the case of superfluid $He^3$. In this system, even before the discovery of superfluidity, there were proposals for $d$-wave pairing based on van der Waals attraction between atoms (Emery and Sessler, 1960). However, based on exchange interactions and the fact that $He^3$ was believed to be almost ferromagnetic, $p$-wave spin-triplet superfluidity was also proposed (Fay and Layzer, 1968). Some years later, $He^3$ was indeed found to be a $p$-wave superfluid (Leggett, 1975; Osheroff *et al.*, 1972), but it was ultimately realized that many interactions contribute to pairing, including density, spin, and transverse current interactions. It was therefore not possible to point to a single pairing mechanism (Leggett, 1975; Norman, 2011), despite the fact that there is a tendency to focus on "spin-fluctuation exchange." "If one is interested in calculating the actual value of the effective pairing interaction quantitatively, it is by no means obvious that it is a good approximation to limit oneself to the exchange of spin fluctuations only", as Leggett wrote (Leggett, 1975) long ago. We feel there is a similar lesson here for correlated superconductors. If one cannot point to a distinct mechanism in a comparatively simple material like $He^3$, it is likely that this issue is even more challenging to resolve in solid-state systems. It may be that electron-phonon mediated superconductivity is a unique case due to disparity between electronic and phononic energy scales and the fact that the lattice and electrons comprise distinctly different sub-systems.

The fact that a specific mechanism may not be able to be implicated in some unconventional superconductors does not mean quantitative questions cannot be asked and answered. One such discussion may revolve around how energy is saved in the formation of a superconductivity (Demler and Zhang, 1998; Hirsch, 1992; Leggett, 1999, 2006a,b; Scalapino and White, 1998). Such a theory might still provide recipes for constructing superconductors with various properties, and enable control over the critical temperature or symmetry of the superconducting gap. Such an approach would clarify if room temperature superconductivity under normal conditions is a realistic possibility and where to look for exotic superconducting gap symmetries (*e.g.* odd-parity superconductors or superconductors with multi-component order parameters). In conventional BCS superconductors for instance, energy is saved through a decrease of the kinetic energy of the *ions* and the total potential energy, which outweighs the penalty from the increased electron kinetic energy (see Ref. (Chester, 1956) and note that this calculation was pre-BCS! Also see Ref. (Norman and Pépin, 2003)).

There is a significant history of related analyses for exotic superconductors, but still substantial room for progress. Scalapino and White (Scalapino and White, 1998) suggested that energy lowering in the cuprate's su-

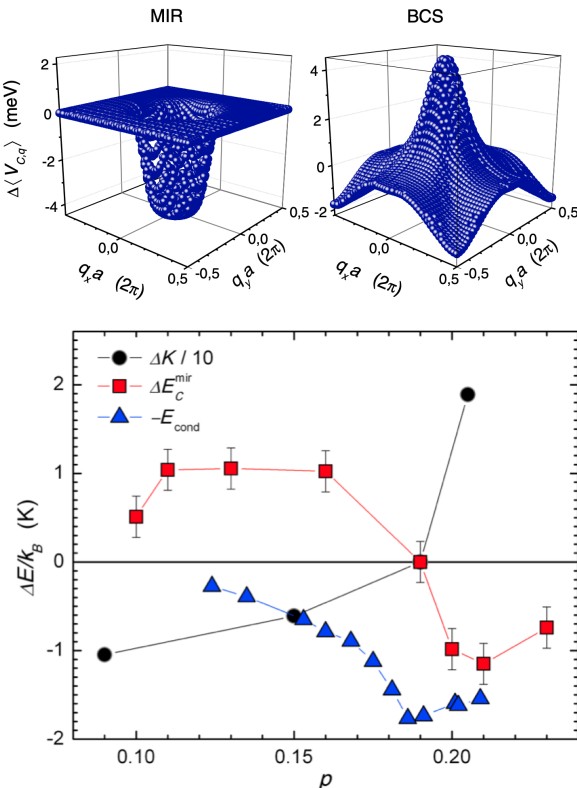

FIG. 2 (top) The difference between the normal and superconducting state Coulomb interaction energy calculated within the "midinfrared" and BCS scenarios (Levallois *et al.*, 2016a). This is one possible model for the source of energy savings in a correlated superconductor without identifying a specific pairing "glue." Here the BCS model calculation is for $d$-wave symmetry, $x = 0.16$ hole doping, and with the interaction adjusted such as to give $T_c = 100$ K. (bottom) The estimated difference between normal and superconducting state Coulomb interaction energies ($\Delta E_c^{\mathrm{mir}}$) for $Bi_2Sr_2CaCu_2O_{8-x}$ crystals, together with the total energy difference from heat capacity ($E_{\mathrm{cond}}$) at different doping levels. Also plotted is the result of a calculation that estimates the changes in kinetic energy ($\Delta K$) when entering the superconducting state. From Ref. (Levallois *et al.*, 2016a).

perconducting state could primarily come through the exchange interaction $J\langle \mathbf{S_i}\cdot\mathbf{S_j}\rangle$. In principle one could quantify magnetic energy savings by looking at differences in the neutron scattering structure factor $S(\mathbf{q},\omega)$ between normal and superconducting state. Demler and Zhang proposed a particular mechanism along these lines (Demler and Zhang, 1998). Experimentally, the exchange energy change was in excess of the condensation energy and it was proposed the difference could be due to the relative cost of kinetic energy in the superconducting state (Dai *et al.*, 1999; Demler and Zhang, 1998) as happens in BCS-style superconductors. This is not necessarily the case however. For instance, Hirsch has predicted that superconductivity is driven by lowering of electronic kinetic energy due to changes in the electronic mass in the

superconducting state (Hirsch, 1992). Under some scenarios, the kinetic energy changes can be measured in optical conductivity experiments. Here the experimental situation is inconclusive in the cuprates with possible qualitative differences in kinetic energy savings from underdoped and overdoped regimes (Carbone *et al.*, 2006; Deutscher *et al.*, 2005). It was proposed by Leggett that superconductivity in many correlated superconductors is driven by a saving in Coulomb energy that takes place predominantly at long wavelengths and mid-infrared frequencies, which results from the increased screening due to formation of Cooper pairs (Leggett, 2006b). This scenario seemingly explained several trends in many known non-BCS high-temperature superconductors, including their quasi-2D nature, their relative insensitivity to other aspects of structure, trends of $T_c$ with the number of $CuO_2$ layers per unit cell in cuprates, and a common prominent mid-infrared absorption. Studies with $q \sim 0$ measurements of the loss function (from optical ellipsometry (Levallois *et al.*, 2016a)) showed that changes to the Coulomb energy can likely account for a significant portion of the condensation energy for overdoped cuprates (Fig. 2), but the analysis had to make estimates for the finite $q$ extrapolation of the data. Therefore, further measurements of the loss function performed at finite $q$ with momentum-resolved electron energy loss spectroscopy are needed to test this theory in more detail. There is also some merit to theoretical analyses or energy savings particularly for numerical works (Fratino *et al.*, 2016; Gull and Millis, 2012; Maier *et al.*, 2004). Related analyses, which in some ways is simpler, can also be performed for the single particle spectral function, that is in principle measurable with Angle-Resolved Photoemission Spectroscopy (ARPES) (Norman *et al.*, 2000). However finite energy and momentum resolution, uncertainties with normalization, and matrix element effects have made such an approach unreliable. Irrespective of the details, the general point is that questions with quantitative answers *can* be asked even without appealing to any paradigms that we associate with BCS, Migdal-Eliashberg theory.

## 3. Central Questions

To organize our thinking, we identify five central questions on the problem of correlated superconductors.

1. **What is (are) the pairing mechanism(s)?** All known superconductors (correlated or otherwise) contain Cooper pairs. The question of the mechanism of superconductivity concerns the effective attractive interaction that is responsible for pair binding. If this interaction arises due to exchange of a soft (low-energy) bosonic excitation, we may speak of the boson as a "pairing glue", in analogy to the role of phonons in BCS, Migdal-Eliashberg superconductors. A fundamental challenge with correlated superconductivity may be that the same electrons that give rise to the pairing interaction are those that form Cooper pairs. Furthermore, as seen in the example of $He^3$, multiple types of fluctuations may contribute to the pairing interaction, such that the pairing mechanism cannot be ascribed clearly to a single process. Moreover, exchange of a pairing boson is not even necessary given that instantaneous (high-energy) interactions can also contribute to pairing in unconventional superconductors (Anderson, 2007).

2. **How/where is energy saved?** In the absence of a clear answer to the question of the pairing mechanism, it is possible nevertheless to ask reasonably model-independent questions concerning the superconducting state. In particular, what kind of energy (*i.e.* kinetic, magnetic exchange, Coulomb) is saved when the system transitions to the superconducting state (Demler and Zhang, 1998; Hirsch, 1992; Leggett, 1999, 2006a,b; Scalapino and White, 1998)? The answer to this question, which can possibly be determined experimentally through optics and momentum-resolved probes such as electron energy loss spectroscopy, inelastic X-ray scattering, inelastic neutron scattering, or ARPES, has significant implications both for developing theoretical models of correlated superconductors and for guiding the experimental search for novel superconductors.

3. **What is the order parameter symmetry?** The symmetry of the superconducting order parameter is a property that is defined independent of any microscopic mechanism. However, it can be used to constrain new theories of correlated superconductors. The most definitive order parameter tests are those that are sensitive to the superconducting order parameter's phase, such as corner SQUID and tri-crystal measurements (Tsuei and Kirtley, 2000). Unfortunately, among correlated superconductors, such studies were – until recently (Kalenyuk *et al.*, 2018) – only performed for the cuprates (Tsuei and Kirtley, 2000; Van Harlingen, 1995). It is unknown to what extent the order parameter symmetry varies between different classes of correlated superconductors. While a single order parameter symmetry seems universal for cuprates, this may not be the case in other classes of correlated superconductors (Hirschfeld *et al.*, 2011).

It is remarkable how challenging it can be to answer even this simplest of questions for correlated superconductors. The community has grappled with it recently in the case of $Sr_2RuO_4$. Largely accepted as the best example of a time-reversal symmetry breaking $p$-wave superconductor (Armitage, 2019; Maeno *et al.*, 1994), recent nuclear magnetic resonance (NMR) and strain experiments appear to have debunked this conclusion and a reevaluation

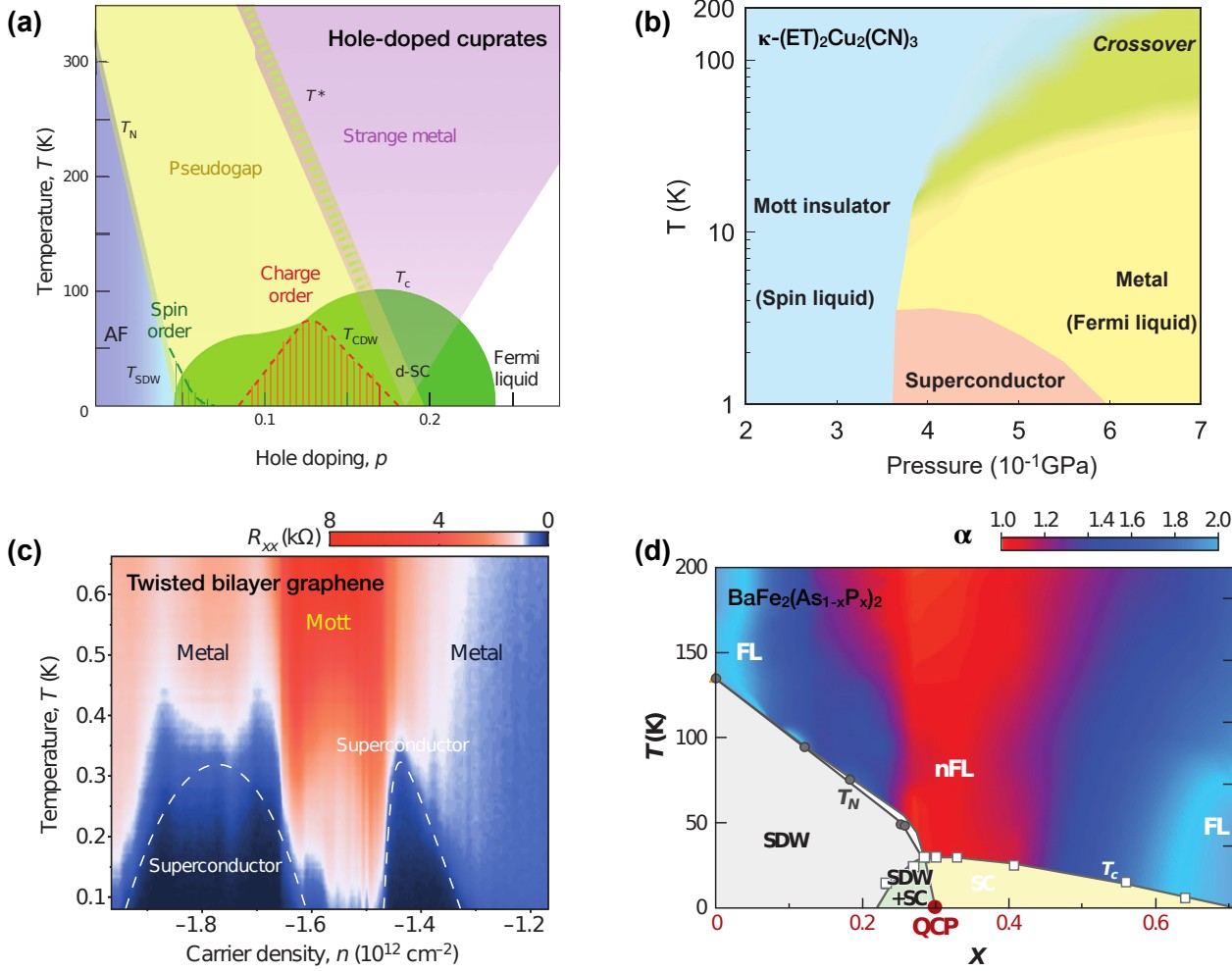

FIG. 3 Phase diagrams of several unconventional superconductors: (a) Temperature-hole doping schematic phase diagram of hole-doped cuprates (Keimer *et al.*, 2015). (b) Temperature-pressure schematic phase diagram of the organic superconductor $\kappa$-$(ET)_2Cu_2(CN)_3$ (Kurosaki *et al.*, 2005). (c) Temperature-carrier density phase diagram of twisted bilayer graphene at the magic angle $\theta = 1.16°$ (Cao *et al.*, 2018). $R_{xx}$ is the longitudinal resistance. (d) Temperature-phosphorus concentration phase diagram of the iron pnictide $BaFe_2(As_{1-x}P_x)_2$ (Shibauchi *et al.*, 2014). $\alpha$ is the exponent of the temperature dependence of the resistivity. All phase diagrams show multiple competing phases in the vicinity of the superconducting dome.

is underway (Hicks *et al.*, 2014; Pustogow *et al.*, 2019).

4. **Why do very different systems have similar phase diagrams?** Many correlated superconductors exhibit striking similarities between their phase diagrams (Fig. 3). These similarities include the existence of additional broken symmetry states, quantum critical points, proximity to magnetism, and superconducting domes (Dagotto, 2005). Proximity between superconductivity and magnetism exists, for instance, in cuprates, organic superconductors, and iron pnictides. Similarly, several systems exhibit a Mott insulating state in a parent compound at half filling (often under pressure), including possibly the recently discovered magic-

angle TBG. The common features between different phase diagrams suggest that similar phenomena might be responsible for superconductivity in different correlated systems. Although some similarities are highly suggestive, there is no consensus on where to draw the line between essential universal features of the phase diagram and system-specific details.

5. **What role does the dimensionality of the electronic structure play?** In many (but not all) correlated superconductors, the normal state electronic properties often exhibit quasi-2D behavior (with some exceptions of quasi-1D for organics and 3D for heavy fermions). Is dimensionality an important criterion for these specific unconven-

tional superconductors? If not, are there key mechanistic similarities between systems with the same electronic dimensionality?

## 4. Future Directions

With the discovery of each new family of correlated superconductors over the past four decades there has been a flurry of experimental activity. To complement and build upon these studies, we must seek new ways to explore properties of both the normal and superconducting states to begin addressing the questions outlined above.

1. Assuming that a pairing mechanism can be identified, new pump-probe style measurements may enable the direct study of the coupling between different degrees of freedom or subsystems. Such measurements include targeted pumping of particular phonon modes combined with time-resolved X-ray and photo emission spectroscopies to separate out the response of the electronic and structural degrees of freedom (Boschini *et al.*, 2018; Cilento *et al.*, 2018; Gerber *et al.*, 2017; Mankowsky *et al.*, 2014). In this way phonon mediated pairing (or lack thereof) may be identified. Similar studies include targeted pumping followed by a broadband measurement of the transient reflection or transmission at frequencies spanning from the THz to the IR region (Conte *et al.*, 2015). Other advances include time-resolved RIXS which can serve as a momentum-resolved and bulk-sensitive probe of changes in the superconducting gap in response to optical quenches and targeted pumping of particular phonon resonances (Cao *et al.*, 2019).

2. Failing the identification of a "pairing glue", certain spectroscopic techniques can directly address from where in $\omega$, $\mathbf{q}$, and $\mathbf{k}$ space the superconducting condensation energy comes (Husain *et al.*, 2019; Leggett, 1999; Levallois *et al.*, 2016a; Li *et al.*, 2018; Senga *et al.*, 2019). It should be possible to determine what region of wave vector and frequency is the Coulomb energy saved (or expended) in the superconducting transition. New generations of momentum resolved probes (see Fig. 2) should be able to give insight in this regard.

3. If electronic correlations are important for superconductivity, understanding how the Coulomb interaction is screened by different dielectric environments in layered superconductors could provide insight to the nature of the pairing (Leggett, 2006a). Revisiting seldom-studied cuprates for example (such as ones with a large number of $CuO_2$ per unit cell), like the recent work on the five-layer cuprate $Ba_2Cu_4Cu_5O_{10}(F,O)_2$ (Kunisada *et al.*, 2020), gives new perspective on a long-standing problem. Advances in sample synthesis will allow

systematic variation of the dielectric environment in correlated superconductors (Božović *et al.*, 2016; Logvenov *et al.*, 2009; Stepanov *et al.*, 2020).

4. Until recently the cuprates were the only correlated superconductors where the order parameter symmetry has been conclusively identified through phase sensitive technique (Tsuei and Kirtley, 2000; Van Harlingen, 1995). Although similar studies have been attempted for other materials, (Nelson *et al.*, 2004; Strand *et al.*, 2009), the experimental identification of the order parameter symmetry via such phase sensitive measurements is missing for most correlated superconductors. A promising development is recent Josephson junction experiments between a conventional $s$-wave Nb and the multiband iron-pnictide superconductor $Ba_{1-x}Na_xFe_2As_2$ that provide evidence for a sign-reversing $s^{\pm}$ symmetry of the order parameter in this compound (Kalenyuk *et al.*, 2018). Extending these measurements to new materials is challenging and typically requires either high quality thin films or the ability to integrate bulk crystals into devices, but is an essential direction going forward. A promising yet relatively unexplored approach to phase-sensitive experiments is conducting mesoscopic imaging experiments on polycrystalline samples (W. Hicks *et al.*, 2008). The local energy scale of the experiments allows phase-sensitive measurements to be performed on samples that are unavailable in thin-film form or cannot be interfaced with other superconductors to form Josephson junctions. Another promising method is Bogoliubov quasiparticle interference imaging that has shown, for instance, that in a number of Fe based superconductors that the gaps have opposite sign on different Fermi surface sheets (Du *et al.*, 2018; Sprau *et al.*, 2017). Similar experiments have provided supporting evidence for gap changing behavior in cuprate (Gu *et al.*, 2019). These can be applied to more classes of materials.

5. Identifying the relevant common features in the phase diagrams of correlated superconductors is an ongoing challenge. Many of these systems can be tuned by a variety of parameters, including chemical doping, external pressure, strain, and applied field. Combining two or more of these tuning knobs can further access unexplored regions of parameter space (Gerber *et al.*, 2015; Kim *et al.*, 2018a) (see Sec. III.G). For example, small uniaxial strains can have as large of an effect on electronic order as a 28 T magnetic field (Kim *et al.*, 2018a). Through broad and systematic studies across each of these variables in specific systems, it may be possible to construct multi-dimensional phase diagrams that could help identify important commonalities or surprising differences between material classes. Of course it will be only complicating if such studies

only result in the number of tuning variables increasing. In this regard it is important to establish the causal relation between the driving force and response. For example, see the work on measures of the nematic susceptibility in iron arsenide superconductors via elastoresistivity (Chu *et al.*, 2012).

Access to new and more extreme experimental conditions will also be useful in exploring phase space. For instance, the recently discovered re-entrant superconducting phases in UTe$_2$ under high field and high pressure (Aoki *et al.*, 2019; Ran *et al.*, 2019a,b) call for a search of potential novel phases in other materials to reveal new common features between correlated superconductors. As shown in Fig. 4, this material exhibits by far the highest upper and lower critical fields of any field-induced superconducting phase, more than 40 T and 65 T respectively.

6. It is also important to understand how distinct phases interact and compete at mesoscopic and microscopic scales. Local probes that can identify and measure specific regions or states in isolation will thus provide important insight. Spatially-resolved imaging of magnetism using diamond nitrogen vacancy centers and nano-SQUID could reveal short-range magnetic correlations (Ceccarelli *et al.*, 2019; Martínez-Pérez and Koelle, 2017; Pham *et al.*, 2011; Vasyukov *et al.*, 2013; Wolf *et al.*, 2015). This is potentially crucial information for the study of the pseudogap phase of cuprates for example, whose origin has been lengthily debated and could be due to antiferromagnetic spin fluctuations. Localized spectroscopic measurements can distinguish between contributions from mesoscopic inhomogeneities or disorder and other competing phases that would otherwise be averaged by more macroscopic measurements.

7. Advances in focused ion beam (FIB) and other nano-fabrication techniques can be used to realize highly-controlled device geometries for investigating dimensional effects and transport anisotropies (Ronning *et al.*, 2017). One potential pitfall given the quasi-2D nature of many of these superconductors would be the neglect of the *c*-axis properties. As demonstrated by the inter-layer tunneling model for cuprate superconductivity, however, understanding the out-of-plane electronic properties should not be neglected (Leggett, 1996; Tsvetkov *et al.*, 1998; Wheatley *et al.*, 1988).

8. It is useful to identify model systems that can be studied exhaustively and tuned extensively. It seems unlikely to find a single model compound for unconventional superconductivity that is well-suited for all probes and that can be obtained in good quality over the whole phase diagram. Nevertheless, recent discoveries of unconventional super-

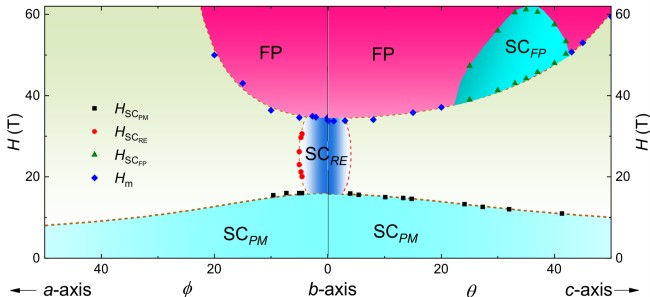

FIG. 4 Magnetic field - angle phase diagram of UTe$_2$ showing three superconducting phases with, among other aspects, the largest upper critical field in a reentrant superconductor. The magnetic field is rotated within the *ab* and *bc*-plane. Sample temperature is 0.35–0.5 K. From Ref. (Ran *et al.*, 2019b). FP is field polarized, PM is paramagnetic, RE is reentrant, and SC is superconductor.

conductivity in van der Waals materials and heterostructures such as TBG (Cao *et al.*, 2018) and one unit cell thick cuprate Bi2212 (Jiang *et al.*, 2014; Sterpetti *et al.*, 2017; Yu *et al.*, 2019) could provide simple enough toy models in which various parameters can be more easily tuned. Growth-integrated techniques for layer-by-layer probes (*e.g.* *in situ* STM (Lv *et al.*, 2015) or MBE growth of isolated "deconstructed" layers (Logvenov *et al.*, 2009)) could help explore the role of single atomic planes or interfaces between different materials.

### B. Quantum Spin liquids

#### 1. Challenges in the field

The essential problem of quantum spin liquids (QSLs) (Balents, 2010; Broholm *et al.*, 2020; Knolle and Moessner, 2019; Savary and Balents, 2016) is that despite a multitude of candidate materials, none have been proven as QSLs. Progress toward positively identifying a QSL faces two roadblocks. The first is the imprecise definition often used for a "quantum spin liquid". Spin systems with no long-range order at $T = 0$ have often been conflated with spin liquids—but this is not sufficient; other effects, such as disorder (Zhu *et al.*, 2017a) or weak interactions (Calder *et al.*, 2010), can also prevent magnetic order. Materials with no long-range magnetic order in the limit of zero temperature are more properly called quantum paramagnets. Moreover, recent theory suggests that QSLs and long-range ordered states are not necessarily mutually exclusive (Brooks-Bartlett *et al.*, 2014; Liu *et al.*, 2019). A useful definition for a QSL should not focus on the absence of conventional features, but on the presence of key properties of interest *e.g.* long-range entanglement (Kitaev and Preskill, 2006) and fractionalized excitations (Broholm *et al.*, 2020). Although there are exactly solvable models that show that

such a state in principle can exist (Takagi *et al.*, 2019), it remains to be shown if such a state exists in real materials, which are subject to disorder, further neighbour and "ring-link" exchange, and spin-lattice coupling.

This leads to the second roadblock; there are no clear experimental signatures of the long-range entanglement and fractionalized excitations which demonstrate the existence of a QSL ground state. The presence of a diffuse continuum of scattering in neutron spectroscopy experiments is often interpreted as a hallmark for fractionalized excitations but this too is insufficient. Disorder-induced glassy behavior can also produce a diffuse neutron spectrum (Zhang *et al.*, 2019d; Zhu *et al.*, 2017a). Furthermore, while the experimental neutron continuum scattering of a 1D spin chain can be accurately related to theoretical models (Mourigal *et al.*, 2013), such straightforward comparisons are not possible in 2D and 3D materials (Knolle and Moessner, 2019; Zhu *et al.*, 2017a).

## 2. Routes to Progress

To make substantial progress in this field, it is necessary to develop a means of directly measuring fractionalized excitations and long-range entanglement. There are many proposed options, most of which require both new theoretical calculations and new experimental techniques. Here we consider some possibilities:

1. **New analysis of neutron spectroscopy.** Neutron scattering measures the energy-resolved spin-spin correlations, and this can encode evidence of fractionalization and entanglement (Hauke *et al.*, 2016; Morampudi *et al.*, 2017). Rather than focusing on 2D plots of diffuse continua, more sophisticated efforts may extract entanglement bounds (Hauke *et al.*, 2016; Laurell *et al.*, 2021; Scheie *et al.*, 2021) or fractionalization signatures in the energy-dependence of response functions (Morampudi *et al.*, 2017).

2. **Thermal transport.** Thermal transport is potentially extremely powerful in probing spin degrees of freedom in electrical insulators. It can be directly sensitive to spin transport or can be sensitive to phonons scattering off of spins. The technique has a long history but is still underutilized in quantum magnets. A linear term as a function of temperature in thermal conductivity is an expected hallmark of a QSL featuring a spinon Fermi surface (Xu *et al.*, 2016), and different spin liquids are expected to have distinct, topological thermal Hall conductivity signals (Kasahara *et al.*, 2018; Katsura *et al.*, 2010; Zhang *et al.*, 2020). A quantized thermal Hall effect is an expected signature of Majorana fermions in a Kitaev spin liquid (Kasahara *et al.*, 2018; Katsura *et al.*, 2010; Zhang *et al.*, 2020) and has been reported in $\alpha-$RuCl$_3$ as shown in Fig. 5.

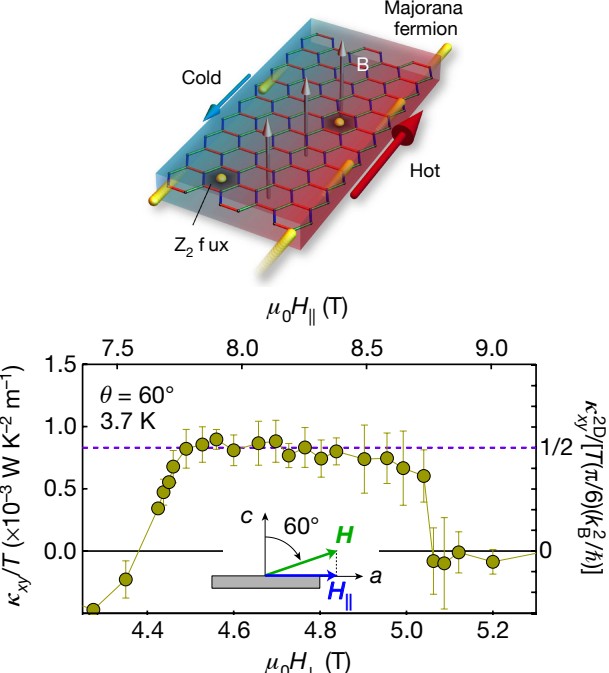

FIG. 5 (top) Schematic illustration of the thermal Hall conductivity of a Kitaev spin liquid, with a magnetic field perpendicular to the sample plane, resulting in the fractionalization of spins into Majorana fermions (yellow spheres) and $Z_2$ fluxes (hexagons). The charge-neutral Majorana fermions are responsible for the conduction of heat by chiral edge currents. (bottom) Half integer plateau reported in the thermal Hall conductance of $\alpha$-RuCl$_3$. Adapted from Ref. (Kasahara *et al.*, 2018).

3. **Imaging spin densities around impurities.** It has been theoretically proposed that nonmagnetic impurities will produce a characteristic spin density pattern in a Kitaev spin liquid (Willans *et al.*, 2011). Similar local spin measurements were key to proving Haldane behavior (*e.g.* fractionalized end spins) for $S = 1$ chains (Hagiwara *et al.*, 1990). Atomic-resolution local spin density measurements may be available in the near future and could be broadly applied to all classes of QSLs (see Sec. III.E).

4. **Entangled neutron beams.** Recent experiments have shown that an entangled neutron beam can, in principle, probe quantum entanglement between different points on a lattice (Shen *et al.*, 2020b). This technique may provide a direct measure of entanglement in solid-state systems.

5. **Device fabrication.** The fractionalized quasiparticles of a QSL are predicted to give unique signals if incorporated in microscopic spintronic devices (Aasen *et al.*, 2020; Barkeshli *et al.*, 2014; Chatterjee *et al.*, 2019; Chatterjee and Sachdev, 2015). QSLs sandwiched between metals, super-

conductors, or ordered magnets, could have their fractionalized excitations directly probed through certain measurements, such as measurements of spin current. Such direct probes would be possible despite the electrically insulating nature of most QSLs.

6. **Spin noise experiments.** Spin noise measurements are underutilized in quantum magnets. Fractional quasiparticle creation and annihilation are likely to give magnetic noise signal distinct from conventional Boltzmann fluctuations. Such experiments have been done for the classical spin ices (Dusad *et al.*, 2019; Watson *et al.*, 2019), and may yield definitive evidence of quasiparticles in QSL candidates. As discussed below in Sec. III.D.2, there have also been proposals for how to measure entanglement in solid-state systems (Laflorencie, 2016) with a globally conserved quantity such uniform magnets. For instance, Song et al. (Song *et al.*, 2012) propose that the noise spectrum can be a probe of entanglement in a $O(2)$ quantum magnet that has a magnetic field partially obscured by a superconducting shield. Related theory (and experiment) could be extended to QSL candidates.

7. **Out-of-equilibrium relaxation.** If a system is perturbed out of its ground state, relaxation back to equilibrium will generally be different for QSL and non-QSL states (see for example Ref. (Claassen *et al.*, 2017a)). For instance, Nasu *et al.* showed that when quenching a Kitaev spin liquid with a magnetic field, the relaxational dynamics are qualitatively different depending on the phase that it is quenched from (Nasu and Motome, 2019). In the case of the quench from the high-field classical ferromagnetic (FM) phase, the two Majorana fermions are strongly coupled and the spin dynamics are conventional, originating from the precessional motion of spins. In a quench from the low-field spin-liquid phase, which is connected to the zero-field Kitaev spin liquid, two Majoranas are observed separately in the time evolutions. Measuring the relaxation after a quench may allow spin-liquid states to be distinguished from other non-fractionalized states.

8. **Multidimensional coherent THz spectroscopy.** This newly developed nonlinear optical technique may be able to resolve the difference between a diffuse continuum from local disorder and a diffuse continuum from fractionalized quasiparticles (Choi *et al.*, 2020; Lu *et al.*, 2017; Wan and Armitage, 2019) and measure the lifetime of multi-spinon excited states (see Sec. III.D). In some cases it allows new information about excitations that can already be seen in linear response (Wan and Armitage, 2019). In other cases it provides spectroscopic information that is inaccessible at the level of nonlinear response (Lu *et al.*, 2017).

As an overall comment, the effort to distinguish a QSL from glassy or other quantum-disordered phases requires paying close attention to sample quality. Disorder or defects may cause "false positives" with regards to creating broad continuum lineshapes even in spin systems that would have well-defined low temperature spin-wave-like excitations (Paddison *et al.*, 2017; Zhang *et al.*, 2018b; Zhu *et al.*, 2017a). Although some theoretical proposals suggest that certain types of disorder can stabilize a QSL (Kawamura and Uematsu, 2019; Savary and Balents, 2017; Wu *et al.*, 2019), lessons learned from superfluid $He^3$ – perhaps the cleanest condensed matter physics system in its pure form – indicate that random disorder tends to destroy quantum coherence. Rather, disorder must be highly correlated and structured in order to induce quantum effects. When superfluid $He^3$ is confined in high porosity ($\sim 98\%$ porous) silica aerogels to act as artificial defect scattering centers, the real-space correlations of the gel on length scales comparable to the superfluid correlation length become vitally important. When the disorder is carefully controlled and thoroughly characterized, a variety of novel superfluid phases can be induced (Li *et al.*, 2013; Pollanen *et al.*, 2012; Thuneberg *et al.*, 1998), including a correlated topological phase (Dmitriev *et al.*, 2015; Zhelev *et al.*, 2016). The most important aspect in stabilizing these phases is the presence of particular types of correlated disorder. Understanding and interpreting these phases and their origins requires a detailed understanding of the underlying disorder itself.

These examples demonstrate that greater caution in claiming experimental evidence for QSL is needed. Finding a QSL state is an exciting possibility in strongly correlated electron physics, but we have to be clear what exactly we are looking for and how to find it.

## C. Strange Metals

### 1. Definition of Phenomenology

The problems presented by "bad" and "strange" metals are simple to pose, but have withstood an understanding for many decades now. Early work (see Fig. 6) showed that the normal phase of high-temperature cuprate superconductors exhibits "bad-metal" transport signatures – most notably the violation of the Mott-Ioffe-Regel (MIR) limit $\rho \lesssim \hbar/e^2$ (in 2D) at high temperatures. The MIR limit corresponds to the notion that metallic transport of electron-like quasiparticles cannot occur with a mean free path much shorter than some microscopic scale of the system (*e.g.* inverse Fermi wavelength, lattice constant) (Gunnarsson *et al.*, 2003). Results like these have considerably challenged the Boltzmann or quasiparticle picture of transport in such systems (Greene *et al.*, 2020;

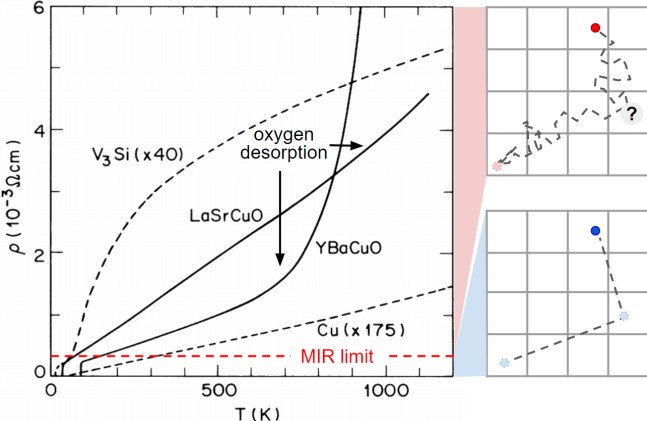

FIG. 6 Electrical resistivity of normal vs. strange metals. Solid lines are in-plane resistivity of two archetypal cuprate high-temperature superconductors - $La_{2-x}Sr_xCuO_4$ and $YBa_2Cu_3O_{7-\delta}$ near their respective optimal hole-doping. The upturn at high temperature in $YBa_2Cu_3O_{7-\delta}$ has been ascribed to oxygen loss. Dashed lines are resistivity curves from normal metals, which either show saturating behavior ($V_3Si$), or maintain a phonon-dominated linear resistivity well below the MIR limit (Cu). Adapted from Ref. (Gurvitch and Fiory, 1987). Cartoons on the right represent electron motion with mean free path shorter (top) and longer (bottom) than the primitive unit cell size (gray grids).

Gurvitch and Fiory, 1987; Hussey *et al.*, 2004). We typically refer to metals as "bad" even if the resistivity does not violate the MIR limit, but appears smoothly connected to the regime where it does. For instance, 19% doped $La_{2-x}Sr_xCuO_4$ right above $T_c$ has a resistivity that is low enough to not violate MIR, but is smoothly connected to the high temperature regime where MIR is violated (Emery and Kivelson, 1995b). Furthermore, many of these materials show a "strange metal" behavior where the resistivity is linear in temperature and smoothly passes through the MIR value (Emery and Kivelson, 1995a). The persistence of a linear resistivity at the lowest temperatures when superconductivity is quenched (Cooper *et al.*, 2009; Daou *et al.*, 2009; Doiron-Leyraud *et al.*, 2009; Jin *et al.*, 2011; Legros *et al.*, 2019) is perhaps one of the most difficult aspects to explain theoretically, due to the lack of a scattering mechanism that gives $\tau \sim 1/T$ at temperatures lower than the Fermi energy, Debye temperature, and other relevant energy scales. These behaviors contrast with metals whose resistivity saturates near the MIR limit (Gunnarsson *et al.*, 2003; Hussey *et al.*, 2011). We note that similar behavior has been recently observed in a cold atom system (Brown *et al.*, 2019). When tuning with magnetic field, possibly related behavior has been found in the cuprates where a linear magnetoresistance is found in magnetic fields up to 80 T, the magnitude of which mirrors the magnitude and doping evolution of the linear in temperature resistivity (Giraldo-Gallo *et al.*, 2018). Related observations have been made in the pnictides (Hayes *et al.*, 2016).

Scaling of observables with temperature with unconventional exponents frequently hints at the possible proximity to a quantum critical point (QCP), see also Sec. II.D. Quantum criticality is a recognized framework that features non-quasiparticle transport; however a linear in temperature resistivity evades simple scaling arguments. Indeed, if charge density scales as dimensionality $d$, the Kubo formula for the conductivity leads to $\rho \sim T^{(2-d)/z}$ (where $d$ is the dimension and $z$ is the dynamic critical exponent), at low temperatures (Damle and Sachdev, 1997; Phillips and Chamon, 2005) (see Ref. (Hartnoll and Karch, 2015) for extensions to scaling theory that accommodate some of the phenomenology of strange metals). Furthermore, although it is tempting to interpret the simple scaling of resistivity with field observed in some strange metals (Hayes *et al.*, 2016; Sarkar *et al.*, 2019) as additional signatures of a QCP, recent work has suggested it could be explained classically from macroscopic disorder (Boyd and Phillips, 2019; Patel *et al.*, 2018).

While strange metals exhibit their most salient features in transport, their connection to spectral signatures remain less clear. Angle-resolved photoemission spectroscopy (ARPES) experiments in optimally hole-doped cuprates show persistent incoherent spectra near the antinode momenta in the Brillouin zone, with a scaling of widths that is consistent with $\omega/T$. Such behavior persists even above the pseudogap temperature, indicating its distinction from the pseudogap. A sudden spectroscopic collapse of the strange metal occurs above a critical doping, which also inflicts sudden changes in lower temperature properties such as the pseudogap, the superconductivity, and electron-boson coupling strength (Chen *et al.*, 2019b; Hashimoto *et al.*, 2015; He *et al.*, 2018b).

An important point is that the transport and single-particle manifestations of strange metals are not obviously equivalent. A good example is the hole-doped cuprate superconductor $La_{1.6-x}Nd_{0.4}Sr_xCuO_4$ which shows perfect $T$-linear resistivity down to $T = 0$ at a doping right above the pseudogap critical doping $p^*$ (Daou *et al.*, 2009). Nonetheless, angle-resolved photoemission (ARPES) finds a well defined electron-like Fermi surface (Matt *et al.*, 2015) and the Wiedemann-Franz law is satisfied (Michon *et al.*, 2018). At the same time, ARPES experiments on overdoped Bi2212 do seem to show a correspondence between the temperatures where sharper features appear in the spectra and an inflection point appears in the resistivity. Therefore, it is not simple to identify which materials should be treated entirely outside of the Fermi liquid framework.

### 2. Outstanding Questions and Perspectives

In correlated systems in general, and strange metals in particular, there is a general lack of connection between single-particle properties (like ARPES) and multi-particle properties (like electrical transport). When in-

teractions are weak, *i.e.* non-linearities in the quantum description of the system can be treated perturbatively, Wick's theorem can be used to reduce multi-body correlation functions into products of single-particle Green's functions, through the use of Feynman diagrams. Development of new multi-particle probes may elucidate the breakdown of Wick's theorem by quantifying non-linearities, or reveal details about interaction effects more generally. Multi-particle photoemission (Haak *et al.*, 1978; Stahl and Eckstein, 2019; Su and Zhang, 2020) where the energy and momentum of multiple emitted electrons are measured simultaneously may be very illuminating in this regard (see Sec. III.D). The principal issue with such experiments thus far is their relatively poor resolution, mainly limiting the method to getting information on the correlation hole around an electron (Schumann *et al.*, 2007). If such problems could be overcome, the two-electron removal function would provide direct information, in the cuprates for instance, concerning the development of the superconducting state from the strange metal normal state. We also believe nonlinear response (Sun *et al.*, 2018) and direct probes of the dynamic charge susceptibility (Mitrano *et al.*, 2018)) are also promising novel routes to probe the strange metal.

Experimentally, coordinated measurements of the carrier density, resistivity, and other physical parameters (magnetic susceptibility, specific heat, thermal diffusivity, single-particle spectral function) at high temperature near the MIR limit will help elucidate the nature of the dissipation mechanism(s). For example, direct measures of phonon self-energy with inelastic X-ray (He *et al.*, 2018c) and neutron scattering (Merritt *et al.*, 2019) can provide a momentum-resolved view of possible lattice dissipation channels. An important issue to manage is the oxygen concentration in cuprates. Because some of the strange metal phenomenology occurs at high temperatures where these systems can be susceptible to oxygen migration, it would be a significant advance for experiments to be able to monitor and control carrier concentration simultaneously. Under a better-defined theoretical framework in the time-domain, ultrafast pump-probe techniques may also help reveal various dissipation mechanisms and even discover new forms of excitations in strange metals.

In addition to the pursuit of a consistent description of the cuprates' strange metal phenomenology seen by different probes, efforts to generalize the description and seek universality in other material systems may provide new perspectives (Cao *et al.*, 2016; Stewart, 2001). For example, iron-based superconductors exhibit transport phase diagrams that strongly resemble the cuprate strange metal region both with and without magnetic field (Hayes *et al.*, 2016; Kasahara *et al.*, 2010). Ruddlesden-Popper series materials $Sr_{n+1}Ru_nO_{3n+1}$ offers a platform to study Fermi-liquid-to-strange-metal crossover at a moderate temperature with highly controllable structural motifs (Mousatov *et al.*, 2020). The recent observation of linear resistivity in magic angle TBG

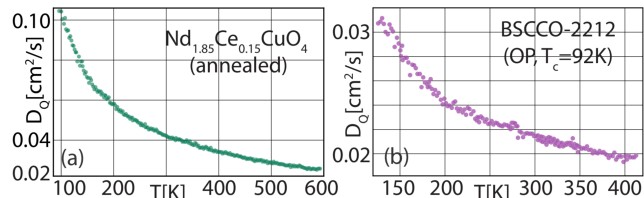

FIG. 7 Thermal diffusivity in the *ab* plane as a function of temperature, measured using the optical setup discussed in Ref. (Zhang *et al.*, 2019c, 2017a) for $Nd_{1.85}Ce_{0.15}CuO_4$ (a) and optimally doped $Bi_2Sr_2CaCu_2O_{8+x}$ (b). The data is consistent with a quantum-limited relaxation time that goes as $\hbar/k_BT$.

may bring in exceptional tunability and control to the study of electron scattering mechanisms and its relation to superconductivity in this system (Cao *et al.*, 2020b; Polshyn *et al.*, 2019).

Another remarkable feature of strange metals is that they appear to exhibit a transport time scale ($\tau_{\rm tr}$) that is close to saturating conjectured bounds (Hartnoll, 2015; Sachdev, 2007) $\tau_{\rm tr} \geq \alpha\hbar/k_BT$, with $\alpha$ a number of order unity. While experimental reports of the Planckian limit, including the recently discovered magic-angle TBG, point towards $\alpha \approx 1$ (Bruin *et al.*, 2013; Cao *et al.*, 2020b; Legros *et al.*, 2019), it is important to note that no rigorous bound has been established theoretically to date. A challenge for the coming years is to sharply define $\tau_{\rm tr}$ and derive such a bound, if it exists. In the meantime, it is crucial that experiments attempt to extract transport time scales with as little bias as possible to constrain $\alpha$. One other puzzle is the seemingly similar values of $\alpha$ for a wide range of temperatures – with resistivities both below and above the MIR value – despite changes in single-particle observables. This aspect is reproduced in certain models (Cha *et al.*, 2020; Mousatov *et al.*, 2019), but a deeper understanding remains absent.

A promising avenue for further experimental study of $\tau_{\rm tr}$ has been opened up by recent measurements of the thermal diffusivity (Fig. 7) by a novel optical technique wherein a local temperature gradient is established with one laser, and the diffusivity is measured by the reflectivity of a second laser (Zhang *et al.*, 2019c). It allows the thermal diffusivity to be obtained directly, without the need to measure the thermal conductivity and specific heat separately. The diffusivity is connected to the more conventional transport coefficients via the Einstein relations; neglecting thermoelectric effects $\sigma = \chi D_e$ and $\kappa = cD_Q$ (see (Hartnoll, 2015) for a more general relation). One immediate and striking result of experiments on cuprates has been the observation that, like charge transport, thermal transport also shows a linear dependence on temperature (Zhang *et al.*, 2019c). Data at high temperatures is consistent with a thermal transport scenario where entropy is carried by an overdamped diffusive fluid of coupled electrons and phonons characterized by a unique velocity and a quantum-limited relaxation time

$\hbar/k_BT$ (Zhang *et al.*, 2017a). The approach to this linear regime is qualitatively consistent with a bound on $\tau_{\mathrm{tr}}$, a result that is especially intriguing given that many of the measured cuprates do not show $T$-linear resistivity in the same temperature range. This apparent wrinkle on the idea of a transport bound highlights the importance of applying novel measurement techniques in the strange metal regime.

As mentioned above, issues related to strange metals are intimately related to those associated with perennial observations of non-Fermi liquid physics. In some systems, the route to non-Fermi liquid (NFL) behavior seems to involve single ion physics (Maple *et al.*, 2010). In fact, the first $f$-electron system $Y_{1-x}U_xPd_3$ in which NFL behavior was observed exhibits an unconventional Kondo effect in which the Kondo temperature decreases with U concentration due to Fermi level tuning before the system undergoes spin glass ordering at $x \approx 0.2$ followed by long-range antiferromagnetic ordering at $x \approx 0.4$ (Maple *et al.*, 1995, 1994, 2010; Seaman *et al.*, 1991). The low-T NFL characteristics scale with the Kondo temperature and the U concentration and have anomalous dependences[6]. Similar weak power law and logarithmic T-dependences of the NFL characteristics have been found in many $f$-electron systems (Löhneysen *et al.*, 2007a; Maple *et al.*, 1995, 1994, 2010; Stewart, 2001). Several of these characteristics were consistent with a quadrupolar Kondo effect, a variant of a 2-channel spin-$1/2$ Kondo effect, which could occur if the U ions were tetravalent and the $f$-electron ground state in the cubic crystal field was a $\Gamma_3$ nonmagnetic doublet which carries a quadrupolar moment (Cox, 1987; Cox and Jarrell, 1996). The quadrupolar Kondo model was first proposed by Cox (Cox, 1987) to account for the low temperature properties of the heavy fermion compound $UBe_{13}$. The observation of NFL characteristics in the $YUPd_3$ system came as a surprise and opened a new era of research on NFL behavior in $f$-electron materials. These observations reinforce the notion that much of NFL behavior and strange metal physics in correlated electron quantum materials arises from the tension between localized and itinerant electron character. In such materials, the hybridization of localized and itinerant electron states is manifested in complex temperature *vs.* composition, pressure and magnetic field phase diagrams consisting of regions in which various correlated electron phenomena and phases reside and, in addition, new phenomena and phases emerge, often in the vicinity of a QCP. Whether this dichotomy between local and itinerant physics is causing strange metal behavior in general – is as of yet – unclear.

---

[6] The following dependencies are seen in $Y_{1-x}U_xPd_3$. Electrical resistivity varies as $-T$, the specific heat divided by temperature $C(T)/T$ diverges as $-\log(T)$ with evidence of a residual entropy $S(0) = (1/2)R\log(2)$, and the magnetic susceptibility varies as $T^{1/2}$.

Both microscopic and phenomenological approaches will be important in tackling the strange metallic problem theoretically. The central challenge in the phenomenological or effective theory approach is an identification of the appropriate collective excitations and their dynamics, potentially realizing marginal Fermi liquid phenomenology (Varma *et al.*, 1989). For example, hydrodynamics assumes that only excitations related to exact or approximate conservation laws survive after fast local thermalization, and thereby directly connects response functions to exact or approximate symmetry assumptions (Davison *et al.*, 2014; Delacrétaz *et al.*, 2017; Lucas and Hartnoll, 2017a). Constructions based on the Sachdev-Ye Kitaev model have provided novel microscopic approaches to strange metallicity (Chowdhury *et al.*, 2018a; Patel *et al.*, 2018; Song *et al.*, 2017). A challenge of any microscopic theory is the emergence of an energy scale $\frac{h}{e^2}T\sigma_{\mathrm{dc}}$ that is far below the Fermi energy (*e.g.* the size of small Fermi pockets (Mousatov *et al.*, 2020)) and a mechanism for $T$-linear transport time down to low temperatures (see *e.g.* Ref. (Patel and Sachdev, 2019) for a recent example in the context of the SYK model). And ultimately, any microscopic theory should give insight into why some metals are strange and some are not.

### D. Quantum Criticality

#### 1. Status of the field

Quantum criticality is found in many correlated electron materials and proximity to a quantum critical point has been proposed to be an organizing principle for many strongly interacting systems. The point of instability between two competing ground states, which each have their own quasiparticle spectrum can have an excitation spectrum that is non-quasiparticle like. Although quantum criticality has been a focus of the field for many years there are still many open issues theoretically and experimentally. Conventionally, it has been well-described within the Landau-Ginzburg-Wilson (LGW) paradigm (Ma, 2018; Sachdev, 2007; Sondhi *et al.*, 1997), where the renormalization group method has been extended to capture dynamics of order parameters at a continuous transition into a broken symmetry state. However, the LGW framework fails to address the case of a continuous transition between two distinctly ordered phases (Senthil *et al.*, 2004a) and provides no description of phases and phase transitions that are not characterized by a symmetry breaking (with the possible exception of the 2D metal-insulator (Abrahams *et al.*, 2001) transition in disordered metals where the critical modes may be identified). And although proximate quantum criticality is frequently invoked to explain anomalous scaling of exponents in strongly interacting metals, there is still no accepted framework for quantum criticality in magnetic heavy fermion metals. Additionally, the role of disorder in many systems is still unclear. As proximity to

a QCP is frequently used as a "go-to" explanation for anomalous system properties, the community would also benefit from having more simple materials to investigate paradigms of quantum criticality. One could then understand the range of parameter space that is affected by a proximate QCP and get better intuition about prospective signatures of quantum criticality in other systems. For detailed reviews of the conventional understanding of quantum criticality, we refer readers to Refs. (Carr, 2010; Gegenwart *et al.*, 2008; Ma, 2018; Sachdev, 2008; Si and Paschen, 2013; Sondhi *et al.*, 1997).

The limitations of LGW have been demonstrated in model systems and material examples. A prominent example involves itinerant metallic systems where the conducting electrons couple to the critical fluctuations of the order parameter. The gapless nature of particle-hole excitations in this case may dramatically alter the nature of the QCP. For example, such a coupling directly renders the transition first-order in ferromagnetic metals (Belitz *et al.*, 1999) and magnetic Dirac semimetals (Belitz and Kirkpatrick, 2019). On the other hand, quantum phase transitions associated with antiferromagnetic order or nematic order can be continuous (Sondhi *et al.*, 1997). The critical theory for metallic systems with antiferromagnetic or nematic ordering is relatively well understood in three dimensions (Sachdev, 1999). In contrast, in two dimensions the critical coupling is strongly relevant and a full understanding of the critical behaviour is still elusive (Sachdev, 2008). There are attempts to perform controlled calculations introducing additional control parameters, such as the number of electronic species or flavors, which predicts non-Fermi-liquid electronic behaviour (Lee, 2018). The self-energy of electrons is shown to scale as $\Sigma(\omega) \sim \omega^{2/3}$ for the nematic transition and $\sim \sqrt{\omega}$ for the antiferromagnetic one, leading to the breakdown of the quasi-particle picture. An important frontier here is going beyond these approximations both analytically and numerically. While some progress has been achieved (Berg *et al.*, 2019; Lee, 2018), it is so far restricted to a few special cases. Developing more general frameworks is thus the work for the future.

Experimentally, strange metal behavior, such as linear-in-temperature resistivity has been observed close to QCPs in several systems including heavy fermions, cuprates and iron-based superconductors (Legros *et al.*, 2019; Löhneysen *et al.*, 1994; Shibauchi *et al.*, 2014). Although such behavior seems to be associated with a 'phase' (see Sec. II.C above), one reoccurring question is whether such non-Fermi-liquid behaviours can be explained by the framework with a Fermi surface coupled to a critical mode as described above (Schofield, 1999). In particular, for heavy-fermion systems there is frequently a clear critical point between an antiferromagnetic metal with small Fermi surfaces and the heavy Fermi liquid with large Fermi surfaces. Thus naively one may think the critical point is just the onset of the antiferromagnetic order (Si and Steglich, 2010), but the standard Hertz-Millis theory of an antiferromagnetic critical point (Sachdev, 1999) fails to explain the NFL behaviour observed in experiment. Meanwhile, a jump of Fermi surface volume has also been observed, for example in a pressure-tuned critical point in $CeRhIn_5$ through quantum oscillation measurements (Shishido *et al.*, 2005). This again cannot be explained simply by the onset of magnetic ordering.

Although as mentioned antiferromagnetic transitions in 3D are frequently mean-field-like, exotic scaling has been predicted at the QCP in nodal metals where antiferromagnetic fluctuations and electrons at a nodal point are strongly coupled. In such cases, the coupling between the electronic and the critical modes can qualitatively change the critical behaviours compared to the pure ordering transition without itinerant electrons. This type of QCP, proposed for the pyrochlore iridates, is beyond the current experimental scope, although there may be indirect evidence from symmetry breaking in a quantum phase transition in $Cd_2Os_2O_7$ (Savary *et al.*, 2014; Wang *et al.*, 2018b).

Heavy fermion metals are replete with systems that seem to evade the simplest considerations for criticality. For instance, in the case of the quantum critical point of $CeCu_{6-x}Au_x$, which goes from a paramagnetic metal to an antiferromagnetic metal as $x$ increases through a critical value, $x_c \approx 0.1$, inelastic neutron scattering has shown that critical scattering with $\omega/T$ scaling occurs all over the Brilloun zone (instead of just at antiferromagnetic wavevectors) (Schröder *et al.*, 1998). This is in contrast to the usual notion that when a metal undergoes an antiferromagnetic quantum phase transition, fluctuations induced by quantum criticality are taken to be long-wavelength fluctuations of the order parameter at the ordering wave vector.

New theoretical frameworks may therefore be necessary to understand such physics. One theoretical scenario is that the heavy fermion critical point is associated with "Kondo breakdown" instead of the onset of antiferromagnetic order (Coleman and Nevidomskyy, 2010; Senthil *et al.*, 2004b; Si, 2010). The key idea is that one electron per site gets "Mott" localized to form a local spin moment and only the remaining electrons can move coherently, resulting in a sudden drop of carrier density. So far there has been only moderate progress in theories where this Kondo breakdown and the onset of antiferromagnetism can happen simultaneously (Khait *et al.*, 2018; Komijani and Coleman, 2019). A separate mechanism has been postulated in the form of spin-charge separation (Coleman *et al.*, 2001). An agreed upon treatment for the realistic 2D or 3D systems is currently lacking. The drop of carrier density from $1 + p$ to $p$ ($p$ being the hole doping) has also been observed in hole-doped cuprates below the pseudogap critical point $p^*$ under high magnetic field (Proust and Taillefer, 2019). In this case, no long-range antiferromagnetic order has been observed below the critical point, which suggests a different mechanism of reconstructing the Fermi surface without involving symmetry breaking order parameters.

In the framework of "local" quantum criticality (Si *et al.*, 2001), the Kondo effect is destroyed because local moments are coupled not just to conduction electrons but also to the fluctuations of the other moments. The destruction of the Kondo effect leads to the vanishing of quasiparticle weight (and hence a divergent effective mass) on the entire Fermi surface. In contrast, in an antiferromagnetic QCP, the quasiparticle spectral weight vanishes only near the "hot spots" (*e.g.* the portions of the Fermi surface that are connected by the antiferromagnetic wave vector). This does not lead to anomalous transport as the current carried by "cold" electrons short circuits the ones in the hot spot.

One may expect related physics underlies the heavy fermion and cuprates critical points. An interesting observation is that both heavy fermions and cuprates are in the "Mott" limit where the large Hubbard $U$ induces a constraint on the Hilbert space by forbidding double occupancy. Therefore a framework taking into account Mott physics may be necessary to give an understanding of both the phases and the critical region. For example, a slave boson theory has been proposed to explicitly respect the restriction on the Hilbert space (Lee *et al.*, 2006). One specific continuous Kondo breakdown transition can be successfully described using this slave boson framework (Senthil *et al.*, 2004b). In this theory, carrier density indeed drops across the critical point. But the theory fails to predict the magnetic ordering onset at the same critical point. Nevertheless, the partial success of the slave boson theory is encouraging and suggests that a language which captures the Mott physics may be the key to understanding these exotic phases and unconventional critical points found in heavy fermion systems and in cuprates. It is worth noting that several theories were able to reproduce the drop in carrier density observed in hole-doped cuprates at the opening of the pseudogap: models based on an antiferromagnetic QCP (Chatterjee *et al.*, 2017; Storey, 2016; Verret *et al.*, 2017b), and other scenarios for the pseudogap involving a Fermi surface reconstruction in the Yang-Rice-Zhang theory (Verret *et al.*, 2017a; Yang *et al.*, 2006), the SU(2) fluctuations theory (Morice *et al.*, 2017) or the FL* theory (Chatterjee and Sachdev, 2016).

The carrier density drop becomes even more acute in metal-insulator transitions. In a clean system, at integer filling a metal insulator transition can be driven by increasing the interaction strength or decreasing the bandwidth. Due to the lack of a broken symmetry order parameter, such a transition is generically beyond LGW theory and study of this transition may provide more intuition for the more intricate metal-metal transition in heavy fermion systems. A pressure-tuned metal-insulator transition has been found in organic materials (Furukawa *et al.*, 2015), but the role of disorder and inhomogeneity may need to be considered (this is discussed in a somewhat different context below). The recently discovered Moiré systems may be a powerful platform to explore this physics. In several systems (such as ABC tri-layer graphene aligned with hexagon boron nitride (Chen *et al.*, 2019a), TBG (Burg *et al.*, 2019; Cao *et al.*, 2020c; Liu *et al.*, 2020; Shen *et al.*, 2020a) and twisted transition metal dichalcogenides (Wang *et al.*, 2020)), both the density and the bandwidth can be gate controlled, which makes it much easier to sweep the phase diagram and study phase transitions than in traditional solid state systems.

The 2D superconductor-insulator transition has been considered to be an important model system for quantum phase transitions (Goldman and Markovic, 1998). In fact, much of our intuition about what happens in the LGW perspective on quantum criticality has been developed considering bosonic models of this transition (Goldman and Markovic, 1998; Wallin *et al.*, 1994). Experimentally, one expects that by destroying superconductivity in a 2D thin film with, for example, an applied magnetic field or disorder, a direct transition to an insulating state at $T = 0$ occurs. However, recent experiments suggest that this is not the full story. In fact, most experimental systems actually show a zero temperature transition from the superconducting state to an anomalous metallic phase which has a resistance that is much lower than in the normal state (Kapitulnik *et al.*, 2019), before ultimately becoming insulating at even higher disorder or fields. Involving at least three phases not readily distinguished by symmetry, this transition will likely require theories beyond the LGW paradigm to explain.

Finally, an additional important point is the possibility of the development of a secondary order near the QCP, the most prominent example of this being superconductivity (Scalapino, 2012) that frequently occurs near magnetic critical points in materials like electron-doped cuprates, iron pnictides, and heavy fermions. In this case, the experimental observation of quantum critical scaling would require suppressing the secondary order, which may make, in some cases, experiments under extreme conditions necessary (such as very high magnetic fields in the case of cuprates). This raises the question of whether the observed scaling is affected by these conditions. On the other hand, the emergence of superconductive pairing at the QCP represents an important open problem by itself. While the critical fluctuations may mediate attraction, but on the other hand they can destroy the coherence of the quasiparticles, exemplified by the 'strange metal' regime, perhaps rendering the ordinary BCS, Migdal-Eliashberg approach inapplicable. Thus, it is possible to imagine that coherence is lost to such an extent that superconductivity never occurs. Indeed, the ultimate fate of the competition between NFL phenomena and manifestations of an ordered state is a topic of current interest (Raghu *et al.*, 2015; Wang *et al.*, 2016).

### 2. New frameworks

The above experimental discoveries of unusual metal-insulator and metal-metal transitions clearly indicate

that we need new theoretical frameworks. In the past two decades, one new critical theory beyond the LGW framework that was developed was that of deconfined quantum criticality (Senthil *et al.*, 2004a). It involves fractionalized degrees of freedom and emergent gauge fields in its description. By employing these degrees of freedoms, one can describe several continuous transitions which are now allowed in the Landau framework. Although there has been little experimental evidence for deconfined quantum criticality (DQC), there are some candidate materials (Guo *et al.*, 2020; Lee *et al.*, 2018, 2019b; Zayed *et al.*, 2017b; Zou and He, 2020). In particular, the famous Kitaev material $\alpha$-RuCl$_3$ (Eichstaedt *et al.*, 2019; Kasahara *et al.*, 2018; Laurell and Okamoto, 2020) may host such a deconfined transition between the field-induced Néel state to putative Ising topological spin liquid phase (Zou and He, 2020). It was also recently claimed (Lee *et al.*, 2019a) that there was a deconfined quantum phase transition in the pressure tuned transition of the Shastry-Sutherland lattice compound SrCu$_2$(BO$_3$)$_2$ (Zayed *et al.*, 2017a). More investigations in this regard would be interesting.

To make connections with the experimentally relevant metal-metal transitions in heavy fermion systems or in cuprates, a more sophisticated generalization of the existing deconfined QCP is needed. Most theoretically explored realizations of deconfined quantum criticality, however are restricted to insulator-insulator transitions. There are attempts to describe metal-insulator transitions and metal-metal transitions with carrier density drop using fractionalized degrees of freedom (Senthil, 2008; Senthil *et al.*, 2004b). It remains to be seen whether these attempts can lead to successful explanation of the mysterious non-Fermi liquid phases in heavy fermion systems, cuprates and other strongly correlated materials.

Although disorder effects broadly exist in correlated systems, they become acutely important near QCPs where susceptibilities tend to diverge (Vojta, 2019). For example, they can fundamentally modify the scaling properties in a second order phase transition (Harris, 1974), or stabilize a QCP by rounding a first-order phase transition (Goswami *et al.*, 2008). To better connect experimental and theoretical understandings of QCPs and the effects of disorder, we propose the following workflow:

1. Identify types and levels of disorders. Typically, experiments only estimate an overall quantitative level of disorder. The qualitative nature of the disorder, for example the distribution of impurities or defects, the shape and size of extended defects, etc. may also play an important role in how it affects the physical observables. In particular, many spectroscopic measurements close to a QCP involve spatial averaging over large scales (*e.g.* , X-ray absorption, angle-resolved photoemission). More detailed local experimental probes may be needed (*e.g.* , electron energy loss spectroscopy, scanning tunneling spectroscopy), which can identify and characterize different kinds of disorder on smaller length scales.

2. Estimate effective impurity potentials with abinitio analysis. Based on the experimental findings from local probes, the next step is to derive effective descriptions that capture the specific qualitative nature of disorder in each case of interest.

3. Perform model calculations (numerical and analytical) that take as input the specific effective impurity potentials derived from experiments in the previous step, and compare them with analogous calculations with or without different types of disorder. This will either reveal a sensitivity of the systems to disorder, which would indicate that disorder is an important factor that affects the physical properties near a QCP, or confirm that the spatial averaging over large scales in spectroscopy measurements does not result in the loss of relevant information. Such calculations could also reveal possibilities to tune correlated materials via disorder.

4. Identify better model material systems for quantum criticality for which both the canonical theory and extensions to it can be tested.

From the numerical perspective (Xu *et al.*, 2019), development of controlled techniques with different implementations of disorder may be important to understand the effects of impurities beyond the DFT level, which may be required in correlated materials. Analytical theory, on the other hand, can move further in the prediction of dynamical and non-equilibrium responses. Nonetheless, predicting dynamical and static quantities in the presence of known impurities allows a more rational comparison with experiments. Further identification of intrinsic behavior and understanding disorder effects can contribute to the ultimate goal of quantum material design in a controllable way.

Further establishing quantum criticality beyond the LGW regime and getting further insight into this physics poses significant challenges for experiments. Currently, a plethora of experimental techniques have been developed to probe order parameters and resolve the phase boundary in the multiple-dimensional phase space of pressure, chemical doping, magnetic field, etc., but more sophisticated and comprehensive techniques to resolve symmetry, topology, dynamic correlations, and scaling laws are needed. The development of these experimental tools is not specific to quantum criticality, and will have broad applications across the study of correlated phenomena.

### E. Correlated Topological Matter

The last decade has seen tremendous theoretical and experimental activity at the intersection of band theory and algebraic topology. Substantial progress has been made in identifying topological insulators and (semi)metals that can in principle be realized by systems

of noninteracting (free) fermions (Armitage *et al.*, 2018; Hasan and Kane, 2010; Hasan and Moore, 2011; Kitaev, 2009; Qi and Zhang, 2011; Ryu *et al.*, 2010; Watanabe *et al.*, 2018; Wen, 2012). It is believe that free-fermion topological insulators can be stable to the perturbative inclusion of many-body interactions (Fidkowski and Kitaev, 2010) *e.g.* , the surface states of a 3D $\mathbb{Z}_2$ topological insulator form a 2D Fermi liquid. The focus has now started to be on topological states of matter, where interactions are important (Here we will not consider topological aspects of systems which can also be characterized as quantum spin liquids.). There are at least two kinds of these systems. In the first possibility, interactions drive an ordered state the properties of which determine aspects of the topology, but ultimately a free fermion with little residual interactions description still applies. Chern insulators (in 2D), axion insulators (in 3D), and magnetic Weyl semimetals (Fig. 8(b)) are of this category. In the second possibility, interactions drive systems into a state that has no non-interacting analog (Wang and Senthil, 2014). Here, there are no known examples where this occurs spontaneously, but the fractional quantum Hall effect provides an example for the kinds of effects that could exist. There may be also be systems that straddle these cases, where the systems are "like" those of the non-interacting ones, but perhaps have remaining large residual interactions. This may be exemplified by the large intra-atomic Coulomb energy characteristic of narrow $4f$ bands in the heavy-fermion, topological Kondo insulators (Dzero *et al.*, 2010) or Kondo-Weyl semimetals (Lai *et al.*, 2018).

One ongoing thrust lies in introducing correlations to topological metals. The resultant correlated phase of matter is not necessarily topological but it often has rich physics. As exemplified by $(TaSe_4)_2I$, charge-density-wave correlations in a Weyl semimetal result in an insulator whose electromagnetic response mimics axion electrodynamics (Gooth *et al.*, 2019). Introducing pairing correlations to topological metals is also known theoretically to lead to unconventional superconductivity, e.g, on the Fermi surface of magnetic Weyl metals, the nontrivial Chern number necessitates nodes in the superconducting order parameter (Li and Haldane, 2018; Murakami and Nagaosa, 2003). It would be especially interesting if superconducting correlations are discovered in the Kondo-Weyl semimetals – this would provide a platform to investigate the interplay of Kondo physics, topology and superconductivity.

The rigorous classification of interacting topological matter is incomplete. Ongoing and future efforts should focus on constructing physically realistic models (perhaps by beginning with minimal, exactly-soluble models), and identifying many-body topological invariants which are translatable to many-body physical observables. Some of these observables exist on the boundaries of samples, which favor either surface-sensitive experimental probes such as photoemission spectroscopy, (Alexandradinata *et al.*, 2020b; Belopolski *et al.*, 2019) or

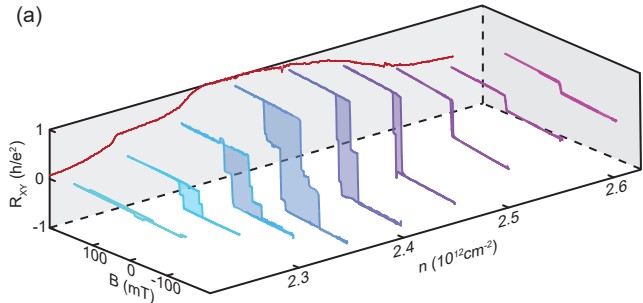

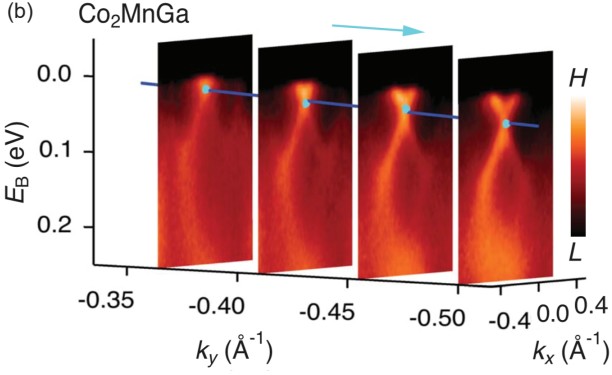

FIG. 8 (a) Intrinsic quantum anomalous Hall effect in twisted bilayer graphene, as adapted from Ref. (Serlin *et al.*, 2020). (b) Angle-resolved photoemission spectrum of a nodal-line degeneracy of $Co_2MnGa$, a ferromagnetic Weyl semimetal candidate. From Ref. (Belopolski *et al.*, 2019)

.

mixed-bulk-surface probes such as quantum oscillations from spatially-nonlocal cyclotron orbits (Moll *et al.*, 2016a). However, not all topological matter has a bulk-boundary correspondence (Helbig *et al.*, 2020; Lepori and Dell'Anna, 2017; Schmidt, 2012; Xiong, 2018; Yao and Wang, 2018). Bulk observables may include thermodynamic quantities such as specific heat, the temperature dependence of which can in principle identify a Kondo-Weyl semimetal (Dzsaber *et al.*, 2017). There has been partial success in exploring the effect of disorder on the stability of many-body topological phases (Wang *et al.*, 2017; Xu and Moore, 2006). Rigorous general results are still lacking, however, and such a dependence may not exist for a subset of topological phases whose robustness rely on crystallographic spatial symmetries. A notion of out-of-equilibrium topological matter is also developing (Cayssol *et al.*, 2013; Ghorashi, 2020; Ghorashi *et al.*, 2018; Lindner *et al.*, 2011a; Nag *et al.*, 2019; Rodriguez-Vega *et al.*, 2018, 2019; Rodriguez-Vega and Seradjeh, 2018; Rudner and Lindner, 2020; Rudner *et al.*, 2013; Titum *et al.*, 2016; Xu and Wu, 2018) and promises to be a rich platform to explore the interplay between correlation effects, topology and many-body localization in periodically-driven Floquet systems (Oka and Kitamura, 2019). It is known that some out-of-equilibrium phases of matter have no equilibrium counterpart (Lindner *et al.*,

2011b; Nag *et al.*, 2019; Xu and Wu, 2018).

Progress in the theoretical understanding of correlated insulators (*e.g.* TBG) has demonstrated an insightful relation between single-particle band topology and many-body correlations; this relation has been missed in previous formulations of effective Hamiltonians for interacting electrons. Namely, it is increasingly recognized that not all crystallographic spacetime symmetries can be imposed locally on the Wannier functions of a band, owing to a topological obstruction (Po *et al.*, 2018). Despite being exponentially-localized in real space, such Wannier functions cannot be localized to a single atomic site (Alexandradinata *et al.*, 2020a), unlike the traditional atomic orbital. These atypical Wannier functions result in non-standard terms in the (generalized) Hubbard model (Kang and Vafek, 2018), with exotic correlated ground states such as an SU(4) ferromagnet predicted (Kang and Vafek, 2019). These issues raise interesting questions about constructing tight-binding Hubbard-like model and considerations about what one might consider to be the strong coupling limit of such models. We speculate that these topics will be important in future studies.

A current bottleneck in the field is that there are very few material candidates of strongly interacting topological matter. An emerging material platform likely to gain more traction is 2D multi-layer heterostructures with van der Waals inter-layer coupling (Ajayan *et al.*, 2016; Geim and Grigorieva, 2013). Their advantage over 3D materials lies in enhanced tunablity through gating, stacking, and twisting. The latter results in artificial Moiré superlattices that can realize strongly-correlated, fractional Chern insulators in fractionally-filled Hofstadter bands (Spanton *et al.*, 2018). Such systems raise the enticing possibility to probe non-abelian quasiparticle statistics. A different class of Moiré superlattices in TBG realizes correlated insulating phases that spontaneously break spin- and valley- symmetries, resulting in an intrinsic quantized anomalous Hall effect (Serlin *et al.*, 2020) (see Fig. 8(a)). The quick realization of this effect in such a relatively clean system (see also Deng *et al.*, 2020) may be juxtaposed with the same effect realized in $(Bi,Sb)_2Te_3$ only after years of optimizing the platform material (Chang *et al.*, 2013). This suggests that disorder is an experimental barrier to discovering correlated topological materials.

The lack of material candidates is especially acute for topological superconductors (Sato and Ando, 2017). There are as of yet little convincing evidence for the original proposal of a proximity effect driven 2D superconductor (Fu and Kane, 2008). Thus far topological superconductivity seems to have been best realized in bulk materials like the iron-based superconductor Fe $Se_{1-x}Te_x$ (Zhang *et al.*, 2018a) or in 1D wires (Mourik *et al.*, 2012; Zhang *et al.*, 2019a). However, twisted van der Waals heterostructures are also promising avenues to realize topological superconductivity (Xu and Balents, 2018). Beside establishing concrete material candidates,

it will be crucial to further develop experimental techniques to identify topological superconductors, such as local probes (*e.g.* , scanning nano-SQUID), which can detect Majorana edge states, as well as bulk probes (*e.g.* , nuclear magnetic resonance (Pustogow *et al.*, 2019)) to constrain the superconducting order parameter. Future theories would hopefully establish why a particular order parameter is energetically favored; sometimes the reasons can be established without reference to a specific pairing mechanism or a detailed microscopic model. One particularly interesting example of topological superconductivity is monopole superconductivity in which the Berry phase structure of a magnetic Weyl system ensures an superconducting order parameter with nodes independent of the mechanism (Li and Haldane, 2018; Murakami and Nagaosa, 2003).

An additional issue in this area is that of finding definitive signatures of strongly correlated topological insulators. It has been proposed that topological insulators are best characterized not as surface conductors, but as bulk magnetoelectrics (Qi and Zhang, 2011) with a quantized magnetoelectric response coefficient whose size is set by the fine structure constant. This magnetoelectric effect was observed in the free fermion systems of $Bi_2Se_3$ through measurements of a quantized Faraday rotation. As alluded to above, one possibility in the case of the strongly interacting topological phases, is the prospect that 3D analogs of 2D fractional quantum Hall phases could be realized. In the same manner as non-interacting topological insulators are expected to show a magnetoelectric effect quantized in units of the fine structure constant, such fractional topological insulators may be expected to show a magnetoelectric effect that is quantized in rational fractions of the fine structure constant (Maciejko and Fiete, 2015; Maciejko *et al.*, 2010; Swingle *et al.*, 2011). Such a fractional phase may be uniquely identified by this fractional magnetoelectric effect. However, a fractional quantized Faraday effect will give a signal smaller that the precision of state-of-the-art THz polarimetry and so new instrumentation may have to be developed.

## F. Revisiting old materials in a modern context

While much of the current research on strongly-correlated electrons focuses on newly developed materials platforms, it is worth considering the value of "old materials" (or "legacy materials") in the context of the strongly-correlated electron problem. Here, by an "old material" we mean a condensed matter system that was studied for a time by the solid state physics community and then largely abandoned as a research field, perhaps many decades ago.

Many old materials are worth revisiting, as in the intervening decades we have developed both new theoretical ideas and better experimental probes. Bringing these probes to old materials often has the advantage

of a greater wealth of expertise about materials synthesis and purification, as compared to more recently-developed materials platforms. At the very least, present-day researchers may find it advantageous to examine old and well-studied materials as test beds for calibrating new experimental methods. Examples of old materials with interesting correlated electron physics can be found across the spectrum of material types: metals, semimetals, and semiconductors. What follows is an illustrative, but far from complete, set of examples. The allure of these examples lies either in the materials demonstrating some unique phenomenon, or in the similarity of some of their properties to those of a more complicated material class (such as high-$T_c$ superconductivity or strange metal behavior).

### 1. Metals

For examples of interesting electron physics, one need not look farther than the elemental metals. Iron is perhaps the simplest example of a magnetic metal, in which magnetism develops in a material with itinerant electrons rather than in an insulator with localized magnetic moments. The magnetism has both itinerant and localized character (Capellmann, 1982; Moriya and Takahashi, 1984; Pepperhoff and Acet, 2013; Stearns, 1978). Similar coexistence of magnetism with metallicity can be found in the iron oxides, such as magnetite (Kukreja et al., 2018; Rozenberg et al., 2006; Zhang and Satpathy, 1991). This type of magnetism has so far eluded a complete theoretical description (see e.g. (Kübler, 2017)).

Even in the absence of magnetism or superconductivity, one can find unconventional transport properties on display in a number of metals, which may give insight into the correlated electron problem. For example, the phenomenon of linear magnetoresistance has recently attracted significant attention due to its appearance in a variety of topological semimetals (Assaf et al., 2013; Butch et al., 2011; Feng et al., 2015; Gusev et al., 2013; Tang et al., 2011; Thomas et al., 2016; Tian et al., 2014; Zhao et al., 2015), as well as in strange metals (Hayes et al., 2016; Sarkar et al., 2019). Yet this effect is on display even in pure potassium, which is ostensibly one of the simplest metals, with a nearly perfectly spherical Fermi surface (Penz and Bowers, 1968; Reitz and Overhauser, 1968). Its postulated origin from the formation of a charge-density-wave (Reitz and Overhauser, 1968) has not been verified, but linear magnetoresistance indeed has been observed in a broad family of density-wave materials, which can generically arise in a partially gapped Fermi surface with sharp corners (Feng et al., 2019). This mechanism, first pointed out by Pippard (Pippard and Pippard, 1989), could also play a role in magnetoresistance of topological semimetals.

Perhaps even more interesting are the liquid metals, which are good conductors for which there is no crystalline order, and therefore no notions of traditional electron or phonon bands (examples include Hg, Ga, Rb, and various alloys, all of which have a melting temperature near or below room temperature). Liquid metals may therefore constitute excellent test beds for the idea of a "Planckian" bound of dissipation (see, e.g. , Ref. Hartnoll, 2015 and Sec. II.C), where the transport scattering time for charge carriers approaches a maximal value and there may be no Fermi liquid-type quasiparticles. Recent experiments have shown that liquid metals exhibit large linear magnetoresistance, which is present only in the liquid phase (Wang et al., 2019).

### 2. Semimetals

A semimetal is a material in which both an electron and a hole band coincide with or are near to the Fermi level; the existence of semimetals has been understood theoretically since the 1930s (see, e.g. , Herring, 1937). Perhaps the most ubiquitously-studied semimetal is graphite, whose band structure has been known since the 1940s (Slonczewski and Weiss, 1958; Wallace, 1947). While graphite seems to exhibit no strongly-correlated physics at zero field, owing to its relatively high band velocities, a magnetic field can quench the electron kinetic energy and greatly increase the role of interactions. A recent investigation using modern pulsed magnetic fields up to 90 T, suggests that graphite may be driven into an excitonic insulator phase (Zhu et al., 2017b).

Experimental studies of crystalline bismuth also have a long history. Large bismuth crystals can be grown with extremely high purity and high electron mobility. In bismuth, light electron bands in three valleys coexist with a heavier hole band, with the electron and hole concentrations being nearly equal and opposite, each on the order of $10^{17}\,\mathrm{cm}^{-3}$ (Jin et al., 2015). This ultralow electron density implies a low Fermi energy in each band, and hence the possibility for electron interactions to play a large role. A magnetic field, in particular, may easily drive crystalline bismuth into a more strongly-interacting phase. For example, one can reach the "extreme quantum limit" of magnetic field, in which only a single Landau level in each band is occupied and strongly-correlated states are expected (Halperin, 1987), using only a few Tesla. Bismuth has been purported to exhibit valley ferromagnetism at sufficiently high fields (Li et al., 2008). The transport properties (and, in particular, the thermoelectric properties) in a magnetic field remain incompletely understood, despite being studied as early as the 1970s (see, e.g. , Ref. Issi et al., 1976). Pressure also drives the charge density of bismuth lower and through a metal-insulator transition near 25 kbar (Balla and Brandt, 1965; Brandt and Ponomarev, 1969). As pressure is applied, the electron band moves up in energy and the hole bands move down (as illustrated in Fig. 9), which in this self-compensated material reduces the charge density (Armitage et al., 2010b). The charge density has been inferred to go to

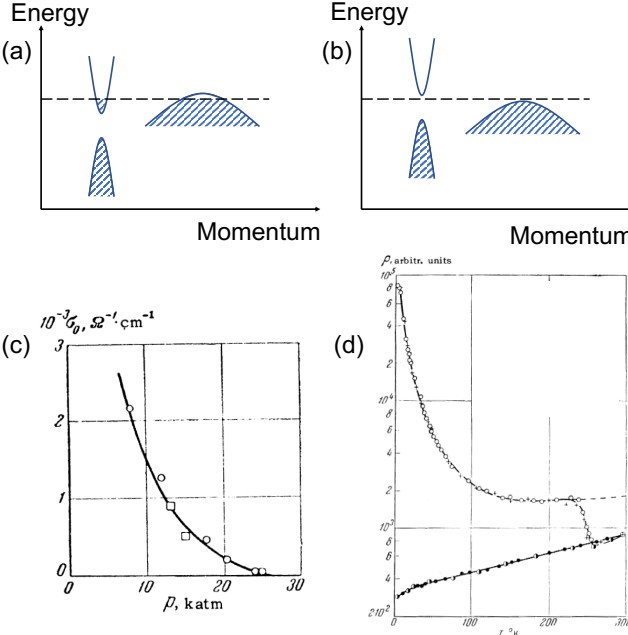

FIG. 9 The metal-to-insulator transition in elemental bismuth as a function of pressure. (a) Under ambient conditions, the band structure of bismuth is such that both electron and hole pockets intersect the chemical potential (dashed line). (b) Under high pressure, the electron pockets move up in energy and the hole pockets move down, which produces a metal-insulator transition. (c) The low-temperature conductivity as a function of pressure (Figure from Ref. Balla and Brandt, 1965), taken at $T \approx 2$ K. (d) The transition is evident in the dependence of the resistivity (vertical axis) on temperature (horizontal axis). The lower curve shows data for a sample at ambient pressure, which has a metallic-like temperature dependence, while the upper curve shows data for a sample under $24,500$ atm. of pressure, which has an insulating-like temperature dependence. (From Ref. Balla and Brandt, 1965.)

zero near 25 kbar, although the nature of the resulting metal-insulator transition is unclear and a simple non-interacting Lifshitz-like transition seems unlikely. Strong correlations and strongly dressed plasmaron quasiparticles (strongly coupled electron-plasmon composites) were inferred from infrared spectroscopy under pressure near the metal-insulator transition (Armitage et al., 2010b; Tediosi et al., 2007).

A poster-child for the value of reconsidering old semimetals is the Kondo insulator $SmB_6$. This material was discovered over 50 years ago, and was the first identified as a "Kondo insulator" – a material for which a heavy electron band (with "nearly localized" magnetic moments) hybridizes with another light one to produce a gap at the Fermi level (Allen et al., 1979; Menth et al., 1969; Nickerson et al., 1971). Reconsideration of this material in the 21st century led to the realization that the Kondo band gap might be topological in nature, and the corresponding topological surface states

helped to explain a 40-year-old puzzle about the saturation of the electrical resistance at low temperature (Dzero et al., 2010, 2016). Recent observation of quantum oscillations in $SmB_6$ increased attention because of their suggestion of a charge-neutral Fermi surface (Tan et al., 2015), although this remains controversial (Li et al., 2020). It has also been known that the Sommerfeld coefficient of the specific heat of $SmB_6$ is unusually large, $\gamma \approx 10$ mJ/mol·$K^2$ (Gabani et al., 2001; Phelan et al., 2014) – 10 times larger than metallic $LaB_6$. Such large fermionic specific heat has been shown to be a bulk effect (Wakeham et al., 2016). Additionally, THz range conductivity experiments of $SmB_6$ have revealed in-gap conduction consistent with a localized response with conductivities orders of magnitude larger than the dc value (Gorshunov et al., 1999; Laurita et al., 2016; Travaglini and Wachter, 1984). Although impurity band conduction is an obvious culprit, the magnitude of these signals is generally orders of magnitude larger than of corresponding impurity bands in conventional semiconductors. However, there is some recent evidence that these issues can be understood by realizing that electronic dispersions are quite unlike the parabolic ones of conventional semiconductors and treating the wavefunctions of impurity states appropriately (Skinner, 2019).

### 3. Semiconductors and insulators

From a practical standpoint, semiconductor physics has been the overwhelming success story of solid state physics, leading to transformative new technologies throughout the second half of the 20th century. It can seem surprising, then, to note that semiconductors continue to harbor surprises and profound mysteries during the 21st century as well.

One prototypical example of an old semiconductor that continues to serve as a fount of new physics is $SrTiO_3$. This relatively large band-gap material generated significant interest in the 1950s and '60s due to its anomalously large dielectric constant at low temperature, which results from an aborted ferroelectric transition at low temperature (Barrett, 1952; Cowley, 1964; Neville et al., 1972; Yamada and Shirane, 1969). The large dielectric constant enables both metallic and superconducting behavior at anomalously low electron density (Bhattacharya et al., 2016; Koonce et al., 1967; Lin et al., 2013). At higher temperatures, experiments have shown a metallic conductivity coexisting with an apparently huge violation of the Mott-Ioffe-Regel criterion (Collignon et al., 2019; Lin et al., 2017). A naive application of the Drude theory yields a mean-free-path that is shorter than the lattice constant, suggesting that the electron transport in this regime may involve a description beyond the traditional kinetic theory of Fermi liquid quasiparticles.

Even the world's best-studied semiconductor, silicon, remains a fruitful platform for addressing unsolved prob-

lems in strongly correlated electron physics. For example, doped silicon (like essentially all semiconductors) undergoes an insulator-to-metal transition (IMT) with increasing doping. There is a long history of studying this transition in phosphorus-doped silicon (Kar *et al.*, 2003; Kravchenko and Sarachik, 2003; Rosenbaum *et al.*, 1980, 1983; Thomas *et al.*, 1983), but the nature of the transition was never completely understood (Dobrosavljevic *et al.*, 2012; Lee and Ramakrishnan, 1985). While IMTs are commonly discussed from the perspective of the Anderson transition, *i.e.* a disorder-driven transition for which interactions play no role, this perspective does not adequately describe the IMT in doped semiconductors. The long-ranged Coulomb interactions between electrons localized on discrete dopant atoms play a crucial role in the transition, making the IMT in doped semiconductors a preeminent strongly correlated problem, for which every site energy depends on the occupation of every other site. As the archetypal example of a doping-induced transition, phosphorus-doped silicon near the IMT remains an excellent platform for testing new experimental probes of temporal and spatial electron correlations (for example, by measuring optical conductivity (Helgren *et al.*, 2002) and in optical pump-probe experiments (Thorsmølle and Armitage, 2010)). Applying such probes to the doping-induced IMT may provide crucial insight to the correlated electron problem.

Finally, it is worth mentioning that certain semiconductors exhibit anomalous electronic properties in the vicinity of a structural phase transition. For example, $Cu_2Se$ undergoes a structural transition at temperature $T \sim 350$ K. It has been known since 1971 that this transition is accompanied by a spike in thermopower, hinting at the possibility of a strong renormalization of electronic carriers (Okamoto, 1971). Revisiting the thermopower of $Cu_2Se$ in 2018 showed that this spike can produce an enormous thermoelectric figure of merit, $zT \sim 300$, in the immediate vicinity of the transition (Byeon *et al.*, 2019). The nature of the charge and heat transport in the vicinity of this transition remains an open question.

## 4. Superconductivity

As alluded to elsewhere in this manuscript, the details of the pairing mechanisms of many "old" superconductors are still poorly understood. From the rare earth borocarbides, amorphous and crystalline bismuth, $SrTiO_3$, doped $BaBiO_3$, or even $MgB_2$ many details are unknown (Buzea and Yamashita, 2001; Gastiasoro *et al.*, 2020; Meregalli and Savrasov, 1998; Shier and Ginsberg, 1966; Takagi *et al.*, 1997). In particular, these systems provide a wealth of test cases to investigate the interaction between superconductivity and magnetism (*e.g.* , reentrance in rare earth borocarbides), structure ($BaBiO_3$) and spin-orbit physics (Bi).

A number of doped semiconductors exhibit "superconducting domes" akin to the cuprates. For example, in the bismuthates (which are diamagnetic semiconductors), doping with potassium yields a superconducting dome with a maximal $T_c \sim 30$ K (Sleight, 2015). The electron-phonon coupling calculated by a density functional and Migdal-Eliashberg theory approach is insufficient to account for the high $T_c$ in the bismuthates. Many of the systems necessitate a non-BCS, Migdal-Eliashberg explanation because the Debye frequency is much larger than the Fermi energy, and therefore electron-phonon coupling of a conventional variety cannot be the mechanism for electron pairing (although it could still be electron-phonon coupling of an unconventional variety). In this regard, recent work (Yin *et al.*, 2013) claims that standard approaches underestimate large nonlocal correlation effects that can enhance the electron-phonon coupling and enhance $T_c$. The mechanism for superconductivity in doped $SrTiO_3$ also continues to generate intense interest in this regard (see Ref. Gastiasoro *et al.*, 2020 for a review).

The superconducting properties of bismuth are similarly fascinating. One would generically not expect superconductivity in bismuth owing to its very low density of electron states, but a very recent study has identified superconductivity in crystalline bismuth (with a $T_c \approx 0.5$ mK) (Prakash *et al.*, 2017). Such superconductivity at low density cannot be described by the conventional BCS, Migdal-Eliashberg theory, since it requires the Fermi energy to be much larger than the Debye frequency, while in bismuth they are comparable. BCS theory also predicts a ratio between the critical magnetic field and critical temperature that is an order of magnitude larger than the value observed. Perhaps even more surprising is that *amorphous* bismuth is also a superconductor, with a $T_c \sim 6$ K that is more than four orders of magnitude larger than in crystalline bismuth (Shier and Ginsberg, 1966). As it happens, bismuth is representative of a larger class of materials for which $T_c$ is higher in the amorphous state than in the crystalline state (Missell, 1985; Tsuei *et al.*, 1977). There is some evidence that both the electronic density of states and the electron-phonon coupling is larger in amorphous bismuth (Mata-Pinzón *et al.*, 2016), but it is fair to say that this is not fully understood. Finally, the existence of liquid metals (and superconductivity in amorphous metals) suggests the tantalizing (and relatively unexplored) question: can there be a liquid superconductor?

A superconducting dome "high-$T_c$"-like phenomenology is also on display in the elemental magnetic metals. In iron the itinerant magnetism in the bcc phase is suppressed with external pressure and enters a superconducting dome in the hcp phase (Shimizu *et al.*, 2001). As another example, elemental chromium exhibits a superconducting dome much like that of the cuprates (albeit with a much lower $T_c$) when doped with ruthenium, rhodium, or iridium (Matthias *et al.*, 1962; Ramazanoglu *et al.*, 2018). One can also produce a relatively high $T_c$ two-dimensional superconductor by growing monolayer or near-monolayer films of lead on appropriate sub-

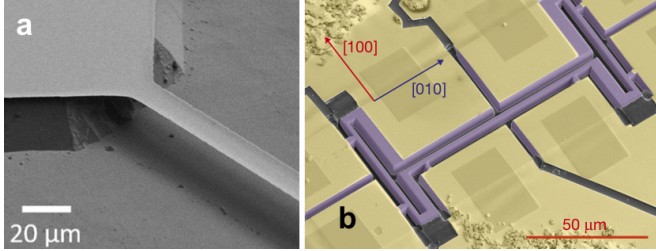

FIG. 10 Micromachined samples. (a) An SEM image of a suspended crystalline sample of BaTiO₃ produced using wet-etch techniques. From Ref. (Lim *et al.*, 2020). (b) A micro-engineered sample of CeRhIn₅ machined from a bulk crystal using focused ion beam milling, ideal for measurements at high magnetic fields. From Ref. (Ronning *et al.*, 2017)

strates (Brun *et al.*, 2016; Sekihara *et al.*, 2013; Toyama *et al.*, 2018). One would like to understand if super-conductivity in these materials is of the unconventional variety.

## III. WHAT CAN AND SHOULD WE DO?

### A. The Role of Materials Synthesis and Discovery

Materials play a key and obvious role in the correlated electron problem, as after all, materials actually host the electrons that are correlated. As McQueen said at the workshop, "A materials discovery has occurred when a known or newly created material exhibits phenomenology and behavior not obviously explainable by our current understanding and theories of the universe." It is the challenge inherent to the correlated electron problem that it is difficult to know *a priori* where to look for new behavior in these materials, but the past provides guides. The discovery of superconductivity in the cuprates was driven by the notion that electronic properties of metallic oxides were underexplored and that strong Jahn-Teller coupling might drive superconductivity in new materials (Bednorz and Müller, 1986). The latter idea is likely not to be the correct explanation of superconductivity in the cuprates, but it was a new physical idea that drove research in a new direction. The discovery of the fractional quantum Hall effect in 2D electron gases was enabled by the development of ultra-clean heterostructures and a long effort in trying to understand the effects of localization and delocalization in two dimensions. Localization is not the driver behind the physics of the fractional quantum Hall effect, but again it pushed the community to look in new and unexplored directions.

Materials oriented scientists are vitally important in at least four capacities: discovering new and interesting materials, improving the quality of existing materials, using aspects like doping to change material properties, and developing heterostructures and new material configurations (*e.g.* "twisted" compounds). Bulk synthesis provides the community with large crystals of both es-

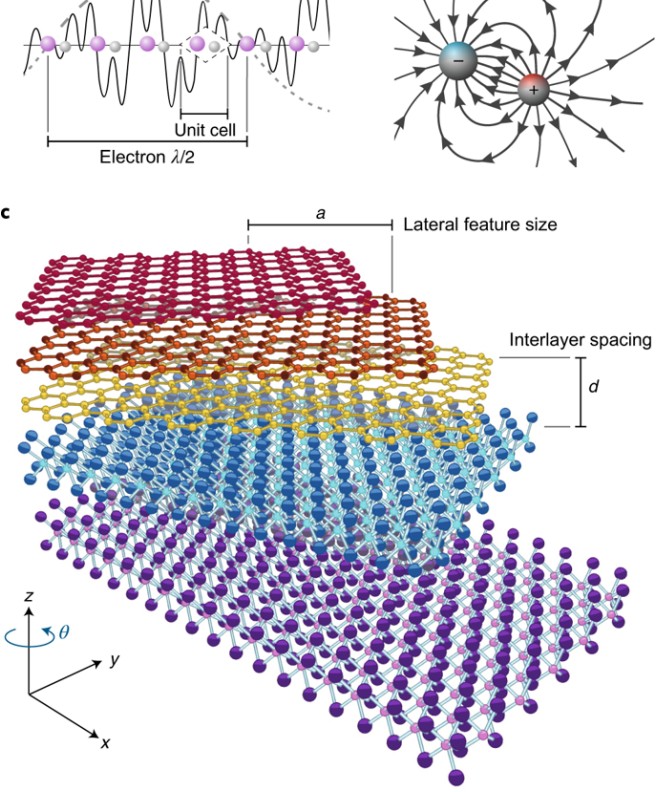

FIG. 11 2D van der Waals heterostructure systems provide an incredible opportunity to tune material properties. (a) The crystal unit cell introduces a repeating structure that modifies the electronic wavefunctions. (b) The dielectric environment, which has reduced screening in 2D, modifies the local Coulomb interaction. (c) 2D van der Waal heterostructure quantum metamaterials are composed of individual 2D layers (transition metal dichalcogenide, graphene or boron nitride) and characterized by lateral repeat distance size $a$, interlayer spacing $d$ and atomic-layer twist angle $\theta$. From Ref. (Song and Gabor, 2018).

tablished and new materials, while atomic-scale growth techniques allow for the synthesis of these systems in epitaxial thin film form and in artificial layered heterostructures where structural and chemical degrees of freedom can be systematically controlled.

As we move forward in investigating correlated electron materials, there is a need to push well-known existing techniques – such as flux growth, floating zone, Bridgman, molecular beam epitaxy, pulsed laser deposition, chemical vapor deposition, etc. – forward by extending them to new frontiers (Canfield, 2019; Schmehr *et al.*, 2019) and by incorporating techniques common in other fields such as chemistry and materials engineering. For example, freestanding films developed from wet etch techniques, plasma growth, or exfoliation (Bhaskar *et al.*, 2016; Lim *et al.*, 2020; Lu *et al.*, 2016; Tatarova *et al.*, 2017) can serve as novel substrates for extending the range of lattice parameters and crystal symme-

tries available for growing strongly correlated electron materials. Free standing single atom thin layers have been achieved in the case of graphene (Du *et al.*, 2008; Tatarova *et al.*, 2017), but one may wonder if it is possible for other systems like cuprates. Topochemical anion exchange allows for the stabilization of oxidation states not available by conventional synthesis techniques (Lefler *et al.*, 2019; Li *et al.*, 2019a). Quite recently, these techniques gave us superconductivity in Ni based oxides (Li *et al.*, 2019a). There is the development of hybrid growth techniques with unconvetional substrates that can lead to interesting materials (Lichtenberg *et al.*, 1992; Yao *et al.*, 2019). Modern techniques such as focused ion beam milling (Fig. 10) should continue to be developed for making engineered samples from bulk crystals (Moll, 2018). We must also continue to innovate new methods for materials assembly, as evidenced by the remarkable continuing progress in the construction of 2D material heterostructures (see Fig. 11) that has culminated in TBG but is not just limited to (Ajayan *et al.*, 2016; Cao *et al.*, 2018; Geim and Grigorieva, 2013; Kim *et al.*, 2016; Novoselov *et al.*, 2016; Rhodes *et al.*, 2019). In the case of TBG, although a magic angle causing relatively flat bands was anticipated (Bistritzer and MacDonald, 2011), the remarkable phenomenology exhibited by multiple superconducting domes and insulating regions was a surprise (Cao *et al.*, 2018; Lu *et al.*, 2019; Yankowitz *et al.*, 2019). This shows that novel combinations of materials can frequently reveal surprising new physics.

A particularly promising direction is the design of materials with novel macromolecular structures. This is of course relevant in the rise of 2D materials heterostructures and superconducting $C_{60}$, but also in the consideration of other molecular structures. For instance, there is a series of "1-2-20" Pr-based "cage compounds" in which the $Pr^{3+}$ and transition metal ion reside in different atomic cages. The localized $Pr^{3+}$ $4f$ electron states hybridize with the ligand states of the 16 surrounding X cage ions resulting in a nonmagnetic non-Kramers $\Gamma_3$ ground state in the cubic crystal field. They have shown evidence or a quadrupolar Kondo effect *e.g.* the two channel Kondo effect. It has been proposed that magnetostriction could be a very diagnostic test for multipolar orders in this material class (Patri *et al.*, 2019). The compounds $PrTi_2Al_{20}$ (Sakai *et al.*, 2012) and $PrV_2Al_{20}$ (Matsumoto *et al.*, 2016) have been reported to display unconventional SC with $T_c$'s of 0.2 K and 0.06 K. The SC coexists with ferroquadrupolar (FQ) order in $PrTi_2Al_{20}$ ($T_{FQ} = 2$ K) and antiferroquadrupolar (AFQ) order in $PrV_2Al_{20}$ ($T_{AFQ} = 0.6$ K). In another group of Pr-based filled skutterudite "cage compounds" the $Pr^{3+}$ ions reside in an atomic cage but have a $\Gamma_1$ singlet ground state in the crystal field. The $PrOs_4Sb_{12}$ (Bauer *et al.*, 2002; Maple *et al.*, 2006) and $PrPt_4Ge_{12}$ (Maisuradze *et al.*, 2010) compounds are nonmagnetic and exhibit an unconventional type of SC with $T_c$'s of 1.86 K and 7.9 K, respectively. The SC appears to have gap nodes and breaks time reversal symmetry. It has been proposed to

be a candidate 3D topological superconductors that could support Majorana fermions (Kozii *et al.*, 2016). This general idea of using large molecular clusters as a building block for new physics is also relevant for searches for new spin liquid platforms in magnetic cluster compounds like $LiZn_2Mo_3O_8$ (Sheckelton *et al.*, 2014), and the remarkable configurability of metal-organic frameworks (Misumi *et al.*, 2020; Takenaka *et al.*, 2018; Yamada *et al.*, 2017; Zhang *et al.*, 2017c). Macromolecular structures represent a whole world of relatively unexplored possibilities.

A key theme in materials synthesis as we consider the future of the correlated electron problem is to understand the role of disorder and defects at both the atomic level and meso-and macroscopic levels. To that end it would be powerful to incorporate atomic and electronic structural and chemical characterization *in situ* during material synthesis to give information on sample properties in real time (He *et al.*, 2018a; Shen *et al.*, 2017). Moreover, it will be important to more rigorously *ex situ* characterize the interplay between defects and phenomena (Cao *et al.*, 2017; Muller *et al.*, 2012). A particular challenge will be to incorporate materials synthesis with measurements of physical properties under extreme conditions such as low temperatures and high fields where correlated electronic effects and phases are experimentally accessible.

Future directions need to emphasize the feedback loop between theory which predicts structural motifs for stabilizing novel phases, materials synthesis, and characterization. The expansion of social networks for collaborative synthesis and cloud efforts may accelerate progress in materials development through greater access to advanced techniques and shared understanding of theory and multi-characterization of similar samples. To enable more efficient exploration and higher throughput in the material synthesis phase space, standardization of protocols and delocalized crowd-sourcing synthesis are arguably as important as methodological innovations. In this regard, we expect future materials innovation to consist of two complimentary modes of operation: i) individual research labs will continue to lead technological innovations and targeted synthesis of novel materials of interest on a case-by-case basis. ii) more distributed, either government sponsored or industry invested "cloud synthesis" stations will become massively parallel to realize high throughput phase space exploration.

Such organization disentangles the standard part of material discovery from more individual-case based explorations, thereby not only liberating more workforce towards process i), but also improves the comparability, repeatability, and efficiency of process ii) by removing the human preference factor. Moreover, with the rapid developments of robotics and AI assisted material prediction today, process ii) can be executed with such standardization that it readily connects to the industrial scene better, and also enables more efficient material recycling (especially for toxic and rare elements). When combined with automated, standardized physical property charac-

terizations such as simple Raman spectroscopy and resistivity, such mode of operation provides a basis for the accumulation of big data, and lays the ground for automatic detection of materials with single or combined "outlier" properties. This will obviously require intensive coordination and it will be necessary for researchers, industrial investors and policy makers to gather frequently and discuss standard protocols in light of new instrumentation and scientific developments from the research sector, much like how IEEE operates today.

## B. Numerical methods

In the traditional division between theoretical and experimental physics, numerical (or computational) physics plays a three-fold role: (1) It acts as a bridge between analytical theory and experiment, for example by connecting microscopic or many-body models with complex experimental systems. (2) It facilitates new types of (numerical) experiments. Numerical simulations can be cheaper, faster, and easier to control than physical experiments. (3) It is a theoretical tool to describe complex systems for which an analytical description is unavailable and perhaps even impossible. Strongly correlated electronic materials and models can be examples of such systems.

For each level of theoretical description in the strongly correlated electron problem, see Fig. 12, there are numerous numerical methods available. We might hope that the parameters of the upper-level theories can be estimated using the lower-level theories *e.g.* models with fewer degrees of freedom may be parametrized from models with more degrees of freedom. However, this may not always be possible or efficient. Some effective theories can only be postulated and their parameters obtained by fit to experiment. Note that the arrows are two-headed as both bottom-up and top-down reasoning are possible. For instance, one can attempt to deduce what materials could give an particular effective Hamiltonian, such as the work that showed the Kitaev model could be possibly realized in strong spin-orbit coupled systems (Jackeli and Khaliullin, 2009; Takagi *et al.*, 2019), or what materials could exhibit certain phenomena (such as high-$T_c$ superconductivity). Alternatively, one can attempt to construct lattice models explicitly from microscopic theory using the downfolding techniques discussed below.

Microscopic model methods, based on directly solving the many-body Schrödinger equation, include density functional theory (DFT) (Hohenberg and Kohn, 1964; Kohn and Sham, 1965), *ab-initio* quantum Monte Carlo (Ceperley and Alder, 1980; Wagner and Ceperley, 2016) and quantum chemistry methods (Shao *et al.*, 2015). There are also many methods to solve effective models, such as Hubbard (Hirsch, 1985; LeBlanc *et al.*, 2015), Heisenberg (Sandvik, 1997; Yan *et al.*, 2011) and other phenomenological lattice models exist. Within these broad classes, there are many

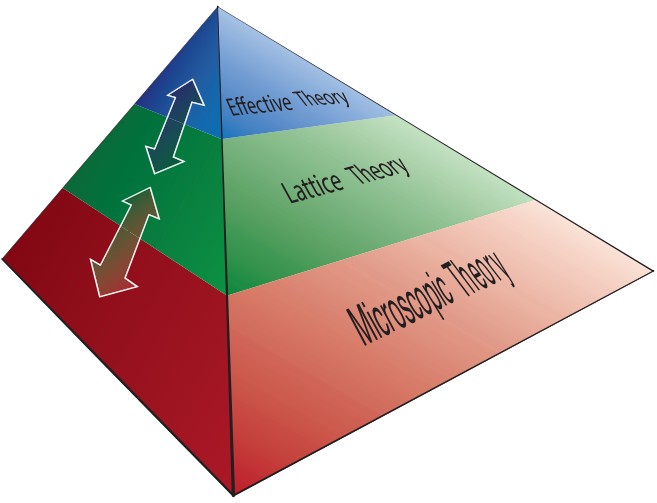

FIG. 12 At least three classes of theory exist: microscopic theory, involving *ab initio* density functional theory based on chemical compositions; lattice theory, involving many-body wavefunctions with reduced number of degrees of freedom in low-energy lattice models; effective models that attempt to capture essential properties of a system with the minimum additional complexity.

types of both exact and approximate algorithms based on numerical techniques, for example, large scale exact (Wietek and Läuchli, 2018) or selected diagonalization methods (Holmes *et al.*, 2016), numerical renormalization group (Wilson, 1975) and density matrix renormalization group (DMRG) (Schollwöck, 2005; Stoudenmire and White, 2012; White, 1992), tensor networks (Changlani *et al.*, 2009; Corboz *et al.*, 2014; Verstraete *et al.*, 2008; Vidal, 2008), dynamical mean field theory (DMFT) (Georges *et al.*, 1996; Kotliar *et al.*, 2006), density matrix embedding theory (DMET) (Knizia and Chan, 2012), and stochastic Monte Carlo methods (Blankenbecler *et al.*, 1981; Booth *et al.*, 2009; Gull *et al.*, 2011; Petruzielo *et al.*, 2012; Prokof'ev *et al.*, 1998; Sandvik, 1999; Trivedi and Ceperley, 1990; Zhang *et al.*, 1997). There has also been progress in connecting the different levels of theoretical description, *i.e.* , building effective models from microscopic models, using techniques known as downfolding (Aryasetiawan *et al.*, 2004; Pavarini *et al.*, 2001), but this is still challenging (Honerkamp *et al.*, 2018; Kent and Kotliar, 2018). There are many advantages and limitations of the methods used to study strongly correlated electrons at each level, with properties such as finite system-size, dimensionality, and nature of interactions that determine the range of applicability.

State-of-the-art computing techniques give accessibility to well-controlled approximations of physical quantities that are often unsolvable in the framework of analytical theory. Strongly correlated electronic systems are composed of a vast, exponential-scaling of many degrees of freedom. Therefore, experimental exploration of all variables is impractical, making simulation-based solu-

tions critical for future success. Numerical methods can establish a practical connection to experimental observations. Wave-function-based methods at the *ab-initio* or many-body model level give access to some excited-state wavefunctions, and therefore can make predictions about experimental spectra (Wang *et al.*, 2018a). Monte-Carlo-based methods, which utilize rigorous statistical sampling to evaluate the properties of large-scale systems, provide unbiased predictions for finite-temperature observables when applicable (Blankenbecler *et al.*, 1981; Gull *et al.*, 2011). Numerical methods can also help guide experiments, for example on symmetry or topology principles, and analytical theory. In some particular cases, numerical methods bring extra mathematical perspectives to describe the physics. For example, in tensor-network-based methods, entanglement entropy, originally introduced in quantum information theory, governs the accuracy of numerical solutions and provides a useful tool for theoretical analysis (Schollwöck, 2005).

Numerical methods should not be treated as black boxes. Their limitations need to be clearly stated and understood both by practitioners and users, including experimentalists looking for a theoretical connection to their findings. Microscopic models, by virtue of fully describing the electronic degrees of freedom, are more complex than many-body effective models. Consequently, the methods that directly target microscopic models tend to rely on more approximations than methods for effective models. This trade off, which applies throughout all levels of the pyramid, can be described on a continuum between "solving an exact model approximately" versus "solving an approximate model exactly." These are imperfect, but complementary approaches, that are especially important in the correlated electron problem, where the validity of approximations often lack *a priori* theoretical justification.

A limitation in numerical methods for correlated electron problems is the fermionic sign problem, which restricts the applicability of some exact Monte Carlo techniques to general models of interacting fermions and frustrated spin systems (Li and Yao, 2019). One school of thought has been to search for, or design, sign problem-free effective models (Alet *et al.*, 2016; Berg *et al.*, 2019; Kaul *et al.*, 2013; Li and Yao, 2019). Another approach is diagrammatic Monte Carlo methods, which turn negative signs arising from fermionic statistics into an advantage (Prokof'ev and Svistunov, 2007). These techniques are based on the stochastic sampling of Feynman diagram expansions at all relevant orders, without uncontrolled truncation. In this formalism, the alternating fermionic signs result in a faster (exponential) convergence of the diagrammatic series, which can offset the exponential scaling of computational time at a given order (Rossi, 2017; Rossi *et al.*, 2017).

While Monte Carlo methods tend to be computationally expensive, it is possible to harness modern computational power and massive parallelism to make progress. Most methods focus on ground state or equilibrium thermal properties, and obtaining dynamical properties and response functions reliably from them remains a major challenge. For example, traditional quantum Monte Carlo methods are formulated in imaginary time which provides access to thermal properties, but calculating frequency dependent measurements requires numerical analytic continuation (Goulko *et al.*, 2017; Jarrell and Gubernatis, 1996). Progress has been made in formulating Monte Carlo methods in real time, providing direct access to the real-frequency axis (Cohen *et al.*, 2015), but these face the challenge of the dynamical sign problem (Mühlbacher and Rabani, 2008). Tensor network or matrix product state methods (Daley *et al.*, 2004; White and Feiguin, 2004) can be formulated in real-time, but suffer from growing entanglement with time propagation, restricting the accuracy of low-frequency properties (Paeckel *et al.*, 2019). Exact diagonalization methods can perform exact calculations of dynamical spectra directly on the real frequency axis (Jaklič and Prelovšek, 1994), but finite-size effects complicate the interpretation of spectra which may exhibit spurious features on small systems. Therefore, using embedding methods and cross-benchmarking multiple numerical methods become important to validating properties in the thermodynamic limit.

There are several difficult-to-compute quantities of interest that are crucial for understanding the underlying physical principles of strongly correlated electron systems, as well as for relating theory and experiment. We need to focus on developing improved methods to study for example, excited states and spectroscopy, finite-temperature systems, dynamical properties, and disorder. Another key challenge is determining the appropriate parameters and energy scales in effective lattice models for describing real materials. This involves finding a controlled and systematic way to relate, for example, microscopic parameters such as the hopping $t$, Hubbard interaction $U$, and charge transfer energy $\Delta$, to experimentally measurable observables such as $T_c$ (for a concrete example relevant to the cuprates, see Ref. (Weber *et al.*, 2012)).

Downfolding techniques connect different levels of description, which will facilitate precise predictions for experiments and enable material design. Equally important is quantifying the accuracy of these low-energy model Hamiltonians – how do interaction parameters depend on doping, pressure, field? Are higher-body effective interactions (such as ring exchange) important to incorporate (Paul *et al.*, 2020)? It must be noted that downfolding methods have been extensively used for deriving tight-binding parameters in the DFT community (Pavarini *et al.*, 2001), and for DFT+DMFT calculations (Kotliar *et al.*, 2006), however, the estimation of interaction parameters requires approximations that are not well controlled or understood (Honerkamp *et al.*, 2018; Kent and Kotliar, 2018). Many-body formalisms that give equal footing to the kinetic and potential energy parts of the problem, which do not depend on tradi-

tional band theory, are being developed (Changlani *et al.*, 2015; Requist and Gross, 2019; Rusakov *et al.*, 2014; Zheng *et al.*, 2018) and ideas from information compression, quantum information, tensor methods, and renormalization groups may have an important role to play. Further discussions and collaborations between the condensed matter physics and quantum chemistry communities is potentially crucial for progress on this front.

Another interesting route related to the interplay between scales is the design of effective models with desired properties. For example, if one desires a superconducting or quantum spin liquid ground state, one can ask what model Hamiltonian realizes such a state. This is an inverse problem that can be approached numerically which may help in the design of future materials and may help develop intuition for where in parameter space to look for desirable strongly correlated phases (Chertkov and Clark, 2018).

Finally, we foresee that the rapidly developing technologies in artificial intelligence, machine learning, quantum computing, and quantum simulation will play a vital role in how numerical techniques will address the strongly correlated electron problem. Deep neural networks (LeCun *et al.*, 2015) and other machine learning methods, such as computational graphs (Kochkov and Clark, 2018), are already actively being used as trial wave functions in variational Monte Carlo (Carleo and Troyer, 2017). Machine learning methods, because of their ability to reveal correlations in large datasets, also hold promise as tools for discovering new strongly correlated material candidates through the analysis of large databases of material properties (Saal *et al.*, 2013). In addition, the emerging technology of noisy intermediate-scale quantum (NISQ) computers (Preskill, 2018) will complement our existing classical computing methodologies. NISQ devices are capable of simulating many-body quantum systems that are difficult to simulate classically, even with the best known supercomputers (Arute *et al.*, 2019). NISQ devices, and their eventual error-correcting successors, are powerful tools for solving effective models of strongly correlated electrons, such as the Hubbard model, using, for example, hybrid quantum-classical algorithms such as the variational quantum eigensolver (Peruzzo *et al.*, 2013). For these reasons, we believe that machine learning methods and quantum computers will help us make progress in tackling these problems. There is also great promising in the areas of quantum simulation where ones handles the exponential proliferation of a Hilbert space that characterizes a large system, by "fighting fire with fire" (Houck *et al.*, 2012) by simulating a one quantum system by another – simpler to control – quantum system. Possible implementations are in superconducting circuits (Houck *et al.*, 2012), quantum dots (Manousakis, 2002), and cold atoms (Bloch *et al.*, 2012).

## C. Analytical methods

### 1. New approaches

The complicated nature of strongly correlated electronic systems severely limits the power of analytical methods that are often based on an expansion with a small parameter around a well-defined ground state. However, considering the difficulties of numerical methods in strongly correlated systems and their limited predictive power compared to the case of single-particle problems, the demand is high for further development of analytical methods to gain unbiased and transparent insight into the physics of strongly electronic systems. Moreover, analytical approaches are, at least in principle, more straightforward to generalize to non-equilibrium problems in a controlled manner, *e.g.* with Keldysh approach, while in their numerical counterparts semi-empirical analytical continuation techniques have to be used. In spite of the difficulties mentioned, analytical approaches are still the most powerful way to understand correlated systems. Building on those successes, we discuss below the possible paths for analytical theory of correlated systems to move forward.

One striking example of a solution to a correlated electron problem is the fractional quantum Hall effect (FQHE), where a ground-state wavefunction was explicitly constructed which explained the properties of the system (Laughlin, 2000). It would be the ideal to use this strategy to approach some other important states of correlated matter, such as NFL metals and QSLs, and perhaps get new examples of their realization amenable to theoretical studies. In particular, for NFL states one may try to find the wavefunctions from known examples of models having a NFL state (such as fermionic systems at a QCP (Löhneysen *et al.*, 2007b; Schofield, 1999) or the Sachdev-Ye-Kitaev (SYK) model (Kitaev, 2015a,b; Sachdev and Ye, 1993)) and introduce additional parameters to obtain families of NFL wavefunctions with the aim of extracting general properties of NFL states. One may hope to find certain generic forms for the wavefunctions of the excitations in the NFL ground state. Essentially, breakdown of quasiparticle description (or, more precisely, the vanishing of the quasiparticle weight) of excitations is just a sign that single-particle electron-like states are a "bad basis" for the Hilbert space of excitations in an NFL. One expects the proper eigenfunctions to be a superposition of states with different number of particle-hole excitations, pointing to some kind of generalized many-body coherent state.

The example of the Kitaev model (Jackeli and Khaliullin, 2009; Kitaev, 2006; Takagi *et al.*, 2019) (discussed above in Sec. 5) for a spin liquid with anisotropic Ising interactions shows that proposing – perhaps unrealistic – toy models with the desired ground state may lead to tremendous progress in both theoretical understanding and future experimental guides. This is also what transpired with Haldane's honeycomb lattice model in the

context of the quantum anomalous Hall effect (Haldane, 1988). In building up theoretical descriptions of many-body systems, exact models – such as Kitaev's – are important (despite their frequent artificiality) because they establish the point of principle that a particular phase of matter could exist. But they can also be motivating to search for new ways to stabilize these states of matter. In the Kitaev case, it has been shown that despite its contrived nature (Ising interactions with different quantized directions on each bond), its anisotropic interactions can actually arise through the effects of strong spin–orbit coupling (Jackeli and Khaliullin, 2009; Takagi et al., 2019). Thus, constructing new toy models with NFL or QSL ground states while using the known ones as a starting point (cf. SYK (Chowdhury et al., 2018b; Patel and Sachdev, 2019; Song et al., 2017) or Kitaev models (O'Brien et al., 2016)) may be fruitful.

Another strategy for making progress in the correlated electron problem is to exploit general symmetries and properties of quantum mechanical descriptions to derive exact statements that are valid regardless of the correlation strength. In the study of gapped topological phases, such as spin liquid or FQH states, Lieb-Schultz-Mattis (LSM)-type theorems (Lieb et al., 2004, 1961) have played an important role in theoretical developments. Basically, such theorems state that one can constrain possible macroscopic physics based on microscopic information, such as symmetries or degrees of freedom per unit cell. For gapless systems, an example is the so-called Luttinger's theorem (Luttinger, 1960) for the volume enclosed by the Fermi surface. It was shown to be of topological origin by Oshikawa (Oshikawa, 2000). This has important repercussions for Kondo lattice systems and possibly the pseudogap state of the cuprates, where in both cases Fermi surface volume differs from simple expectations based on weakly interacting electrons. It might be useful if we can come up with something similar for generic gapless phases of matter, i.e. , "liquid" phases.

Additionally, bounds for certain observable quantities can be deduced analytically from rather general considerations, which makes them also applicable to correlated systems. The examples include the aforementioned bounds on diffusivity and resistivity, which are based on a coarse-grained hydrodynamic description (Hartman et al., 2017; Lucas and Hartnoll, 2017b) or the quantum mechanical "Lieb-Robinson bound" (Mousatov and Hartnoll, 2020). Energetics of correlated systems may be also better understood using exact relations between the potential and kinetic energy derived from the virial theorem (Leggett, 1998; Levallois et al., 2016b). Studying the consequences of general quantum-mechanical relations/theorems for strongly correlated systems and applying the resulting statements to the analysis of experiments may expand our view of the correlated electron problem and potentially lead to new and useful phenomenologies.

Difficulties regarding strong correlations are shared with other branches of physics as well. One possible way to tackle these is through the development of dual theories between a strongly correlated limit and a weakly correlated regime by breaking down the problem to summation of multiple "less complicated" problems. A successful example in high-energy physics is holographic theories such as anti-de Sitter Space/conformal field theory, which has garnered attention in the condensed matter context (Hartnoll et al., 2018). Within condensed matter physics, there have been many attempts to map interacting systems to single-particle physics, among which bosonization in 1D is probably one of the most successful. Most of the existing approaches, however, have limitations. Further development and generalization of the existing methods to higher dimensions and/or more general correlated systems along the line of some of the ongoing works such as higher-D bosonization (Castro Neto and Fradkin, 1994; Houghton and Marston, 1993; Kopietz, 2008; Luther, 1979; Lüscher, 1989), or patch theories (Polchinski, 1994), is therefore highly desirable. Ideally, as in the case of a Fermi liquid, a mapping between a strongly correlated phase to a simple one would enable one to learn physics of the "difficult" regime from "easy" regime.

In a related fashion, "analogue theories" are a research program which investigates analogues of a particular field of physics within other physical systems, with the aim of gaining new insights into the corresponding problems. For example, the utilization of analogue theories of gravity and cosmology in various low-energy fields such as ultracold atoms, acoustic and condensed matter systems have lead to many fruitful results (Barceló et al., 2011). As a result, they have motivated numerous interesting experimental setups which simulate puzzling problems in gravity such as black holes. However, "condensed matter analogue theories" are much less explored (Argüello-Luengo et al., 2019; Ghorashi et al., 2020; Gruzberg et al., 2017; Lewenstein et al., 2007). Aside from applications of adS/CFT ideas (Hartnoll et al., 2018), particular examples, are recent attempts to map the problem of disordered fermions to gravitational theories with ambitious perspective to use the developments of quantum gravity to gain further understandings in these systems (Ghorashi et al., 2020; Gruzberg et al., 2017). Another recent example, of both analytical and computational importance, is a development made by a group of mathematicians which proposed an interesting method for finding eigenvalues of large random matrices without solving eigenvalue problems (Arnold et al., 2019; Beenakker, 2019), which develops the idea of "localization landscape" proposed in (Filoche and Mayboroda, 2012). Therefore, considering the tremendous developments of sophisticated analytical and mathematical methods and techniques which are developed in other fields such as high-energy physics and/or mathematical physics, it makes the exploration via "analogue correlated electronic models" in these fields a desirable goal. They could provide an improved toolbox to tackle correlated systems. More-

over, they may also guide further simulation of correlated electronic phases in other physical systems.

### 2. Non-equilibrium

Finally, let us discuss an important avenue, where analytical methods do have a certain advantage. Investigating non-equilibrium phenomena in an already quite complicated correlated electronic system requires further developments of an analytical toolbox. One particularly important direction is periodically driven many-body systems. It has been found theoretically that there exist long time scales in which interacting periodically driven quantum many-body systems or Floquet systems can be described by an effective time-independent theory (Abanin *et al.*, 2017, 2015; Canovi *et al.*, 2016; Else *et al.*, 2017; Machado *et al.*, 2019; Mori *et al.*, 2016; Weidinger and Knap, 2017). Since this makes it possible to use existing equilibrium techniques to understand strongly correlated Floquet systems, a lot of effort has been spent on deriving effective time-independent Hamiltonians that allow such a description (Abanin *et al.*, 2017; Bukov *et al.*, 2016; Eckardt and Anisimovas, 2015; Goldman and Dalibard, 2014; Itin and Katsnelson, 2015; Maricq, 1982; Mikami *et al.*, 2016; Mohan *et al.*, 2016; Rahav *et al.*, 2003; Schweizer *et al.*, 2019). While much progress has been made in the case of non-interacting systems, progress has been slower in the strongly correlated case (with some results obtained using Keldysh formalism (Kandelaki and Rudner, 2018)). Most current work focuses on the particular limit of high frequencies. A variety of methods called Floquet Magnus-, van Vleck- or Brillouin-Wigner-type expansions (Mikami *et al.*, 2016; Mohan *et al.*, 2016) have been developed for this regime. However, there has not been much progress for more generic, interesting and experimentally relevant mid- to lower frequency regimes (Vogl *et al.*, 2020).

Progress has been limited for two reasons. First, generic interacting systems have an algebra that does not "close", which has stymied progress because it leads to complicated effective Hamiltonians. This means the following. High frequency expansions include a set of nested commutators of the Hamiltonian at different times $[H(t_1), [H(t_2), ...]]$. These generate higher and higher order interaction terms as higher corrections are included and can become quickly uncontrolled – this is called operator spreading. If new terms keep being generated ad infinitum we say that the algebra is not closed. Second, for the interacting case, high frequency expansion are at best to be considered asymptotic expansions rather than convergent expansions. This means higher order corrections might improve predictability in the high frequency regime but do not extend the regime to lower frequencies (Abanin *et al.*, 2017). Recent work (Vogl *et al.*, 2019) has circumvented part of this problem by using an renormalization group-flow like approach to partially re-sum one of the high frequency expansions. However,

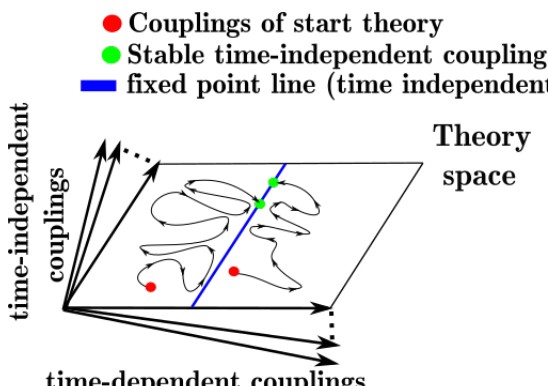

FIG. 13 The graphic shows schematically how couplings in a time-dependent theory flow in the approach of (Vogl *et al.*, 2019). One finds that as the couplings flow they repeatedly approach a line of fixed points, which ultimately turns out to be unstable until eventually a stable fixed point is reached.

even for this case additional work is needed to improve the method. We anticipate that the flow equation approach (Vogl *et al.*, 2019) can be improved by a better choice of generator for the underlying unitary transformations. Let us motivate this idea.

Recent years has seen the development of another similar approach for interacting time independent Hamiltonians – the so-called Wegner flow approach (Wegner, 1994). What this approach does is dynamically construct a unitary transformation that diagonalizes an interacting Hamiltonian in an effective non-interacting basis (Kehrein, 2007). An effective Hamiltonian flows until a non-interacting Hamiltonian is reached at a fixed point. This method also suffers from the issue of operator-spreading - before a fixed point is reached many interaction terms are generated along the flow. However, with a clever choice of unitary transformations the issue can be avoided (Mielke, 1998). In the Floquet case the fixed points of the flow equations are time independent Hamiltonians. Along the flow one also suffers from operator spreading. However, it is found that the source of this spreading is that there are many time-independent unstable fixed points that are approached closely as sketched in Fig. 13. It might be possible to stabilize these fixed points by a better choice of generator. Finding such a generator could allow reaching lower frequency regimes with less interaction terms being generated. Success in this regard will lead to progress in understanding of out-of equilibrium strongly correlated systems.

### D. Novel Spectroscopic Approaches

A key challenge of the correlated electron problem is that the electron momentum **k** and momentum transfer **q** between electrons in many cases cease to be good variables to describe systems across a wide range of length, energy, and time scales. If interactions are strong enough, umklapp processes prevent even Bloch states, let alone

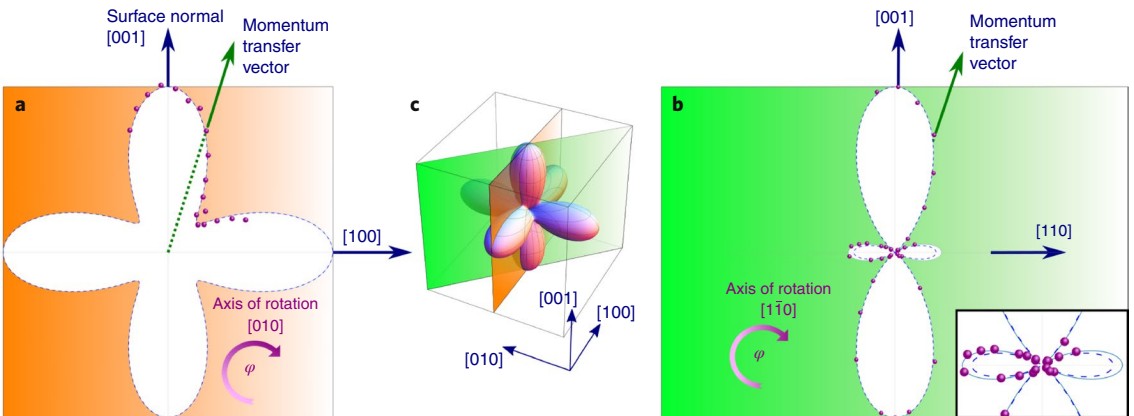

FIG. 14 Integrated intensities of $M_1$ ($3s \to 3d$) X-ray edge spectra plotted on the projections of the orbital shape of the ${}^3A_2$ ${}^3A_2 3d(x^2 - y^2) 3d(3z^2 - r^2)$ hole density. Here the projections of the 3D orbital shape on two planes are defined by [001] and [100] (a) and [001] and [110] (b). Note that the $3d(x^2 - y^2)$ contribution vanishes in the [001]–[110] plane and so only the $3d(3z^2 - r^2)$ may be seen. From Ref. (Yavaş *et al.*, 2019).

free-particle states, from forming a good approximate basis to approach the problem. Moreover, crystal imperfections and disorders on the atomic level often affect electronic properties at much longer wavelengths and even macroscopically (Cao *et al.*, 2017; Mesaros *et al.*, 2011). Mesoscopically, the competition between different nearly degenerate broken symmetry states can result in short-ranged and anisotropic correlations. In other cases, correlated electron systems are characterized by patterns of order (hidden-order, non-quasiparticle transport, topological effects, fractionalization) that we have only imperfect tools to characterize. It is imperative to develop new spectroscopic tools to address the above aspects.

### 1. State-of-the-art spectroscopies

Addressing these challenge calls for a holistic and concerted effort — not only do we need experimental methods that probe the relevant degrees of freedom (charge, orbital, spin and lattice) in the form of both single-particle spectral and two-particle (and higher) correlation functions, but also ones that combine these probes in multimodal approaches to reveal the cooperation and competition between orders. This requires both utilizing existing experimental tools by pushing their capabilities and resolutions as well as developing new experimental methods.

Over the last few years, multimodal experimental studies combining complementary probes have revealed insights in a wide range of materials (Comin *et al.*, 2014; Gerber *et al.*, 2017; da Silva Neto *et al.*, 2014; Zong *et al.*, 2019) that cannot be obtained otherwise. In such studies one hopes that identical sample condition across probes enables more reliable comparison. In cases such as *in-situ* ARPES/STM studies of epitaxial films grown by MBE, this approach is essential due to inherent limitations of the experimental probes and air sensitivity

of their surfaces. Part of this effort requires extending the regions of phase space (temperature, pressure, magnetic fields) over which the combined techniques overlap. Additionally, these efforts encompass the integration of novel tuning parameters with existing techniques. One such example is the integration of uniaxial strain tuning to spectroscopy methods such as ARPES, photon scattering, STM and NMR (Andrade *et al.*, 2018; Kim *et al.*, 2018a; Kissikov *et al.*, 2018; Pfau *et al.*, 2019). Techniques that can probe and disentangle contributions from multiple degrees of freedom constitute an important part of the multimodal approach. One such example is resonant elastic and inelastic X-ray scattering, which is sensitive to lattice, charge and spin degrees of freedom and has provided a considerable impact in the study of cuprates and iridates (Kim *et al.*, 2012). RIXS has now demonstrated sub-30 meV energy resolution at the Cu $L_3$ edge, and sub-10 meV resolution at the Ir $L_3$ edge (Kim *et al.*, 2018b). We also believe that there will continue to be exciting developments in momentum resolved electron energy loss spectroscopy that can measure the frequency- and wave-vector-dependent density-density correlation function (Vig *et al.*, 2017) and inelastic neutron scattering.

There are also a number of new techniques that can give previously inaccessible information. Recently, *s*-orbital non-resonant inelastic X-ray scattering using modern synchrotron facilities with high brilliance allows the direct resolution of the orbital occupation (Leedahl *et al.*, 2019; Yavaş *et al.*, 2019). This is an improvement over typical methods of deducing wavefunctions from optical, X-ray and neutron spectroscopy methods in which spectra must be analyzed and interpreted using modeling of spectroscopic information, for example through crystal field excitations (Zhang *et al.*, 2014). Shown in Fig. 14 is the quadrupolar scattering intensity as a function of the momentum transfer direction in the canonical Mott

insulator NiO. It directly shows the three-dimensional (3D) orbital hole density of the Ni high-spin $3d^8$ configuration in an octahedral coordination, namely the $^3A_2$ $3d(x^2-y^2)3d(3z^2-r^2)$. As the $3d(x^2-y^2)$ contribution vanishes in the [001]–[110] plane the small lobes of the $3d(3z^2-r^2)$ contribution remain. This technique can also be used for itinerant systems, and may be invaluable to determine the local orbital in systems where both band formation and electron correlations are important, for example in the entangled spin–orbit states in ruthenium and iridium materials. It could be interesting to apply it to systems with rare earths where the ground state is often composed of an admixture of complex $4f$ orbitals.

### 2. Prospects for future developments

Most spectroscopies focus on the spectral function of quasiparticle excitations or two-particle correlation functions in the limit of linear response. Correlated electron systems may host symmetry protected or symmetry broken phases that do not manifest directly in these spectral and correlation functions (Morimoto and Nagaosa, 2016; Zhao et al., 2017, 2018a). In this regard, we believe it will be essential to move beyond the conventional confines of linear response techniques to gain insights about strong correlation effects. This calls for spectroscopies that explicitly probe higher order susceptibilities $(\chi^{(2)}, \chi^{(3)}\ldots)$ across a range of energies and wave-vectors. Key examples include 2D coherent spectroscopy (THz to IR) as a probe of fractionalized excitations in quantum spin liquids (Choi et al., 2020; Lu et al., 2017; Wan and Armitage, 2019). THz emission spectroscopy and optical second harmonic generation (SHG) can be used to probe subtle symmetry-broken and hidden order phases of matter (Fiebig et al., 2005; Zhao et al., 2017, 2018a). In addition to its role in probing symmetries, it has also been proposed that nonlinear response is sensitive to Berry's phase effects in non-interacting systems (Morimoto and Nagaosa, 2016; Sipe and Ghahramani, 1993; Virk and Sipe, 2011). It will be interesting to see if such physics can be extended to interacting systems. One important consideration in designing or conceiving new spectroscopic techniques, is that they should measure a well-defined response function. Many ultrafast pump-probe experiments using typical 800 nm light (which corresponds to 1.55 eV), which have been applied to correlated systems are interpreted as measuring some general relaxation time without clear perspective of what exactly is relaxing or how. This has limited the impact of such experiments.

A number of correlated systems show the phenomenon of "hidden order" (HO) e.g. they may exhibit a clear sign of a phase transition in thermodynamic quantities like heat capacity or signs of a gap developing in spectroscopy, but conventional probes of symmetry breaking give little information on the nature of the ordered state. Most fa-

mously, $URu_2Si_2$ shows a large peak in specific heat at $T_o = 17.5$ K, which indicates a classic second-order phase transition (Bourdarot et al., 2005; Tripathi et al., 2007; Villaume et al., 2008). Although intensive theoretical and experimental studies have been performed, the order parameter of the state below $T_o$ is still undetermined. For instance, it is difficult to reconcile the small size (if any) of the ordered moment ($< 0.03\ \mu_B$) with the large jump of $\Delta C/T_o = 0.3$ J/mol K$^2$. It is well established that at low-pressures the HO phase is not simple antiferromagnetism, although there is a transition at a pressure of 0.5 Pa (Fig. 15) through a first-order transition to a long-range antiferromagnetic state (Villaume et al., 2008). Theories ranging from orbital currents, singlet-triplet $d$-density wave, hexadecapolar, antiferro-quadrupolar, to "hastatic" order have been proposed (Mydosh and Oppeneer, 2014). The interest in $URu_2Si_2$ is reinforced by the appearance of unconventional superconductivity at $T_{sc} = 1.2$ K under ambient pressure, which disappears at 0.5 GPa. It is likely that the issue has withstood thirty years of investigation because we do not have the experimental tools that easily couple to the static order or the elementary excitations of this broken symmetry state. For instance, some of the proposed orders are expected to have unconventional excitations with selection rules not easily accessible by conventional electric and magnetic dipole excitations in linear optical response. In some cases, the excitations can be revealed, but it takes a detailed analysis. Recently a combination of information from Raman and neutron scattering has been used to understand the nature of the broken symmetries in $URu_2Si_2$ (Buhot et al., 2014; Kung et al., 2015). A sharp excitation of 1.7 meV with $A_{2g}$ symmetry in the Raman response shows that vertical and diagonal reflection symmetries are broken at the uranium sites. The appearance of the same excitation in neutron scattering at (001) (corresponding to the inverse the $c$-axis lattice constant) requires the hidden order to be staggered alternating along the $c$ direction. Such order with alternating left- and right-handed states at the uranium sites has no modulation of charge or spin and is hidden to all probes at the zone center except for scatterings of $A_{2g}$ symmetry. Further development of nonlinear optical techniques that evade the conventional selection rules regarding the linear response of electric and magnetic dipole excitations or enhance the cross-section of unconventional excitation may prove useful to further reveal the nature of these or other hidden ordered states.

It is possible that hidden-order states are very common and give some of the confusing phenomenology of other correlated materials. For instance, signatures of broken symmetry also exist for the "pseudogap" in the cuprates (He et al., 2011; Xia et al., 2008; Zhao et al., 2017). However, the precise nature of the ordered phases remains unresolved despite intense experimental and theoretical efforts. Some candidates like the $\mathbf{q} = 0$ and $\mathbf{q} = (\pi, \pi)$ orbital current orders are challenging to verify with conventional scattering technique (Bourges and

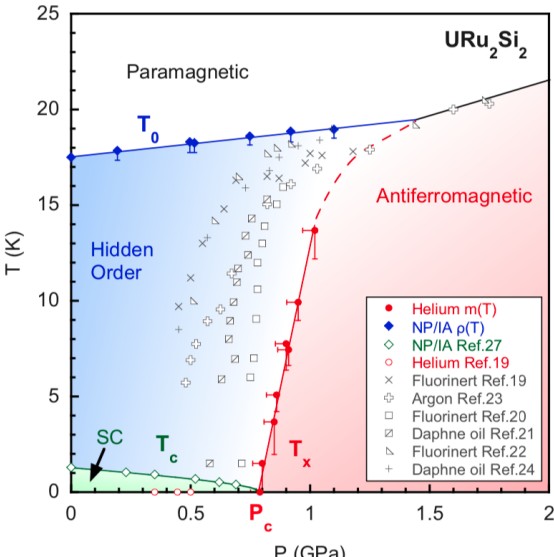

FIG. 15 $(T,P)$ phase diagram of $URu_2Si_2$ from resistivity and neutron scattering in the low-pressure hidden-order phase and the high-pressure antiferromagnetic phase. $T_0$ is defined by the local minimum in the resistivity data; the error bar represents $\Delta T_0$. $T_x$ is defined where the neutron ordered moment reaches half of its full value, the vertical error bar 90%. The horizontal error bar represents a 5% uncertainty in pressure. The superconductivity is suppressed at $P_c$ where antiferromagnetism appears. A comparison to other published data shows that the value of the AFM critical pressure $P_c$ is substantially higher under the hydrostatic conditions of Ref. (Butch et al., 2010) than many previous experiments. References correspond to those given in Ref. (Butch et al., 2010).

Sidis, 2011; Chakravarty et al., 2001; Croft et al., 2017; Huang et al., 2012; Varma, 1997) and new techniques that give information on their broken symmetries and possible unconventional excitations will be useful.

In a related fashion, future promising directions include tools that involve using and/or measuring pairs of particles (e.g. entangled neutrons, photons, electrons) to extract response functions that are inaccessible to "conventional" spectroscopies. One exciting direction is the implementation of "coincidence" experiments, such as Auger-photoelectron coincidence spectroscopy (APECS) (Haak et al., 1978; Stefani et al., 2002), which should be revisited with the advent of improved photoemission detection technology. As mentioned above, the principal issue with such experiments is their relatively poor resolution, mainly limiting the method to getting information on the correlation hole around an electron (Schumann et al., 2007). However if this could be overcome, in addition to allowing some related recent proposals (Stahl and Eckstein, 2019), such experiments would probe particle-particle correlations (Su and Zhang, 2020) at a finite momentum or a given time, in contrast to most other two particle probes that probe particle-hole correlations. Here one can imagine, for instance, prob-

ing Cooper pair correlations by looking at coincidence in $\pm\mathbf{k}$ emitted electrons. This could give direct access to the anomalous self energy of the pairing interaction (off-diagonal term), instead of via an indirect entry into the diagonal terms in the $2 \times 2$ Green's function matrix in the Nambu-Gorkov representation.

As an essential property of quantum systems, probing long-range entanglement would be very powerful, but may pose an even bigger challenge. One example is using two neutrons prepared in an entangled state (Shen et al., 2020b) to scatter off different areas of a possible spin liquid. Under such conditions, probing the final state of the neutron pair one might be able to obtain the entanglement information of the spins in the material. This type of experiment highlights a future direction in which spectroscopic measurements, in this case beyond conventional neutron scattering, may be able to probe long-range entanglement in strongly correlated systems. Entanglement entropy has been measured using ultracold bosonic atoms in optical lattices where identical copies of a many-body state are prepared and then interfered (Islam et al., 2015). Solid-state systems have the obvious problem in this regard that they cannot generally be easily partitioned and interfered. There have been proposals for how to measure entanglement in solid-state systems (Laflorencie, 2016), but they have been mostly limited to measuring systems with a globally conserved quantity, for instance the particle number for Fermi gases or the subsystem magnetization for quantum magnets and are thus far from general. Klich and Levitov proposed that quantum noise in a quantum point contact can be used as an entanglement meter when driven by a periodic pulse train (Klich and Levitov, 2009). In a related fashion, Song et al. (Song et al., 2012) propose that the noise spectrum can be a probe of entanglement in a $O(2)$ quantum magnet that has a magnetic field partially obscured by a superconducting shield.

### E. Local probes

Many strongly correlated systems exhibit a multitude of nearly degenerate phases that either compete or coexist locally even in clean systems. Moreover, given the degree of inhomogeneity inherent to many correlated electron systems (Hamidian et al., 2016; Liang and Wang, 2018; Zhang et al., 2016), local probes provide crucial tools for the identification and isolated measurement of differing local states or environments. Furthermore, in the presence of nearly degenerate states, or states that either compete or coexist, it is generally difficult to understand the behavior of the whole system as a simple composition of the microscopic parts. Spatially-resolved measurements are therefore crucial for probing individual phases not only in isolation but also for understanding how their interactions contribute to macroscopic behavior (see Fig. 17). This idea is illustrated by nanoscale imaging experiments in the colossal mag-

netoresistive manganites as shown in Fig. 17a. In these materials, microscopic competition between the metallic ferromagnetic and insulating antiferromagnetic phases determine the macroscopic transport properties. Beyond the phases themselves, there further exist many intriguing questions to explore regarding the boundaries or walls between such domains which similarly require local visualization.

In Fig. 16, we compile several microscopic methods, the degrees of freedom to which they are sensitive, and their spatial resolutions. Some of these microscopies are relatively well-established, *e.g.* transmission electron microscopy (TEM), and scanning tunneling microscopy (STM), but have been utilized recently in cutting-edge and previously unexpected ways (El Baggari *et al.*, 2018; Enayat *et al.*, 2014; Hamidian *et al.*, 2016; Mundy *et al.*, 2014). Efforts to probe local spins through spin-resolved tunneling and superconducting pairs with superconducting tips have been demonstrated (Enayat *et al.*, 2014; Hamidian *et al.*, 2016) in STM. Recent advances in high-resolution scanning TEM (STEM) enabled in part by the development of new imaging detectors and data processing techniques have expanded the accessible phase space for atomic resolution real-space imaging across a much wider range of temperatures and other *in-situ* conditions (Coll *et al.*, 2019).

New microscopy methods have also been developed in recent years, showing promising sensitivity and spatial resolution despite their infancy. Novel developments in the design and fabrication of nano SQUID devices on a tip has enabled magnetic imaging with single spin sensitivity and 10s of nm spatial resolution (Ceccarelli *et al.*, 2019; Vasyukov *et al.*, 2013). Diamond nitrogen vacancy (NV) microscopy has recently demonstrated room-temperature field sensitivities as high as $0.9 \times 10^{-12}$ $T/Hz^{1/2}$ (Wolf *et al.*, 2015) with a spatial resolution tens of nanometers and below depending on the microscope design, NV center-to-sample distance, etc. (Levine *et al.*, 2019; Tetienne *et al.*, 2014). Utilizing the coherence of X-rays at existing and upcoming diffraction-limited synchrotron facilities, nano X-ray diffraction and coherent X-ray imaging have demonstrated up to $10^{-6}$ strain sensitivity and spatial resolution as high as 1 nm in metals and semiconductors (Hruszkewycz *et al.*, 2017; Pfeiffer, 2018; Robinson and Harder, 2009), and are expected to be applied to the study of lattice and electronic orders in correlated materials (Assefa *et al.*, 2020; Cao *et al.*, 2020a; Chen *et al.*, 2016a; Robinson *et al.*, 2020).

It should be noted that a host of microscopy approaches have proven to be or will become powerful spectroscopic tools, as in the case of STM and STEM. Few-meV energy resolution and $\sim$Å spatial resolution have both been demonstrated by electron energy loss spectroscopy (EELS) in the STEM (Muller *et al.*, 2008). Recent experiments have also begun to probe $\mathbf{q}$ on still small spatial scales with momentum-resolved EELS mapping the dispersion curves in graphene nanostructures (Senga *et al.*, 2019). As spectroscopic resolution improves in spatially localized probes, spatial localization is similarly improving for many spectroscopic "gold standards". With tightly focused laser pulses, second harmonic generation (SHG) and magnetic optical Kerr effect (MOKE) could deliver single-digit micron spatial resolution. Micro- and nano- ARPES have recently been realized at synchrotron user facilities across the world (Cattelan and Fox, 2018). Meanwhile, next generation time-of-flight photoemission "momentum microscopes" are also becoming commercially available, enabling simultaneous 2D data collection either in electron momentum or real space (Tusche *et al.*, 2015). For more discussion, see Section III.D.

While many well-established techniques routinely probe down to the atomic scale (including STM, AFM, STEM, EELS), physical constraints of the advanced instrumentation required for these techniques often limits their application to specific sample conditions or geometries which in many cases do not extend to the phases of interest for condensed matter systems. For example, electron energy loss spectroscopy (EELS) can be used to probe core electronic structure down to the atomic scale, enabling the direct measurement and visualization of charge at polar interface(Mundy *et al.*, 2014) (Fig. 16c). Compared to other core spectroscopy techniques such as X-ray absorption spectroscopy, however, the stability requirements and subsequent signal limitations of such high resolution EELS experiments are, as yet, generally limited to ambient conditions under vacuum, precluding the detailed study of how such states evolve under temperature, pressure, or other applied stimuli. Thorough exploration of competing and coexisting states in many correlated systems will require improved flexibility of existing imaging experiments as well as the development of new imaging techniques in order to probe local phenomena across a wide range of conditions and systems. A number of potentially important areas for future work include:

1. Sample environments and operation protocols satisfying the unique needs of correlated materials can be further developed. Because interesting electronic properties often emerge at temperatures substantially lower than room temperature, ongoing efforts for realizing stable and compact sample environments are underway for electron, coherent X-ray, and force microscopies (Assefa *et al.*, 2020; El Baggari *et al.*, 2018; Robinson *et al.*, 2020). The additional integration of other *in situ* conditions such as pressure, strain, external fields, etc. will expand these types of characterization to unexplored regions of phase space. The successful integration of such environments into modern microscopes will require innovative and inspired engineering given the space limitations and stability requirements of these techniques.

2. Inputs from theoretical modelling are essential for fully understanding the experimental measurements and for connecting experimental observations with theoretical calculations of electron re-

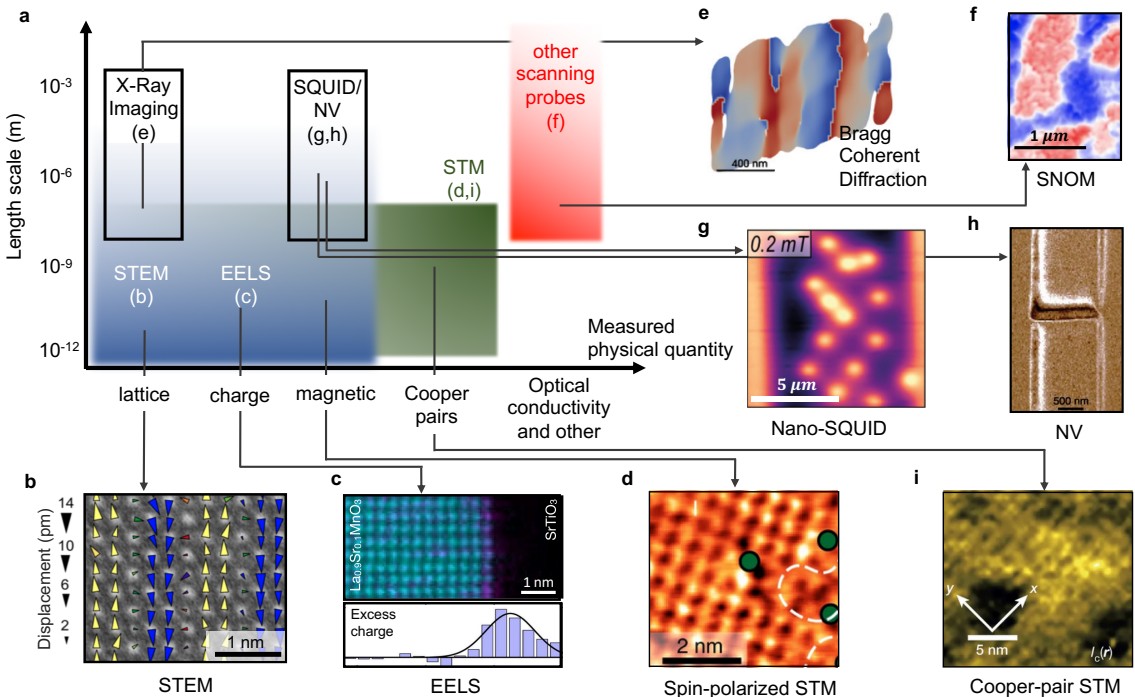

FIG. 16 Local probes with different sensitivities and spatial resolution. (a) Survey of spatially resolved probes with access to information at the picometer to millimeter length scales. (b) STEM imaging and mapping of picometer lattice displacements in charge-ordered phases. From Ref. (El Baggari *et al.*, 2018). (c) Spatially resolved EELS of valence and charge in an oxide interface. From Ref. (Mundy *et al.*, 2014). (d) Spin-polarized STM detection of magnetic moments at the atomic scale. From Ref. (Enayat *et al.*, 2014). (e) Coherent Bragg X-ray imaging of structural and charge order domains. From Ref. (Assefa *et al.*, 2020). (f) SNOM imaging of coexisting metallic and insulating domains. From Ref. (McLeod *et al.*, 2017). (g) Advanced SQUID microscopy with sub-micron resolution. From Ref. (Ceccarelli *et al.*, 2019). (h) NV imaging with high sensitivity to spins. From Ref. (Tetienne *et al.*, 2014). (i) STM with a superconducting tip enables Cooper pair tunneling and nanoscale imaging of the superconducting condensate. From Ref. (Hamidian *et al.*, 2016).

sponse and correlation functions. Ultimately, most microscopy measurements provide some sort of contrast which is only physically meaningful if the contrast mechanism can be identified or understood. One example involves scanning near-field optical microscopy (SNOM) (Atkin *et al.*, 2012; McLeod *et al.*, 2017, 2020). Simulations of the electromagnetic field distribution around the tip and the sample were crucial in revealing the sensitivity to the plasmon oscillations in graphene (Fei *et al.*, 2012).

3. We encourage establishing a consortium, or a collaboration mechanism for multimodal explorations of correlated materials at the same spatial location under the same condition. This will require developing (1) fiducials that could be used across microscopies, and (2) measurement protocols where air-sensitive, *in situ* experiments precede *ex situ* and/or potentially destructive ones. Similar standard operating procedures are already common in other fields, for example the biological science cryo-EM community.

4. Microscopy studies provide a natural playground for the application of machine learning. The advent of pixelated area detectors across many modern microscopic methods in the last decade fuel the acceleration of data generation. For example, modern coherent X-ray imaging can generate sub-terabytes of data within 24 hours. Its generation, transfer, and storage will require new data infrastructures and management plans not only for user facilities but also individual research labs in the foreseeable future. Moreover, identifying key features in the image, streamlining the data analysis and cross-comparing different microscopic studies will benefit from different approaches of artificial intelligence (Cherukara *et al.*, 2018; Laanait *et al.*, 2019). Machine learning has been recently applied to STM (Cheung *et al.*, 2020).

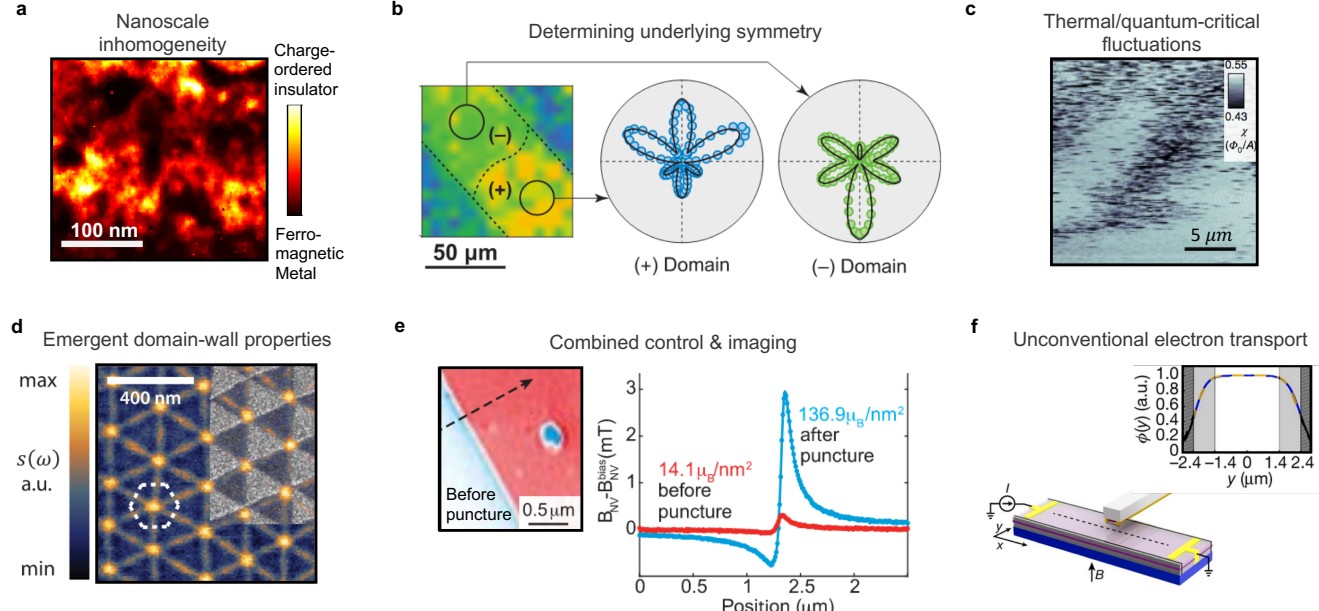

FIG. 17 Example problems addressed with local measurements. (a) Nanoscale phase coexistence in the $(La,Pr,Ca)MnO_3$ manganite. The competition between charge-ordered insulating patches (bright) and ferromagnetic metallic regions (dark) is visualized using dark-field transmission electron microscopy. The contrast reflects the amplitude of the charge order superlattice. Image adapted from (Uehara *et al.*, 1999). (b) Endowing bulk probes such as SHG with spatial resolution can help disentangle the underlying symmetry from its sample-averaged counterpart, especially in the presence of domains or competing states. Such capability revealed parity domains in the parity-breaking electronic nematic metal $Cd_2Re_2O_7$ (Harter *et al.*, 2017). The 2D false-color map reflects the second harmonic intensity; the polar plots were measured by rotating light polarizations. (c) Scanning SQUID imaging of the diamagnetic susceptibility reveals quantum fluctuations in the disordered superconductor NbTiN at mesoscopic scale (Kremen *et al.*, 2018). (d) Domain walls often carry exotic properties distinct from the bulk, due to the suppression of a particular order parameter or local change in symmetry. IR nano-imaging, for instance, measures enhanced optical conductivity due to plasmons localized at the domain walls in TBG (Sunku *et al.*, 2018). (e) Local magnetization in the layered magnetic material $CrI_3$ measured using a NV magnetometer. By adding an *in situ* mechanical stimulus, NV imaging further reveals a local enhancement of the magnetization coupled to structural degrees of freedom (Thiel *et al.*, 2019). (f) Visualizing electronic transport with enhanced spatial resolution is a promising approach for understanding exotic phenomena such as electron hydrodynamics, strange metals and topological edge modes. In Ref. (Sulpizio *et al.*, 2019), a scanning carbon nanotube single-electron transistor, which is sensitive to the potential of flowing electrons, reveals Poiseuille flow in high-mobility graphene devices (see also Ref. (Ku *et al.*, 2020)).

## F. Spectroscopies and microscopies out of equilibrium

Nonequilibrium spectroscopies provide a new avenue to disentangle different degrees of freedom, and enable the study of collective excitations, metastable and transient states, and fluctuations (Fig. 18). Many of the previous time-resolved nonequilibrium studies on strongly correlated materials, such as time-resolved reflectivity and time-resolved photoemission spectroscopy, have heavily utilized photoexcitations at energies around $\sim 1.5$ eV (Demsar *et al.*, 1999; Gedik *et al.*, 2005; Yang *et al.*, 2015). This is largely attributed to the commercial development and widespread adoption of 800-nm Ti:sapphire femtosecond lasers. Their photon energy is, however, orders of magnitude larger than the relevant energy scales for collective excitations in correlated material systems. One important future direction is to develop pumps with photon energies targeted in resonance with underlying low-energy excitations, such as

phonons, magnons, and other emergent particles. This has been attempted on limited basis in correlated materials: using mid-infrared pumping in resonance with a vibration mode of cuprate superconductors to induce possible nonequilibrium superconductivity (Först *et al.*, 2011); using targeted pumping to establish transient metastable ferroelectric states (Li *et al.*, 2019b; Nova *et al.*, 2019); and using orbital excitations across the Mott gap to observe the evolution of spin waves in iridates (Dean *et al.*, 2016). The development of a continuously tunable, $> 100$ kHz repetition rate, $> 1$ $\mu$J pulse energy THz to mid-infrared sources, such as the one at Helmholtz-Zentrum Dresden-Rossendorf (Green *et al.*, 2016), will enable future time-resolved spectroscopies to accommodate a wide range of materials with characteristic low-energy excitations in the range of 1–200 meV. Recent effort to achieve intense mid-infrared pulses with variable duration (ps to ns) would further enable sustained optical driving and stabilize transient states (Budden *et al.*,

2020) (Fig. 18a).

In contrast to the THz/mid-infrared sources mentioned above, sub-fs pulses coming from the high-harmonic generation (HHG) process represent the high end of the spectrum. It is not immediately obvious that the sub-fs and the 10s–100s eV scales are relevant for most processes in solids, but HHG-based techniques offer two important pieces of information in the time domain. First, element specificity through XUV/X-ray absorption spectroscopy is afforded through table-top or free-electron laser-based absorption spectroscopies that have been used to track element-specific evolution of local bonding, magnetization, and lattice structure (Geneaux *et al.*, 2019; Jager *et al.*, 2017) (see Fig. 18c). Without the constraint of the uncertainty principle, both time and energy resolutions can be optimal (as to fs in time and $\sim 10$ meV in energy). Second, phase sensitivity through a holographic detection for photoelectrons is allowed by leveraging the interference between different quantum paths of photoemission among successive harmonics in an HHG pulse train. The phase of a complex wave in atomic orbitals has been recently imaged (Huismans *et al.*, 2011; Villeneuve *et al.*, 2017). It is our hope that a similar holographic detection can be applied to materials, which may enable phase-sensitive photoelectron spectroscopy and, for instance, allow the investigation into the sign of the superconducting gap in correlated superconductors.

We envision that novel nonequilibrium spectroscopies will facilitate addressing some specific pertinent issues that are discussed above as well as driving new phenomena. For instance, theoretical calculations have predicted that circularly polarized light can provide a knob to break time-reversal symmetry and drive frustrated Mott insulators into a chiral spin liquid (Claassen *et al.*, 2017b; Quito and Flint, 2020). This can be realized using spectroscopies such as mid-infrared pumped time-resolved second harmonic generation. In a Kondo breakdown QCP, electron lifetime diverges following a multiscale temperature scaling law, which was proposed to be addressed by time-resolved optical reflectivity and photoemission spectroscopies (Paul *et al.*, 2008). The challenge here is to have sufficient sensitivity when approaching a zero-excitation limit to minimize transient heating. In correlated superconductors, the amplitude mode of superconductivity, which is often termed the condensed matter analogue of the Higgs boson, has been reported by THz pump-optical probe reflectivity measurements (Katsumi *et al.*, 2018). The pump photon energy has to be smaller than twice the superconducting gap (or the antinodal gap in the case of a $d$-wave superconductor). Future developments which combine THz pumping and other spectroscopic probes such as photoemission or scanning tunneling spectroscopy can further detail the momentum- and spatial dependence of such collective modes. Notably, the collective mode is a direct manifestation of the order parameter (Baldini *et al.*, 2020). Last but not least, studies of order parameters by driving competing phases may provide insight into the phase competition at their funda-

mental interaction timescale (Kogar *et al.*, 2020; Wandel *et al.*, 2020) and may help find the roots of node-antinode dichotomy in cuprate superconductors (Hashimoto *et al.*, 2014).

In addition, time-resolved STM (Fig. 18b), time-resolved scanning SQUID (Cui *et al.*, 2017), time-resolved neutron diffraction, and time-resolved RIXS are tangible future directions for nonequilibrium spectroscopies (Cao *et al.*, 2019; Dean *et al.*, 2016). The pump excitation here can be a pulsed electric or magnetic field which facilitates transitions between different correlated states. Near-field imaging techniques in combination with femtosecond laser excitations may enable the imaging of hidden ordered phases in correlated materials (McLeod *et al.*, 2020). A further extension of time-resolved near-field imaging is to employ second harmonics as a pathway to reveal symmetry-breaking phases (Neacsu *et al.*, 2009). In such novel nonequilibrium imaging techniques, a method to acquire a broad field-of-view image in a single shot can be a paradigm-shifting development in studying spatially inhomogeneous correlated phases (Liang and Wang, 2018; Zhang *et al.*, 2016).

## G. Experimental probes in extreme environments

Probing correlated electron phenomena and accessing energy scales relevant to underlying interactions often require extreme experimental environments, such as ultra low temperatures, high pressure, high magnetic fields, and high electric fields. Extreme environmental conditions can induce new correlated states or may be used to probe the energy scale of correlations involved in an existing phase. In Table I, we list the present limits in extreme environments discussed below.

### 1. Ultralow temperature

Ultralow temperature allows for the observation of new ground states and quantum effects that may be masked or destroyed by thermal excitations. Experiments in this extreme limit have led to notable discoveries, most recently establishing superconductivity near a magnetic QCP as a relatively universal phenomenon (Schuberth *et al.*, 2016), demonstrating a superconducting phase at extremely low electron density in crystalline bismuth (Prakash *et al.*, 2017), or finding delicate phases in the 2D electron gasses under high field and very low temperature (*i.e.*, even-denominator fractional quantum Hall phases (Willett *et al.*, 1987) are found in some cases at temperatures only as low as 5 mK (Xia *et al.*, 2004)). Advancing experimental capabilities at ultralow temperature could lead to more significant advances. For example, thermal conductivity at ultralow temperature, *i.e.*, below the tens of millikelvins that are typically achievable in a dilution refrigerator, may help clarify the ground state in quan-

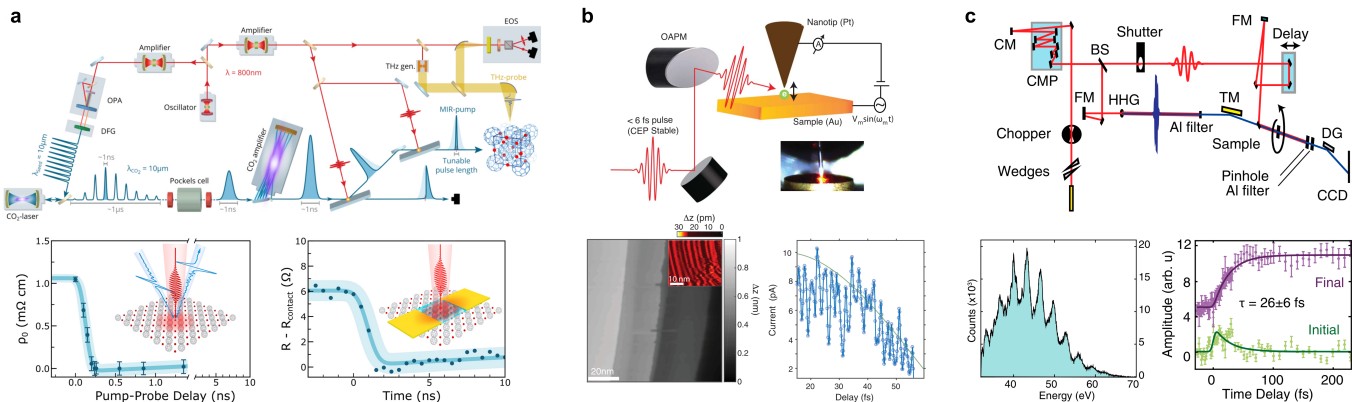

FIG. 18 Frontiers of nonequilibrium spectroscopies and microscopies. (a) Signatures of MIR pulse-induced superconductivity in $K_3C_{60}$ at the nanosecond timescale. *Top*: Schematic of the setup that produces pulses of duration variable between 5 ps and 1.3 ns, centered at 10.6 $\mu$m wavelength, and with a pulse energy of up to 10 mJ. *Bottom*: Transient resistivity obtained from extrapolation of optical conductivity (left) and 2-point transport measurement (right). From Ref. Budden *et al.*, 2020. (b) Attosecond coherent field-driven STM. *Top*: Schematic and photo of the setup where < 6-fs carrier-envelope phase (CEP)-stable pulses are focused at a Pt/Ir tip. *Bottom left*: Surface topography on a Au surface, generated solely by laser-induced tunneling electrons. Inset shows atomic reconstruction on the Au surface. *Bottom right*: Laser-induced tunneling current for various pump-probe delays, featuring a 1.7-eV plasmon mode in a Au nanorod. From Ref. Garg and Kern, 2020. (c) Table-top attosecond XUV spectroscopy. *Top:* Schematic of the pump-probe setup using few-fs CEP-stable NIR pump and attosecond probe. The probe is produced by high harmonic generation (HHG) and its spectrum spans from 30 to 60 eV (bottom left). *Bottom right*: Characteristic timescale of insulator-metal-transition in $VO_2$ ($26 \pm 6$ fs) revealed by absorption changes from the vanadium $M_{2,3}$ edge. From Ref. Jager *et al.*, 2017.

tum spin liquid candidate materials like herbertsmithite if samples of sufficient quality become available.

### 2. High electric/magnetic field

Strong electric fields, up to $\sim$1 V/Å, constitute another knob for tuning phase transitions in strongly correlated systems. In the dc limit, electrostatic gating of ultrathin materials has enabled the precision control of carrier concentration and band structure (Goldman, 2014). In the ac limit, strong fields in mid-infrared or terahertz laser pulses have led to new dynamical states of matter, such as symmetry-breaking or topologically nontrivial phases, some of which do not exist at equilibrium (McIver *et al.*, 2020; Salén *et al.*, 2019). Furthermore, nonlinear response associated with extreme fields offers a sensitive probe of symmetry (Torchinsky and Hsieh, 2017), topology (Sodemann and Fu, 2015), electron correlation (Silva *et al.*, 2018), and perhaps spin fractionalization (Wan and Armitage, 2019). Therefore, access to high electric fields is instrumental in both manipulating and measuring properties of many correlated systems.

With 100 T reached in pulsed field at Los Alamos National Laboratory in 2012 (and now available in other high field laboratories) as the highest non-destructive magnetic field ever realized, these last decades have witnessed great advances in magnetic field technologies (Battesti *et al.*, 2018). The development of Megagauss magnets (semi-destructive fields where the coil is destroyed at each pulse but the sample space preserved, see Fig. 19), which are able to reach as high as $\sim$

300 T (Portugall *et al.*, 1999), foresees brand new kinds of experiments in condensed matter. High magnetic fields have already proved themselves to be an effective tool in disentangling competing and coexisting states of matter. For example, the normal state of several unconventional superconductors, like high-$T_c$ cuprates, has been extensively studied down to low temperatures by suppressing the superconductivity with magnetic field (Proust and Taillefer, 2019; Shi *et al.*, 2020). High magnetic fields can also be extremely useful to observe quantum oscillations that can give a direct measurement of the fermiology and the quasiparticles behavior in quantum materials (Sebastian and Proust, 2015). High magnetic fields allow the stabilization of a remarkable field induced unconventional superconducting phase in $UTe_2$ (Fig. 4) that has the highest upper and lower critical fields of any field-induced superconducting phase (more than 40 T and 65 T respectively) (Ran *et al.*, 2019a,b). High magnetic fields are also essential to probe integer and fractional quantum Hall effects in two-dimensional electron systems, and are expected to continue to play a critical role as the field of 2D materials is developed further (Dean *et al.*, 2011). Moreover, they allow the study of the "quantum limit", in which all charge carriers are confined to the lowest Landau level (Moll *et al.*, 2016b), a particularly pertinent state to explore in correlated topological materials. Unfortunately, magnetic field cannot be applied in traditional ARPES experiments, although photoemission experiments with large currents ($10^6$A/cm$^2$) have been performed (Kaminski *et al.*, 2016). For at least that application on superconductors, they perhaps induce a related

| Technique | Low temperature | Pressure | dc B field | Pulsed B field | Ultrafast E field[§] |
|---|---|---|---|---|---|
| Electrical transport | 6 mK (Pan *et al.*, 2008) | 200 GPa (Drozdov *et al.*, 2015) | 45 T (Fang *et al.*, 2020) | 95 T (Ramshaw *et al.*, 2018) | 400 kV/cm (McIver *et al.*, 2020) |
| Thermal transport | 50 mK (Toews *et al.*, 2013) | 50 GPa (Hohensee *et al.*, 2015) | 45 T (Grisson-nanche *et al.*, 2014) | | |
| Heat capacity | 0.6 mK (Greywall, 1986) | 4.4 GPa (Zheng *et al.*, 2014) | 45 T (Riggs *et al.*, 2011) | 60 T (Terashima *et al.*, 2018) | |
| Magnetic properties | 0.2 mK (Prakash *et al.*, 2017) | 20 GPa (Jackson *et al.*, 2005) | 45 T (Jaime *et al.*, 2012) | 75 T (Zuo *et al.*, 2015) | 9 MV/cm (Schlaud-erer *et al.*, 2019) |
| Broadband FTIR | 0.15 K[¶] | 16.5 GPa (Chal-lener and Thomp-son, 1986) | 35 T (Brinzari *et al.*, 2013) | | |
| Broadband NIR | | 400 GPa (Loubeyre *et al.*, 2020) | 35 T (Brinzari *et al.*, 2013) | 74 T (Zaric *et al.*, 2006) | |
| Raman and PL | 20 mK (PL) (Hayne *et al.*, 1999) | 1 TPa (Ra-man) (Dubrovin-skaia *et al.*, 2016) | 45 T (Raman) (Kim *et al.*, 2013) | 89 T (PL) (Crooker and Samarth, 2007) | |
| Time-domain THz | 0.4 K (Curtis *et al.*, 2016) | 34.4 MPa (Zhang *et al.*, 2017b) | 25 T (Baydin *et al.*, 2020) | ∼30 T (Baydin *et al.*, 2020) | 70 MV/cm (Schu-bert *et al.*, 2014) |
| X-ray | 220 mK (Suzuki *et al.*, 2002, 2004) | 1 TPa (Dubrovin-skaia *et al.*, 2016) | 10 T (Paolasini *et al.*, 2007) | 43 T (Narumi *et al.*, 2012) | 1 MV/cm (Kozina *et al.*, 2019) |
| Neutron | 30 mK (Ross *et al.*, 2011) | 94 GPa (Boehler *et al.*, 2013) | 15 T (inelas-tic) (Council, 2013)[*] | 40 T (diffrac-tion) (Duc *et al.*, 2018) | |
| EPR/ESR | 1.4 K (Takahashi and Hill, 2005) | 2.5 GPa (Sakurai *et al.*, 2015) | 45 T (Takahashi and Hill, 2005) | 63 T (Zvyagin *et al.*, 2011) | |
| NMR | 20 mK (Pustogow *et al.*, 2019) | 90 GPa (Meier *et al.*, 2018). | 45 T (Frachet *et al.*, 2020) | 56 T (Tokunaga *et al.*, 2019) | |
| ARPES | 1 K (Zabolotnyy *et al.*, 2012) | | $10^6$ A/cm$^2$ (Kamin-ski *et al.*, 2016)[†] | | 25 kV/cm (Reimann *et al.*, 2018) |
| EELS | 10 K (Zhao *et al.*, 2018b) | | | | |
| STM | 10 mK (Song *et al.*, 2010) | | 34 T (Tao *et al.*, 2017) | | 100 MV/cm (Garg and Kern, 2020) |
| STEM | 4.5 K (Behler *et al.*, 1993) | | | | |
| SNOM | 20 K (Yang *et al.*, 2013) | | 7 T (Yang *et al.*, 2013) | | |
| MIM | 450 mK (Allen *et al.*, 2019) | | 9 T (Ma *et al.*, 2015) | | |

TABLE I: Present limits of extreme environments used in the study of strongly correlated systems. The values listed are the highest or lowest implemented to the best of our knowledge. The list is not meant to be exhaustive, but to highlight areas of recent activity or avenues for future improvement. Empty cells indicate that the environment is either incompatible with the technique or no reports were found. Here we have not included the considerable research into destructive fields that can reach up to 1200 T (for a controlled explosion) (Nakamura *et al.*, 2018), that enables even higher pulsed magnetic fields for electrical transport and magnetic property measurements. To explore current capabilities of high field measurements in U.S. national laboratories, one can also refer to the NHMFL website https://nationalmaglab.org/. [§]dc or quasi-dc electrostatic gating on 2D materials which can attain a field exceeding $\geq 1$ MV/cm, is compatible with most environments listed. Here, numbers and references are limited to E fields generated by an ultrashort light pulses. [¶]https://kbfi.ee/chemical-physics/research-facilities/?lang=en. [*]HZB did have 26 T steady field (Prokeš *et al.*, 2017), but that facility is now closed. The highest field inelastic facility in the world is now at ILL at the modest 15 T. [†]Magnetic field cannot be applied in traditional ARPES, but the large currents that were applied in the referenced experiment perhaps induce a related effect.

effect. Magnetic fields can be applied in another novel     momentum- and energy-resolved tunneling spectroscopy

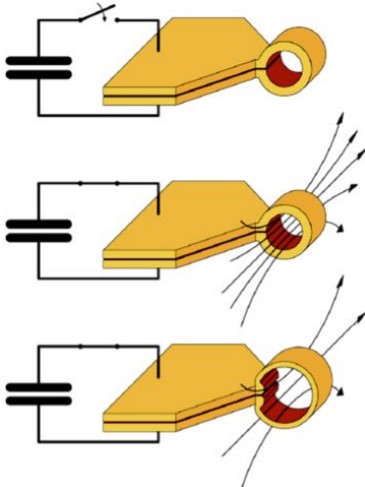

FIG. 19 Sketch of a single-turn coil used in Megagauss facilities to reach magnetic fields significantly higher than 100 T in pulsed field (up to $\sim$ 300 T in a 5-mm-diameter bore) for a few microseconds. A current pulse of $\sim$ 3 MA from a capacitor bank destroys the coil as it generates the field (a semi-destructive technique). From Ref. (Battesti *et al.*, 2018).

technique that also probes single-particle spectral functions (Jang *et al.*, 2017).

There are of course many other interesting applications of large fields not listed here, especially in the domain of magnetism. We need new technological developments to access even higher fields for longer periods of time, for example to study the normal state of some cuprates with a large upper critical field (YBCO, Hg1201, Bi2212, etc.). These could be obtained thanks to advances in the megagauss technology that currently reaches around 150–200 T, but are not widely developed to date. Improvements in the pulse duration and the cooling time for standard pulsed field magnets would also help, respectively by expanding the types of measurements that can be achieved in pulsed field (thermal transport for example is more challenging than electrical transport, see Table I) and by increasing the number of data points taken. Finally, advances in all-superconducting dc magnets, as the 32 T magnet in service at NHMFL since 2017, will avoid the vibrations due to water cooling used for resistive coils, allowing new types of vibration-sensitive experiments, and reduces operating costs. These types of magnets though remain challenging to implement because of the use of high-$T_c$ superconducting coils that are delicate to manufacture.

### 3. High pressure/strain

Application of hydrostatic pressure is one of the cleanest ways to continuously tune the interplay between spin, charge, lattice and orbital degrees of freedom. Due to the often competing interactions in correlated systems,

novel electronic and magnetic phenomena can be quickly masked by the presence of disorder induced by chemical substitution. This is a well-known problem in investigations that attempt to tune correlations using chemical doping. The best examples of materials where experiments under high pressure have contributed a wealth of information for understanding quantum phase transitions are heavy-fermions (Chen *et al.*, 2016b; Si and Steglich, 2010). Not only can 3D interactions may be fine-tuned with pressure, but also the role of inter-planar coupling, *e.g.* for organic superconductors, can be studied via pressure-induced dimensional crossover, such as quasi-1D to quasi-2D, quasi-2D to 3D (Nagata *et al.*, 1998; Pashkin *et al.*, 2010; Valla *et al.*, 2002; Zhang *et al.*, 2019b). Pressure has also proven to be powerful in tuning quantum critical points and quantum spin liquid states in magnetically frustrated systems (Dressel, 2011; Klein *et al.*, 2018; Mirebeau *et al.*, 2002; Powell and McKenzie, 2011). However due to the stringent requirements such as small sample volume, space for pressure cells, etc., there is a constant need for development to improve high-pressure generation technologies. One successful example is the combination of high pressure diamond anvil cell technology with synchrotron X-ray techniques. Here improved source brilliance and small focus sizes allows diffraction, scattering, and spectroscopy in ways not allowed previously (Shen and Mao, 2016; Wang *et al.*, 2019). IR spectroscopy at synchrotron beamlines also allows advantages for high pressures (Kimura and Okamura, 2012; Piccinini *et al.*, 2005). There have been recent successes with combining NMR experiments with pressures as high as 90 GPa (Meier *et al.*, 2018) and neutron diffraction experiment above 90 GPa (Boehler *et al.*, 2013). Although earth sciences and other areas of condensed matter physics have mastered the art of general high pressures using diamond anvil cells, combining very high pressure with many of the techniques used for quantum materials has been prohibitive. For example, combining high pressure with highly surface sensitive techniques such as ARPES has remained completely unaccomplished for obvious reasons.

As most correlated electron systems are anisotropic, hydrostatic pressure is often not the ideal tuning parameter. Therefore, uniaxial strain is becoming one of the most commonly used technique to study electronic and magnetic properties in correlated systems (Mito *et al.*, 2012; Tarantini *et al.*, 2011; Tokunaga *et al.*, 2008). Furthermore, uniaxial and/or biaxial strain/pressure allow us to overcome some of the limitations of hydrostatic pressure generation methods. In some cases such as superconducting heavy fermions, it is even desired to increase the structural anisotropy to tune $T_c$ (Oeschler *et al.*, 2003). Uniaxial strain has been successfully combined with experimental probes for which the application of hydrostatic pressure is currently not possible. For example, experimental probes such as STM, SEM and ARPES can use uniaxial strain on single crystals or thin films to tune the system as a substitute for hydrostatic

pressure (Flötotto *et al.*, 2018; Riccò *et al.*, 2018; Trainer *et al.*, 2019). Using the substrate of thin films to create unaxial or biaxial strain that manipulates material properties is a well established technique in the field of correlated electron systems such as cuprates (Abrecht *et al.*, 2003), manganites (Liao *et al.*, 2014) and titanates (Zhang *et al.*, 2013), but the recent focus has been on using strain as a continuous in situ probe of materials with piezoelectric stacks. This has been proven to be extremely powerful when probing nematic correlations in the pnictides (Chu *et al.*, 2012). For instance, Ref. Chu *et al.*, 2012 showed how measurement of the divergent nematic susceptibility of the iron pnictide superconductor $Ba(Fe_{1-x}Co_x)_2As_2$ can distinguish an electronic nematic phase transition from a simple ferroelastic distortion. In situ strain and STM have been combined to show how nematic fluctuations and nematic order in an iron-based superconductor change across the phase diagram (Andrade *et al.*, 2018). It was shown that sizable nematic correlations persist to high temperatures and that there is strong *nonlinear* coupling between structure and electronic nematicity even at temperatures above the tetragonal to orthorhombic transition.

### 4. Challenges and outlooks

Despite the rapid progress in pushing various experimental limits, probes in *multiple* extreme environments are uncommon due to the high level of experimental difficulty. We believe that efforts should be undertaken to push boundaries in these experimental techniques by both integrating setups with new extreme environments, and by combining multiple environments. Due to the complexity of many-body interaction in correlated electron material, it is often necessary to leverage more than one extreme environment. For instance, the combination of high magnetic fields up to 60 T and hydrostatic pressure up to 4 GPa in $URu_2Si_2$ has recently shed new light on the subtle competition between the hidden-order state and neighboring magnetically ordered states (Knafo *et al.*, 2020). In some situations, one extreme environment may be employed to help access a phase boundary within the experimental limits of a second environment. For example, on YBCO, the extremely high critical fields of 150 T at optimal doping hindered investigations of the normal state at low temperature. With the application of high pressure, however, critical field values can in principle be lowered to a more accessible field regime, thereby allowing studies of the quantum phases below the superconducting transition. The use of this extreme environment, acting similarly as doping, enabled for instance the access to the entire overdoped regime in pristine YBCO by lowering $T_c$ and moving the end of the superconducting dome (Alireza *et al.*, 2017). This approach also minimizes effects from varying the sample quality and the environments in different set-ups or laboratories, as multiple experiments are performed simultaneously on the same sample.

The experimental complexity of many advanced scattering, spectroscopy, and microscopy measurements also poses great challenges for accessing extreme sample conditions. For example, static magnetic fields for inelastic neutron scattering are still nowadays limited to a modest field of 15 T at the Institut Laue-Langevin[7]. Implementing extreme conditions in these setups necessitates a collaborative effort to surmount numerous engineering challenges, but we expect that such endeavor is of great interest to the community and will pay off in the long run.

## IV. CONCLUDING REMARKS

In this manuscript, we have attempted to lay out the results of our discussions on the Future of the Correlated Electron Problem. These are hard problems and progress on them takes time. But progress has cycles and going forward it will make sense to stubbornly come back to stubborn old problems with new ideas.

We have so far stayed away from sociological and philosophical aspects surrounding the Future of the Correlated Electron Problem, but such discussions frequently surfaced during the workshop. Issues such as questionable reproducibility, over-analysis/interpretation, over-abundance of jargon, pressure to produce high-impact publications, and a pursuit of "novelty" and therefore a lack of systematic studies, are definitely not unique to the correlated electron community. Nonetheless, both senior and junior scientists in this field should make a concerted effort to address these issues, which only complicate the already complex problems. As a community, we can take simple, concrete steps, and one example would be giving more credits to reporting conflicting or null results. Indeed, recently we have seen several high-profile cases concerning reproducibility of experimental data in the literature *e.g.* triplet-pairing in superconducting $Sr_2RuO_4$ (Ishida *et al.*, 1998; Pustogow *et al.*, 2019), giant current-induced diamagnetism in $Ca_2RuO_4$ (Sow *et al.*, 2017; Zhao *et al.*, 2019), and chiral Majorana fermions in a quantum anomalous Hall-superconductor device (He *et al.*, 2017; Kayyalha *et al.*, 2020). These examples serve to remind us of the high standards and open debate are important when pushing forward in this field.

We have made the above forecasts, predictions, and recommendations not from an expectation that we will

---

[7] The Helmholtz-Zentrum-Berlin facility that used to have a 26 T steady field magnet (Prokeš *et al.*, 2017), but it is now shut down.
https://www.helmholtz-berlin.de/pubbin/news_seite?nid=14076;sprache=en;seitenid=74699
https://www.helmholtz-berlin.de/projects/rueckbau/ber/index_en.html

be ultimately be proven correct. It is of course "difficult to make predictions, especially about the future"[8]. Our hope, however, is that the topics we have presented will provide inspiration for others working in this field and motivation for the idea that significant progress can be made on very hard problems if we focus our collective energies. Irrespective of the particular path taken, it is clear that the Future of the Correlated Electron Problem will be full of fascinating physics and unexpected twists and turns that will challenge us for years to come.

## V. ACKNOWLEDGEMENTS

We thank NSF CMP program for suggestions regarding the topic and general structure of the workshop. This project was supported by the NSF DMR-2002329 and The Gordon and Betty Moore Foundation (GBMF) EPiQS initiative. We would like to sincerely thank A. Kapitulnik, A. J. Leggett, M.B. Maple, T.M. McQueen, M. Norman, P. S. Riseborough, and G. A. Sawatzky for their lectures at the workshop and advice on the writing of this manuscript. We would also like to thank G. Blumberg, C. Broholm, S. Crooker, N. Drichko, and A. Patel for helpful consultation on topics discussed herein. A number of individuals also had independent support: (AA, EH; GBMF-4305 and GBMF-8691), (IMH; GBMF-9071), (HJC; NHMFL is supported by the NSF DMR-1644779 and the state of Florida), (YH, AZ; Miller Institute for Basic Research in Science), (YC; US DOE-BES DE-AC02-06CH11357), (AS; Spallation Neutron Source, a DOE Office of Science User Facility operated by ORNL), (SAAG; ARO-W911NF-18-1-0290, NSF DMR-1455233), (YW; DOE-BES DE-SC0019331, GBMF-4532).

**Competing interests:** The Authors declare no Competing Financial or Non-Financial Interests.

**Data availability:** No data sets were generated or analysed during the current study.

**Author contributions:** All authors contributed to the writing of the manuscript. NPA organized the workshop and edited the manuscript.

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
