# Peer review of "The Future of the Correlated Electron Problem"

_SciPost Physics Community Reports, doi:SciPost Phys. Comm. Rep. 8 (2025)_

## Round 3 · Referee Report · Anonymous (Referee 1) · 2022-12-12

Strengths

1) An expertly written outlook on the future of the field of correlated electron systems (CES). Despite the very large number of co-authors, the structure and the language throughout the manuscript is coherent. 2) I do believe this is an important document that will serve as an interesting and informative reference for the CES community in the coming years. It is refreshing to find that so many of the co-authors are the more junior members of the community and Prof. Armitage is to be applauded, not only for taking the initiative of this, but also for striking an excellent balance of contributors.

Weaknesses

There are only a couple of minor points to make here: 1) While the manuscript covers a wide range of topics within the field, I felt that it could have benefitted to from a dedicated discussion about interacting quasi-one-dimensional materials, in particular the breakdown of either the Luttinger liquid or the Fermi-liquid as the dimensionality of the electronic ground state is tuned. Given that both end points are well understood, yet their crossover is still so strongly disputed, this aspect epitomizes the challenge the community faces in building up a complete picture of the interacting many-body problem. 2) While many of the figure captions were sufficiently self-contained, I felt that those for Figures 8, 10, 13 and 14 could benefit from being expanded to make them more self-contained. This is particularly important for those figures that have only a cursory mention in the main text. 3) I spotted a number of typos throughout the text (no more than usual). I would be happy to send the list to the authors if that would be helpful.

Report

I recommend that the paper should be published with only the minor changes highlighted above.

Requested changes

I doubt that the authors could address point 1) above, given that it is unlikely to have featured at their workshop, so I am happy if the authors simply address my point 2) above regarding some of the figure captions.

  • validity: high
  • significance: high
  • originality: high
  • clarity: high
  • formatting: excellent
  • grammar: excellent

Author:  N. Peter Armitage  on 2024-12-29  [id 5070]

(in reply to Report 1 on 2022-12-12)
Category:
answer to question
reply to objection

Dear Reviewer and Editor,

We thank the referee for the helpful comments. We apologize for the exceedingly long time we have taken to get back to SciPost on this manuscript. Various intervening life events meant that we were slow.

We appreciate all 3 points brought up by the referee.

1) Indeed 1D systems did not feature prominently in the workshop. And as we had mentioned in the manuscript it was not our intention to write a review. As we wrote, "It is important to note that this manuscript is {\it not} a review and no attempt to be complete has been made. The topics and opinions expressed are idiosyncratic, and reflect the particular interests and preferences of the people who spoke at and attended the workshop." We agree that 1D systems are important test cases and host to tremendously interesting phenomena by themselves. But for whatever happenstance they did not feature prominently in the workshop. 2) We have added additional clarifying comments to captions for Figures 8, 10, 13 and 14 as requested by the referee. We have also added additional information to Figures 11, 12, 19 3) We have read through again and corrected a number of typographical errors.

A new version of the manuscript has been submitted to the arXiv.

In rereading through the manuscript now, we continue to think the manuscript has tremendous value both in its scientific content as well as an example of how a manuscript can be written in a distributed fashion using modern tools. And of course the exercise of writing it was particularly important for the young people involved. It is important to restate that all co-authors except for NPA were "young" researchers at the time of the writing e.g. senior grad students, postdocs, and early career assistant professors. I (NPA) got many emails from other young scientists who found the message inspiring and useful for their research. I am also impressed about how prescient the choices of science were and how many areas are still promising and in indeed in the future. The science inside is still largely "the future of the correlated electron problem".

I hope with the requested changes it can be published in SciPost without further delay. Thank you for consideration.

Very best regards,

N. Peter Armitage for his co-authors

Anonymous on 2025-01-06  [id 5086]

(in reply to N. Peter Armitage on 2024-12-29 [id 5070])

I am satisfied with the changes made by the authors and am happy to recommend publication of this important work in SciPost.

---

## Round 4 · Author Response

We have replied to the referee and already gotten a favorable reply. We have resubmitted the manuscript under SciPost Physics Community Reports. We hope that it can be published soon.

---

## Round 4 · List of Changes

We have also added additional information to Figures 11, 12, 19.
We have read through again and corrected a number of typographical errors.

---

## Editorial Decision

published